evolution

behavioural divergence, birdsong, urbanization, evolution, song frequency shifts

**Author for correspondence:**
Pamela J. Yeh
e-mail: pamelayeh@ucla.edu

†These authors contributed equally to this work.

# No evidence of repeated song divergence across multiple urban and non-urban populations of dark-eyed juncos (*Junco hyemalis*) in Southern California

Felisha Wong[1,†], Eleanor S. Diamant[1,†], Marlene Walters[1] and Pamela J. Yeh[1,2]

[1]Department of Ecology and Evolutionary Biology, University of California, Los Angeles, CA 90095, USA
[2]Santa Fe Institute, Santa Fe, NM 87501, USA

FW, 0000-0003-0739-3620; ESD, 0000-0003-1727-5855

Urbanization can affect species communication by introducing new selection pressures, such as noise pollution and different environmental transmission properties. These selection pressures can trigger divergence between urban and non-urban populations. Songbirds rely on vocalizations to defend territories and attract mates. Urban songbirds have been shown in some species and some populations to increase the frequencies, reduce the length and change other temporal features of their songs. This study compares songs from four urban and three non-urban populations of dark-eyed juncos (*Junco hyemalis*) throughout Southern California. We examined song length, trill rate, minimum frequency, maximum frequency, peak frequency and frequency bandwidth. We also compared songs recorded from one urban junco population in San Diego nearly two decades ago with recently collected data in 2018–2020. Over all comparisons, we only found significant differences between UCLA and the 2006/2007 UCSD field seasons in minimum and maximum frequencies. These findings partially support and are partially in contrast to previous urban song studies. As urban areas expand, more opportunities will arise to understand urban song divergence in greater detail.

# 1. Introduction

Urbanization presents a growing threat to wildlife as it alters natural ecosystems and creates new selection pressures [1–3]. For birds that use song for territory defence and mate attraction, urban selection pressures may affect song communication due to differences in habitat structure and transmission properties of artificial surfaces. Additionally, intense, low-frequency, ambient noise from cars, aeroplanes and machinery can restrict the range over which songs can be heard by potential mates and rivals [4–12].

Urban landscapes are evolutionarily more novel in comparison to rural and wild areas [8,9,13]. Urban landscapes often have higher proportions of hard, flat surfaces (e.g. high buildings and pavement) [14,15], reduced vegetation complexity [15–19] and potentially more open spaces [20,21]. These unique environmental properties may influence songbirds to adjust the features of their songs to maximize transmission. Echoes and reverberations reflected by vertical surfaces have been found to correlate with longer songs sung at lower frequencies [22–24], while faster trill rates may benefit singers in open environments [16,25–27]. Furthermore, the typically dense vegetation found in mountainous areas absorbs and scatters higher frequencies, suggesting that higher frequencies transmit better in urban areas, where vegetation cover is lower [4,22,28,29]. As a result, researchers have hypothesized that urban songbird populations will broadly exhibit longer song lengths, reduced trill rates, and higher song frequencies [4,16,28].

Aside from changes to the landscape composition in urban environments, noise pollution has also been found to change avian song characteristics. Because urban ambient noise is typically high amplitude and low frequency (below 2000 Hz) [30], some urban songbirds sing songs with increased frequency, reduced frequency bandwidth and longer length to avoid auditory masking around vehicle traffic and other artificial sources of noise [30–39]. However, some studies have found that increasing frequencies might only yield a small increase in signal transmission [40] and may actually be detrimental to an individual's vocal performance [41]. Other studies in urban areas have found increases in other song traits such as amplitude, as Lombard effects—the tendency to increase amplitude in response to noise—are known to be common among animals that communicate vocally [42–44]. Furthermore, higher song frequency may be generally associated with higher amplitude [45]; for this reason, some bird species may be using higher frequency song elements because they are also of a higher amplitude. However, oscine species, including dark-eyed juncos (*Junco hyemalis*), likely sing at higher frequencies independent of singing at higher amplitude in response to noise [46,47]. Song length [48] has also been found to increase in response to varying levels of noise without a corresponding change in frequency.

Frequency shifts as well as song length differences have been found in comparisons between urban and non-urban populations of the same species [16,20,49–54]. The results among different studies of the same species, however, are not always consistent. One study found frequency shifts between urban and rural populations of song sparrows (*Melospiza melodia*) in Portland, Oregon [55]; another found no frequency shifts among urban and rural song sparrows in the Washington DC and Baltimore area [56]. This difference in results could potentially be attributed to the geographical distance between these two studies, and therefore differences in dialects which may interact with noise levels differently [55,56].

Here, we study the songs of multiple populations of dark-eyed juncos (hereafter, 'juncos') in both urban and non-urban sites across Southern California. While studies conducted on urban junco populations have illustrated rapid evolutionary changes in physiology, morphology and behaviour over the course of just a few decades [1,57–61], most of these studies are limited to one urban population—San Diego, California [1,56–58,60]—which was likely colonized by juncos in the 1980s [62]. Over the past twenty years, juncos have colonized Los Angeles (likely in the early 2000s [63]) (P. Yeh 2008, personal observation) and Santa Barbara, as well as cities across the western North American coast [64]. Little is understood about how traits diverge or converge across multiple urban populations of juncos in comparison to non-urban populations and how those traits might have changed over time.

Urban song in juncos has been particularly well studied in San Diego [16,49,65], making song traits especially relevant to explore in other urban populations. Junco song can be described as a simple, repeated trill with a repertoire size of approximately two to eight song types, though singing males usually use only one or two song types per bout with minimal song type sharing between males [49,66,67]. Though females have sometimes been noted to sing territorially [68], it is mostly male juncos that commonly sing for territory defence and mate attraction [67,69]. There is substantial variation in certain elements of their song and in song type use between different populations

[16,49,66,67,70,71]. This variation allows juncos to produce different songs based on the environment they are in [61]. For example, Slabbekoorn *et al.* [16] and Newman *et al.* [49] found that one population of San Diego urban juncos sang shorter songs compared to songs from the nearby mountain population. Slabbekoorn *et al.* [16] found differences in minimum frequency, while Newman *et al.* [49] reported differences in maximum frequency. Mechanistically, song variation in juncos has been explained by social learning, novel acquisition [65,66], and genetic or early life effects. In San Diego, Reichard *et al.* [72] investigated the mechanism through which one urban and one non-urban population diverged via a common garden experiment and demonstrated that song frequency changes are maintained through either genetics, parental effects, or early life exposure to noise. Together, this work raises the question of how and if junco song traits across urban populations adjust in convergent ways in response to urban noise stressors.

In this study, we specifically examine (1) whether the songs from urban and non-urban male juncos from locations across Southern California differ in six song characteristics: song length, trill rate, minimum frequency, maximum frequency, peak frequency and frequency bandwidth; (2) whether songs from two urban locations (UCSD 2006/2007 and UCLA 2018–2020) at the same time-since-colonization (approx. 15–20 years later) were similar; and (3) whether there were changes in song in one urban population—the San Diego population—over the course of more than a decade, using songs from 2006/2007 and 2018–2020 breeding seasons.

# 2. Material and methods

## 2.1. Study sites

Recordings were taken from juncos at seven different Southern California locations: (1) University of California Los Angeles (UCLA; 34°4′10″ N, 118°26′43″ W), (2) University of California San Diego (UCSD; 32°52′30.95″ N, 117°14′10.08″ W), (3) University of California Santa Barbara (UCSB; 34°24′52.57″ N, 119°50′44.92″ W), (4) Occidental College (34°07′40.80″ N, 118°12′39.60″ W), (5) the Angeles National Forest (34°18′33.88″ N, 117°57′31.79″ W), (6) the UC Stunt Ranch Santa Monica Mountains Reserve (UC Stunt Ranch Reserve; 34°5′27″ N, 118°39′27″ W), and (7) the UC James San Jacinto Mountains Reserve (UC James Reserve; 33°48′30″ N, 116°46′40″ W) (figure 1). We used HowLoud [73] to estimate the amount of noise present in each location using the soundscore computing system, where the lower the soundscore, the noisier the location. These estimates are determined using models rather than measured on the ground. HowLoud models ambient noise from traffic, proximity to airports, and local sources. Traffic noise is estimated using an established transportation model by the Federal Highway Administration [74]. Together, the sources' noise levels (dB) are averaged over a 24 h period and converted into a scaled 'soundscore' of ambient noise [73]. This tool has been used in environmental health research [75] but has not yet been used in song research. Noise is scaled from 50 (very loud) to 100 (very quiet), with higher soundscores having lower average ambient noise. The first four locations mentioned are urban environments, with soundscores between approximately 50 and 80, while the last three locations are considered non-urban, mountain environments, with soundscores between approximately 90 and 100. The urban locations are all college campuses situated in cities with high pedestrian traffic that fluctuates throughout the day, depending on class schedules. Student population density varies across campus sites (UCLA = approximately 107.27 students/acre; UCSD = approximately 36.72 students/acre; UCSB = approximately 26.61 students/acre; Occidental College = approximately 17.125 students/acre) [76–79]. In terms of building composition, each campus varies in building height and density. UCSB, Occidental College, and UCSD have buildings that are relatively spaced apart, while UCLA has many tall buildings that are closer together. Each campus also has a different density of forest-like areas. The non-urban environments have relatively low pedestrian traffic, a mix of chaparral shrub forest vegetation (UC Stunt Ranch Reserve and portions of the Angeles National Forest), open woodland (portions of the Angeles National Forest and the UC James Reserve), and sparse, low buildings. All seven of these sites were chosen due to the presence of juncos and their location in Southern California.

## 2.2. Song recordings

Recordings were taken during juncos' breeding seasons during and after the establishment of their territories. UCLA, UCSD, UCSB and Occidental College juncos were recorded from January to June

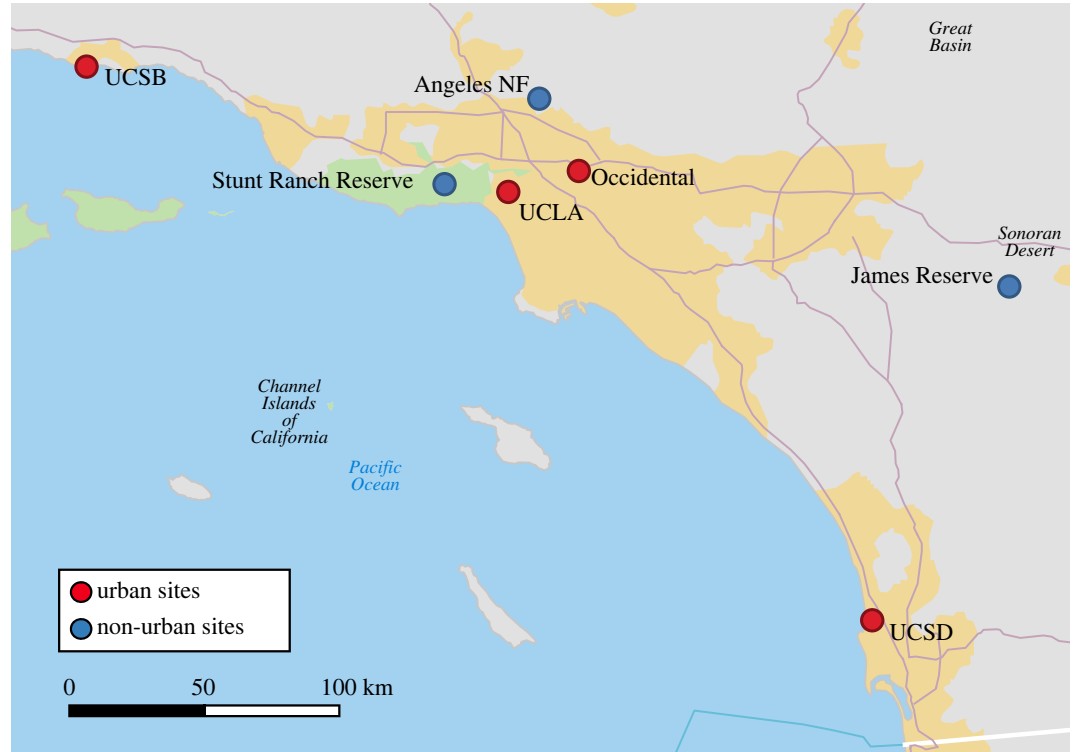

**Figure 1.** Map of all seven locations where male juncos were recorded: UC Los Angeles, UC Santa Barbara, UC San Diego, Occidental College, UC James Reserve, UC Stunt Ranch Reserve and the Angeles National Forest (Angeles NF). A blue dot indicates the location is a non-urban site while a red dot indicates an urban site.

2018, 2019 and 2020 from 6.00–12.00 h. Juncos at the UC Stunt Ranch Reserve, the Angeles National Forest and the UC James Reserve were recorded from mid-April to July 2018 and 2019 from 6.00–12.00 h. Recordings were made using a Marantz PMD661 solid-state digital recorder, a Sennheiser ME66 omnidirectional microphone, and an Audio-Technica AT815b microphone. All recordings were saved to WAV files using a sampling rate of 44 kHz. All recordings were of spontaneous songs and not in response to playback, which could potentially affect the song's form [71]. We defined song bouts as a series of one or more songs separated by variable intervals of silence [80,81] (figure 2). Each male was recorded for a minimum of 1 to a maximum of 142 songs, and the mean ± s.d. number of songs per male was 23.05 ± 22.26. Mean number of songs varied across populations: urban populations ranged from a mean ± s.d. of 8.67 ± 3.93 (UCSB) to 44.35 ± 35.56 (Occidental College) and non-urban populations ranged from a mean ± s.d. of 8.6 ± 16.59 (the UC James Reserve) to 19.62 ± 9.85 (the Angeles National Forest). Individual males were recorded on multiple days throughout the season. In urban locations, males were identified and differentiated by colour bands ($n = 54$). A subset of males ($n = 55$) was unbanded, but unbanded males consistently defended stable territories neighboured by banded males, which made it possible to differentiate among unbanded individuals. In non-urban locations, some males were individually banded ($n = 9$) and some were not ($n = 12$). Non-urban male juncos maintain stable territories during the breeding season. As such, we ensured that we were recording different individuals by measuring unbanded males that either had a banded neighbour or were greater than 50 m away from the nearest unbanded bird.

Recordings were made opportunistically after first hearing and locating a singing male. Observers carried the recording equipment in hand and approached singing males on foot. Males were then recorded until they stopped singing for a significant period of time (at least 30 s) or flew too far away (outside of an approximately 10 m radius of the microphone) to obtain a high-quality recording. After starting the recording, the observer attempted to close the distance between the bird and the microphone without scaring the individual away to increase the signal-to-noise ratio; however, exact distance between the bird and microphone was not recorded, and noise profiles were not made for each recording. After recording the bird, we marked exact GPS coordinates for each individual recording. The resulting recording dataset consisted of a total of 130 individual males, with the length of each recording ranging from 1 min to more than 1 h. Of the 130 male juncos recorded, 57 were

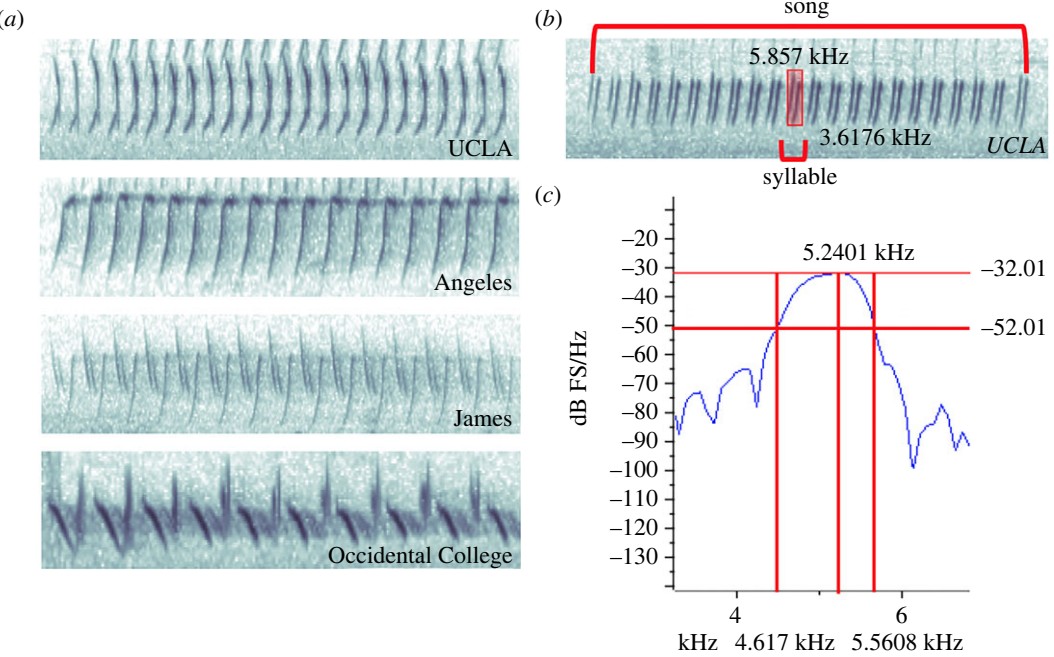

**Figure 2.** (*a*) Spectrograms of a select few song types that were recorded from the corresponding locations. The first two are monosyllabic while the latter two are a combination of monosyllables. (*b*) Annotated spectrogram of what constituted a song. A song bout consists of multiples of these songs. Each individual repeated series of notes is considered a syllable. In the spectrogram for the highlighted syllable, we received minimum frequency (3.6176 kHz) and maximum frequency (5.857 kHz). (*c*) A mean power spectrum of the same highlighted syllable in (*b*). We measured minimum (4.617 kHz) and maximum frequency (5.5608 kHz) 20 dB from the peak. The peak frequency (5.2401 kHz) was found at maximum amplitude.

**Table 1.** Number of dark-eyed juncos recorded at each of the 7 locations visited, plus the total number of urban and non-urban male dark-eyed juncos recorded. The total number of non-urban dark-eyed juncos included 19 natural songs from male dark-eyed juncos in Southern California from the Macaulay Library Repository of Cornell Lab of Ornithology (Cornell Lab of Ornithology, Ithaca, 2020). Total number of song types analysed per habitat type. The following recordings from the Macaulay Library at the Cornell Lab of Ornithology were used: ML104914591, ML141793701, ML158949451, ML165278021, ML165792881, ML166068651, ML166509621, ML166509641, ML171008801, ML47853201, ML47853301, ML104914551, ML165792981, ML54608211, ML62694621, ML210591, ML98625141, ML102524971, and ML103606961.

| location | no. males recorded | habitat type | no. males recorded | no. song types |
|---|---|---|---|---|
| UC Los Angeles | 57 | urban | 109 | 51 |
| Occidental College | 28 | | | |
| UC San Diego 2017–2020 | 17 | | | |
| UC Santa Barbara | 7 | | | |
| UC Stunt Ranch reserve | 3 | non-urban | 21 + 19 = 40 | 27 |
| UC James Reserve | 5 | | | |
| Angeles National Forest | 13 | | | |

from UCLA, 28 from Occidental College, 17 from UCSD, 7 from UCSB, 3 from UC Stunt Ranch Reserve, 5 from UC James Reserve and 13 from the Angeles National Forest (table 1). This gave us a total number of 109 urban juncos recorded and 21 non-urban juncos (table 1).

Because the sample sizes varied substantially between urban and non-urban populations, we supplemented our sample of non-urban songs with recordings from the Macaulay Library song repository [82], which had a total of 1142 dark-eyed junco recordings. Of the 1142 recordings, 42 of them were from locations throughout Southern California and recorded during the dark-eyed juncos'

breeding season between the years 2017 and 2019. We mapped the coordinate locations of all the junco repository recordings from Southern California and then determined whether they were from urban or non-urban sites based on proximity to urbanized areas (urban: less than 10 km; non-urban: greater than 10 km) and whether their location contained high-density vegetation and sparse buildings. In total, we added 19 individual male junco recordings from non-urban mountain sites throughout Southern California from the repository, for a total of 40 non-urban junco individuals (table 1). Distance to microphone was unknown for these data, though how this might affect results is relatively uncertain as the relationship between amplitude and frequency varies between studies [46,83]. It is also important to note that the recording protocol for these recordings is not known and may differ from our methods, and thus may affect the song analysis process. For example, these recordings may have been recorded on a phone, which stores the song in a compressed audio format.

To compare song traits from a population of urban juncos in UCSD from the years 2006/2007 to those from the years 2018–2020, we used an existing dataset of song measurements from 151 individual male juncos from UCSD and the Laguna mountains taken during the 2006 and 2007 breeding seasons [70,84]. Of the 151 males recorded, 101 were from UCSD and 50 were from the Laguna mountains. For the purpose of our study, we only used the measurements from the 101 UCSD juncos.

## 2.3. Data extraction

All songs were transferred into the program Raven Pro 1.5 [85]. They were inputted at a sample rate of 44 100 Hz at 16 bits. All songs were normalized for amplitude, and measurements were taken from a spectrogram and mean power spectrum. The following spectrogram parameters were used for each song: window sample size of 512 points (11.6 ms), hop size = 5.8 ms (frame overlap = 50%), frequency grid spacing = 86.1 Hz (DFT size = 512 samples) and 3 dB bandwidths of 124 Hz. We used a spectrogram analysis and mean power spectrum to obtain: (1) minimum frequency, (2) maximum frequency, and (3) peak frequency of the middle syllable within each song, as well as the (4) song length and (5) trill rate of the entire song. We visually identified the start and end time for each song by creating a bounded selection starting from the onset of the first syllable and ending at the onset of the final syllable of the song. After making the selection box, we counted the number of syllables, subtracted 1, and then divided by the song length to obtain the trill rate. For cross-site comparisons, we performed a mean power spectrum analysis to obtain the minimum frequency, maximum frequency, and peak frequency per song. From the mean power spectrum, we used a threshold of minus 20 dB relative to the peak to find minimum and maximum frequency. We measured minimum, maximum and peak frequency from each song within a song bout. Average minimum, maximum and peak frequencies were calculated across all songs of the same song type from all males separately for each location. For example, if one song type was used in 10 songs at UCLA—regardless of which male sang it—each raw minimum frequency would be added together, and the total divided by 10. This would produce the average minimum frequency for that song type at UCLA. Song types were visually determined via spectrograms based on their shape. Multisyllabic songs were treated as separate song types. A more detailed visualization into a few different song types, what constituted a song and syllable, and how minimum and maximum frequency were taken from the spectrograms and mean power spectrum are included in figure 2. In total, we analysed 51 urban song types and 27 non-urban song types (table 1). Individual males that we recorded sung a mean ± s.d. of 1.13 ± 0.41 song types. Mean song traits per song type were used as the main unit of replication in our statistical analyses.

For the UCSD 2006/2007 comparison to UCSD 2018–2020 and to UCLA frequency measurements, we used spectrogram analysis and created a bound selection box where we visually identified the lower and upper limits by finding the lowest and highest frequency measurement. We understand the implications of using spectrogram analysis over mean power spectrum analysis for this comparison, as noted in many studies [86–89]. Visually determining measurements from a spectrogram can lack repeatability between individuals and, therefore, account for a larger variation in the dataset than exists in the population. However, because the UCSD 2006/2007 data were extracted using the spectrogram method, we decided to be consistent in our comparisons.

We $\log_{10}$ transformed all frequency measurements obtained to better reflect how sound is perceived in animals [72,90,91]. To obtain the frequency bandwidth, we subtracted the $\log_{10}$ transformed minimum frequency from the $\log_{10}$ transformed maximum frequency.

## 2.4. Analysis

We ran Bayesian linear mixed-effect models (BLMMs) with a Gaussian distribution in R v. 4.0.4 [92] to model the relationship between each individual song trait (song length, trill rate, minimum frequency, maximum frequency, peak frequency and frequency bandwidth extracted from the mean power spectrum) with location as a random effect and an urban or non-urban category as a fixed effect. In all populations where we had a sample size of at least 10 individuals, we found that song features met the assumptions of normal distribution and homoscedasticity. For most analyses, our models were over-fitted and resulted in singularity when using a generalized linear mixed-effect model (GLMM). As such, we ran a Bayesian method using 'blme' [93] to prevent singularity. To correct for multiple tests, we set our alpha to 0.0083. For the urban and non-urban population comparisons, UCLA, Occidental College, UCSD, and UCSB were considered urban sites while the UC Stunt Ranch Reserve, the UC James Reserve, and the Angeles National Forest were considered non-urban mountain sites. The repository data were included in this part of the analysis as part of the non-urban mountain sites grouping.

We further ran a linear model (LM) to test the relationship between each individual song trait (trill rate, minimum frequency, maximum frequency, and frequency bandwidth extracted using a spectrogram analysis) with population (UCLA, UCSD 2006/2007, UCSD 2018–2020) as the fixed effect to determine whether song traits shifted over time and in a different city. We did not run a *post hoc* pairwise comparison because we were interested in comparing UCLA and UCSD to each other at a similar time since colonization (approximately 15–20 years), and UCSD to itself over time. Thus, we only compared UCSD 2006/2007 to UCSD 2018–2020 and UCLA 2018–2020 by making UCSD 2006/2007 level '0' in our LM and thus a baseline to compare to. Because the data obtained from UCSD 2006/2007 were extracted using a spectrogram analysis, we conducted this model separately. To account for multiple tests, we set our alpha to 0.01. We obtained 95% confidence intervals on effect sizes using the R package *lmerTest* [94]. We used the package *emmeans* [95] to convert coefficients into Cohen's *d* values, taking into account variance. As such, we compared *d* values to benchmarks of 'small', 'medium' and 'large' effects ($d = \pm0.2$, 0.5 and 0.8 respectively). For reference, assuming similar standard deviation and sample size, a change in minimum frequency of 500 Hz, as has been reported in previous urban junco song work, would produce an effect size of $d = -0.2$ (95% CI = −0.433, 0.034).

We conducted a power analysis on each model and resulting coefficient estimates using R package *simr* [96]. Here, we calculated power based on 100 simulations. We also used the package *emmeans* [95] to determine approximate coefficient estimates for medium effects [97,98]. We then simulated power based on 100 simulations with adjusted coefficients to determine whether our model would be sufficient at detecting medium effects in song traits. Power analysis results demonstrate the probability that our study could detect (e.g. find significant results for) an effect of a certain size based on our study design, variance and sample sizes [99,100]. We determined the power for the found effect sizes, for medium effect sizes ($d = 0.5$), and for the effect sizes that a 500 Hz minimum frequency shift would cause. Further analyses that test for discrete differences with populations as fixed effects are included in the electronic supplementary material (text S1).

## 2.5. Visualizing urban population expansion

We accessed *eBird* data on dark-eyed juncos from Los Angeles county and San Diego county that were curated by *eBird* to only include reliable observations, decreasing the potential number of inaccurate sightings [64]. We took a subset of these data to only include observations between April and July of each year. Then, we mapped the coordinates of observations in 10-year intervals (2000–2009, 2010–2020).

# 3. Results

## 3.1. Vocalization analysis

The means for all traits were broadly similar among populations and between urban and rural juncos (figure 3). After running a BLMM with each trait as the dependent variable, location as a random effect and urban/non-urban categorization as a fixed effect, we found no significant differences for all song traits. We calculated coefficient estimates with 95% confidence intervals for the effect of 'urban/non-urban' on all traits (song length: $-0.007 \pm 0.12$, $p = 0.91$; trill rate: $0.42 \pm 2.16$, $p = 0.070$; minimum frequency: $0.004 \pm 0.03$, $p = 0.80$; maximum frequency: $0.007 \pm 0.02$, $p = 0.54$; frequency bandwidth:

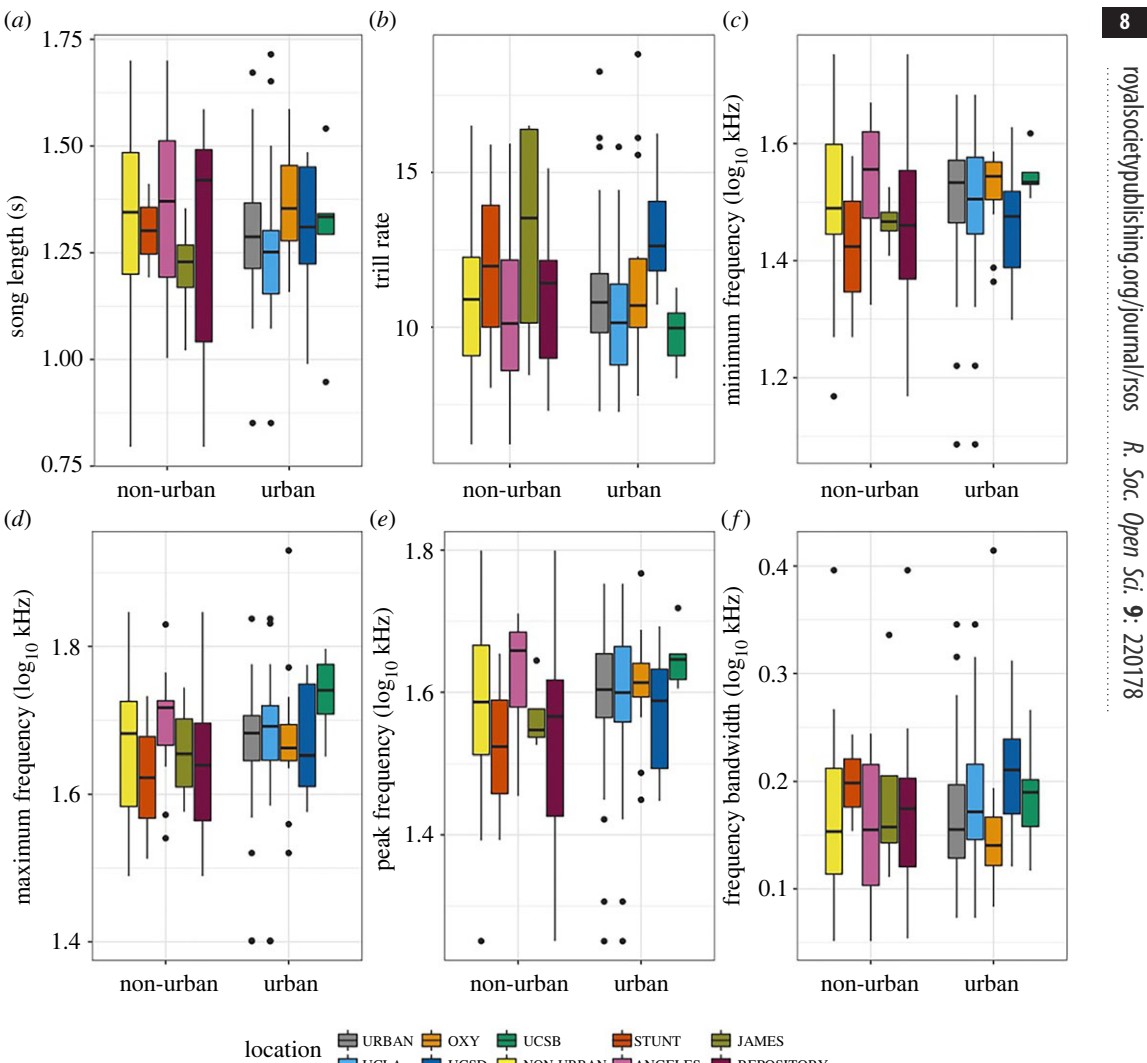

**Figure 3.** Boxplots of differences across sites in each of the different characteristics studied per song type: (*a*) song length in seconds, (*b*) trill rate, (*c*) minimum frequency ($log_{10}$ kHz), (*d*) maximum frequency ($log_{10}$ kHz), (*e*) peak frequency ($log_{10}$ kHz) and (*f*) frequency bandwidth ($log_{10}$ kHz). In all characteristics, there is overlap in the ranges. There are no statistically significant differences in these six characteristics across sites and between non-urban and urban populations (BLMM; $p > 0.05$). Each boxplot is a representation of the distribution of its respective song characteristic for all populations. URBAN = all urban sites ($n = 51$); UCLA = University of California, Los Angeles ($n = 23$); UCSD = University of California, San Diego 2018–2020 field season ($n = 9$); UCSB = University of California, Santa Barbara ($n = 5$); NON-URBAN = all non-urban sites ($n = 41$); OXY = Occidental College ($n = 14$); STUNT = UC Stunt Ranch Reserve ($n = 2$); JAMES = UC San Jacinto James Reserve ($n = 4$); ANGELES = Angeles National Forest ($n = 12$); REPOSITORY = Macaulay Library at the Cornell Lab of Ornithology ($n = 9$).

$0.004 \pm 0.01$, $p = 0.74$; peak frequency: $0.011 \pm 0.03$, $p = 0.52$). Effect sizes for all traits were at or below the threshold of a 'small' effect (Cohen's $d = \pm 0.2$); Cohen's $d \pm 95\%$ CI was calculated across all traits (song length: $0.01 \pm 0.33$; trill rate: $-0.20 \pm 1.44$; minimum frequency: $-0.02 \pm 0.17$; maximum frequency: $-0.03 \pm 0.15$; frequency bandwidth: $-0.02 \pm 0.14$; peak frequency: $-0.02 \pm 0.10$). When the location was removed as a random effect, urban/non-urban was not a significant variable in any model (urban/non-urban: $p > 0.5$ in all models). Further analyses are presented in the electronic supplementary material, finding no significant or strong differences between populations with an individual male as the statistical unit or without location (electronic supplementary material, table S1 and text S2).

## 3.2. Vocalization analysis between UCSD 2006/2007, UCSD 2018–2020, UCLA 2018–2020 populations

In comparing the song characteristics of the 2006/2007 seasons with the 2018–2020 seasons at UCSD, we found no statistically significant differences in frequency bandwidth ($p = 0.97$, estimate $\pm 95\%$ CI = −

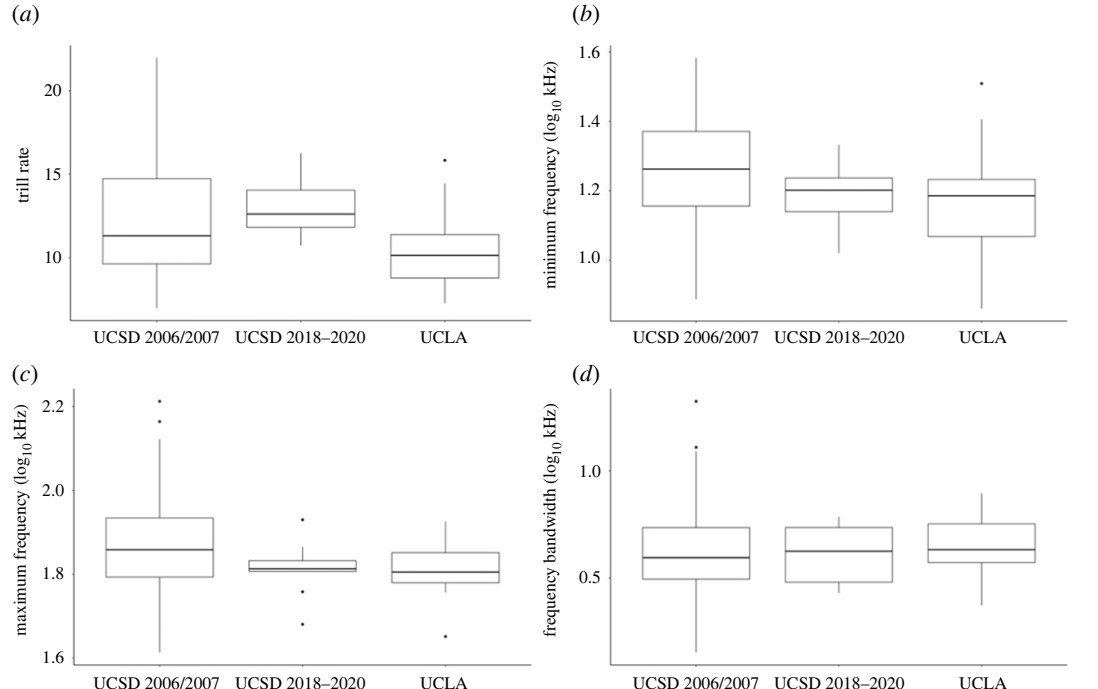

**Figure 4.** Boxplots of differences in the UCSD population during the 2006/2007 and the 2018–2020 breeding seasons and the UCLA population in (*a*) trill rate, (*b*) minimum frequency ($\log_{10}$ kHz), (*c*) maximum frequency ($\log_{10}$ kHz) and (*d*) frequency bandwidth ($\log_{10}$ kHz) per song type. For all traits except minimum frequency, there are no statistically significant differences between the songs taken from UCSD juncos during the 2006/2007 breeding seasons and those taken during the 2018–2020 breeding seasons. There are significant differences found between the minimum frequency and marginally significant differences found in the maximum frequency of UCSD 2006/2007 and UCLA. UCSD 2006/2007 = University of California San Diego 2006/2007 field season ($n = 168$); UCSD 2018–2020 = University of California San Diego 2018–2020 field season ($n = 9$); UCLA = University of California Los Angeles ($n = 23$).

$0.001 \pm 0.05$; $d \pm 95\%$ CI $= 0.01 \pm 0.67$), trill rate ($p = 0.57$; estimate $\pm 95\%$ CI $= 0.73 \pm 2.52$; $d \pm 95\%$ CI $= 0.20 \pm 0.68$), minimum frequency ($p = 0.30$; estimate $\pm 95\%$ CI $= -0.02 \pm 0.04$; $d \pm 95\%$ CI $= 0.36 \pm 0.68$), or maximum frequency ($p = 0.13$; estimate $\pm 95\%$ CI $= -0.02 \pm 0.08$; $d \pm 95\%$ CI $= 0.52 \pm 0.68$) (figure 4).

Between the UCSD 2006/2007 and UCLA populations (i.e. approx. 15–20 years after colonization for each), we found significant differences in minimum frequency ($p = 0.006$; estimate $\pm 95\%$ CI $= -0.04 \pm 0.03$; $d \pm 95\%$ CI $= 0.62 \pm 0.44$) and marginally significant differences in maximum frequency ($p = 0.017$; estimate $\pm 95\%$ CI $= -0.025 \pm 0.02$; $d \pm 95\%$ CI $= 0.54 \pm 0.44$), but not frequency bandwidth ($p = 0.39$; estimate $\pm 95\%$ CI $= 0.015 \pm 0.03$; $d \pm 95\%$ CI $= -0.19 \pm 0.44$) and not trill rate ($p = 0.45$; estimate $\pm 95\%$ CI $= -0.63 \pm 1.63$; $d \pm 95\%$ CI $= -0.17 \pm 0.44$). UCSD 2006/2007 had higher minimum frequency and maximum frequency in comparison to UCLA 2018–2020 (figure 4).

Similar results were found in alternative analyses using an LM and song type with no random effect and a GLMM to conduct pairwise comparisons on all locations, as presented in electronic supplementary material, text S3 and S4.

## 3.3. Power analysis

When comparing urban and non-urban juncos, power for the Bayesian mixed-effect models varied across response variables for the effect sizes found in our models. For minimum frequency, power (95% CI) was 4.00% (1.10%, 9.93%). For maximum frequency, power was 3.00% (0.62%, 8.52%). For frequency bandwidth, power was 6.00% (2.23%, 12.6%). For peak frequency as the response variable, power was 10.0% (4.90, 17.6%). Power for song length was 3.00% (0.62%, 8.52%). Power for trill rate was 4.00% (1.10%, 9.93%). In other words, if the differences we found between populations were 'true', we would have low power at detecting them because their effect size is so low. On the other hand, our models exhibited high power (95% CI) in determining medium effect sizes (Cohen's $d = 0.5$) for most traits (minimum frequency: 100% (96.4, 100%); maximum frequency: 100% (96.4, 100%); frequency bandwidth: 100% (96.4, 100%); peak frequency: 100% (96.4, 100%); song length: 97.0% (91.5%, 99.4%))

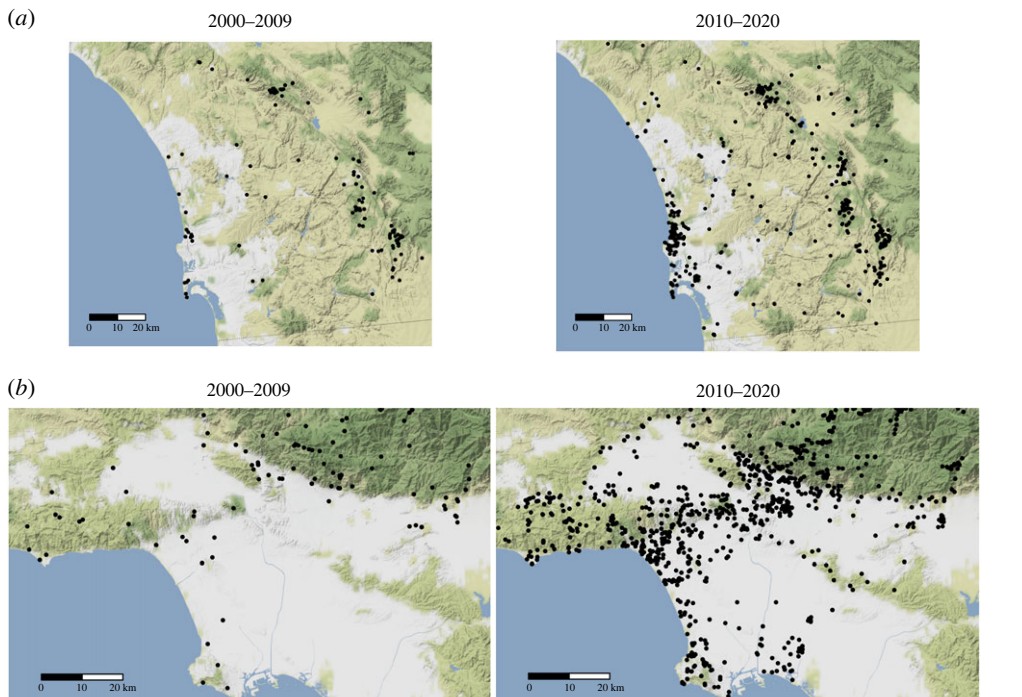

**Figure 5.** The expansion of dark-eyed juncos from April to July of 2000–2020 at 10-year intervals in the (*a*) San Diego area and (*b*) Los Angeles area with community science data collected by and accessed through *eBird* [64]. These data were curated by *eBird* to only include reliable observations. Black circles represent locations where dark-eyed juncos have been observed.

except for trill length (power (95% CI) = 11.00% (5.6%, 18.8%)). As such, we did not simulate power for high effect sizes (Cohen's *d* = 0.8). This suggests that we had relatively high power to detect moderate effect sizes across all traits except for trill rate. For reference, power for predicting a 500 Hz shift in minimum frequency is 87% (95% CI = 78.8, 92.9%).

When comparing UCSD 2006–2007 to UCLA and to UCSD 2018–2020, our models had various power levels for the effect sizes we found. Our LM with trill rate as a response variable yielded a power (95% CI) of 12.0% (6.36%, 20.0%). With minimum frequency as a response variable, power was 74.0% (64.3%, 82.3%). With maximum frequency as a response variable, power was 68.0% (57.9%, 77.0%). For frequency bandwidth as a response variable, power was 11.0% (5.62%, 18.8%).

## 3.4. Visualizing urban population expansion

In combination with previous knowledge of a lack of breeding juncos in Los Angeles until the early 2000s, we found that breeding bird populations have expanded in recent years in urban San Diego and urban Los Angeles (figure 5).

# 4. Discussion

Our results did not find significant differences between junco songs from urban and non-urban environments in Southern California in the traits of song length, trill rate, minimum frequency, maximum frequency, peak frequency and frequency bandwidth across sites and between urban and non-urban groupings. These null results could be driven by low sample size in non-urban environments and contemporary data from UCSB, UCSD and Occidental College, especially since dark-eyed junco song is highly variable due to a large amount of improvisation that occurs during song development. This results in very little song sharing between males and their neighbours. Such high variation poses a risk to having such small sample sizes because it increases the likelihood that outliers exist within the data, thereby skewing the results. However, power analyses and effect sizes suggest that if there are differences between urban and non-urban juncos, they are of small effect. We noted significant differences between UCLA and UCSD 2006/2007 song traits often associated with urban noise—minimum and maximum frequency—at a similar time-since-colonization. These

populations were recorded approximately 20–25 years since colonization for UCSD and approximately 15–20 years since colonization for UCLA. This suggests that while San Diego juncos seemed to shift their songs upon colonizing the city in response to noise, Los Angeles juncos may have not done so to the same degree, assuming that non-urban juncos from local mountains independently colonized urban Los Angeles. This assumption has been made since urban juncos are largely sedentary and likely from migratory birds becoming sedentary in wintering grounds. San Diego juncos further began as an island population at UCSD, which is indicative of low dispersal distance. This finding is despite the fact that they generally experience similar levels of ambient noise, though our study was limited by not directly measuring noise profile. Noise profiles would allow us to measure noise-induced plasticity. Nonetheless, this is a somewhat surprising result given that the physical composition and levels of ambient noise pollution of the urban environments and their nearby non-urban mountain sites seem to be largely different [20,46]. Not only that, but a number of previous studies have already found significant differences in junco song between one urban and multiple non-urban settings in San Diego [20,49,70,72]. We discuss several possible reasons why we would not see any strong contemporary differences in urban and non-urban junco songs and why the Los Angeles population might not have adjusted its song to the degree that San Diego had, including song being a by-product of other urbanization adaptations, individual plasticity, counteracting transmission demands and a lack of time to evolve.

First, while much of song research focuses on the evolution aspect of song itself, it is possible that these song differences come as a by-product of the evolution of other adaptations to urban environments. For example, song characteristic differences may not always be directly a result of selection due to noise pollution [42]; they can also be the result of morphological changes that also affect song production and frequency [5,29,101–104]. Urbanization can cause changes in morphology (e.g. selecting for changes in beak and body sizes) as fluctuations in food abundance and type often characterize the urban environment [103,105,106]. There is, however, evidence that differences in body size were not correlated with song frequencies in juncos [107], and this lack of correlation was also found in 529 and 842 urban and non-urban populations of songbird species, respectively [108,109].

Second, while there may be individual phenotypic plasticity in adjusting song characteristics and selecting particular song types that allow for better transmission in a particularly noisy environment [110,111], we did not find evidence of song shift. A number of studies have shown that in response to increased and changing ambient noise levels in an urban landscape, urban songbirds can shift their song frequencies with short-term, immediate flexibility (in a time span of minutes) [112–118]. On the other hand, work done on vermilion flycatchers (*Pyrocephalus obscurus)* found a lack of short-term, immediate flexibility in song minimum frequency [119]. Vermilion flycatchers are sub-oscines, while juncos are oscines that learn their songs socially or through improvisation [65,69,120]. This could be a factor as birds that learn their song may also be better at adjusting their song based on environmental factors than those that do not learn their song. While phenotypic plasticity might affect junco song change, Reichard *et al*. [72] found evidence suggesting evolutionary change and/or early parental effects underlie song shifts in UCSD juncos, potentially explaining a lack of strong frequency shifts in other urban environments.

Third, while it is hypothesized that longer songs of higher frequencies and reduced trill rates allow for better signal transmission in urban environments due to the echoes and reverberations caused by the tall buildings, open spaces and intense low-frequency ambient noise [4,16,28], this may be counteracted by differing habitat composition and noise levels within different parts of both urban and non-urban areas. While we did not quantify this, each urban and non-urban location studied had noticeably distinct environmental characteristics in different areas of the study site. For example, UCSD has an urban forest on campus, making that area less urbanized compared to other parts of the campus. These kinds of urban forests, parks and gardens may attenuate sound and reverberations similarly to non-urban forests [40] and may affect a bird's behaviour differently than if the area were entirely urban [121]. In addition, depending on the time of day, the amount of noise pollution fluctuates in cities depending on people's routines and schedules. Future studies on the junco populations in these urban sites should consider this urbanization gradient, as this may affect song parameters [38].

Fourth, if significant frequency shifts in songs occur, they may be due to population-wide changes that occur over longer evolutionary timescales. Zollinger *et al*. [42] found that increased song frequency shifts were not explained by developmental plasticity or chronic noise induced individual plasticity in great tits, suggesting that any observed changes in the song of urban and non-urban birds may be the result of slower evolutionary processes rather than immediate plastic responses. If this is the case, then perhaps not enough time has passed since the establishment of the newer urban populations studied here to detect differences in song traits. Because the UCSD population was established in 1980 to early 2000s,

enough time has likely passed to allow for distinct differences to appear between the UCSD and native-habitat populations. Meanwhile, the UCLA population was likely established in the early or mid-2000s—the first reported junco during the breeding season at UCLA on *eBird* [64] was in April 2007, and juncos have been seen breeding since at least 2008 (P. Yeh 2008, personal observation). The Occidental College and UCSB populations were first sighted in April and May 2013, respectively [64]. The UCSD population has been established for approximately 40 years. A recent study found that juvenile juncos from UCSD raised in captivity maintained the significantly higher minimum frequency seen previously in the UCSD population [72], suggesting that these frequency shift differences in song can be heritable. Thus, it is possible that while evolutionary changes in some song traits could occur relatively rapidly, some junco populations may not have been established long enough to show significant song frequency shifts or may be limited by the song traits of the initial colonizing population. Juncos have established populations in the urban areas studied (Los Angeles and Santa Barbara) within the last 10–20 years, making their expansion into these areas a relatively recent development.

Approximately two decades have passed since the juncos studied in the Slabbekoorn *et al.* [16] and Newman *et al.* [49] papers were recorded (1998 and 2001; 2002 and 2003, respectively); therefore, there may have been significant changes to the population since then that could affect differences in their song. At that time, UCSD was an island population that experienced founder effects [57]; however, since the mid-2000s, their breeding population has expanded beyond their prior borders (figure 5) [64]. UCSD may no longer be an isolated population. This increased gene flow could counteract founder effects. Indeed, urban populations throughout Southern California are rapidly expanding (figure 5). While we know the first reported sightings on *eBird* of juncos in all the locations studied during the breeding season of both urban and non-urban birds (April–July) [64], we do not know the actual first instances of juncos in these urban areas, and we do not know the sources of these populations nor the extent of gene flow between our urban and natural populations. This could be a possible explanation of why we detected differences in junco song between UCSD 2006/2007 and UCLA but not between contemporary UCSD and UCLA populations. Juncos do, however, learn their songs from what they hear around them and can improvise novel song types during development [69,120]. Juncos can also undergo rapid divergence in song and behaviour in short timescales [57]. In spite of this, we still do not see much divergence in song characteristics between the urban and non-urban sites despite likely strong differences in ambient noise levels, which we did not directly measure in this study. The significant differences between UCSD 2006/2007 and UCLA 2018–2020 suggest that there could be geographical variation and potential heterogeneity over time in response to urban noise. Longer, faster and higher frequency songs could also be associated with bird population densities [122], which tend to be higher in urban environments [105,123] and may differ over time and between populations. Perhaps differences in the behavioural ecology of the birds between the two cities might lead to differences in song characteristics.

Our findings are only partially in line with previous research studying similar characteristics in junco songs in Southern California. Our data support the finding of Slabbekoorn *et al.* [16] of no statistically significant difference in song length and trill rate, but we did not find an increased minimum frequency between urban populations compared to non-urban populations. On the other hand, our results do support the finding of Newman *et al.* [49] of no statistically significant difference between trill rate and minimum frequency, but not their finding of an increased maximum frequency and reduced song length ($p < 0.1$ alone, $p < 0.05$ combined with Slabbekoorn *et al.* [16]). Given that differences in minimum frequency were revealed for the San Diego population in a common garden experiment [72] and in studies with larger population sizes [46,65], we suspect our very low sample sizes in some of these populations could affect our results that contrast locations other than UCLA and UCSD 2006/2007. As UCSD 2006/2007 data were extracted by other observers, differences could be partially subject to individual differences in measuring methods. In all other traits, there were no significant differences found. There is also extensive overlap in these traits between the sampling years (figure 4). There was a large difference between the sample sizes compared from the UCSD 2006/2007 (101 males) and UCSD 2017–2020 (17 males), and further data collection could possibly yield different or additional insights.

# 5. Conclusion

We did not find strong differences contemporarily between multiple urban and non-urban sites. However, we found that the song characteristics of two urban populations—Los Angeles and San

Diego—were not similar to each other approximately 20 years after colonization in each city (San Diego 2006–2007 and UCLA 2018–2020). This was a surprising finding given these two cities' similar ambient noise levels. Future studies could expand on this work by focusing on noise-induced plasticity by measuring noise profiles and the possibility of further population-level shifts over time in multiple populations. Investigating how and why juncos and other species adjust their song across multiple urban and non-urban populations would allow us to start understanding how generalizable and predictable species' song responses are to urbanization.

Ethics. All procedures were approved by the UCLA IACUC Ethics Committee and in accordance with IACUC protocol no. ARC-2018-007. Banding was conducted under the United States Geographic Survey Master Banding Permit no. 23809 and California Department of Fish and Wildlife Specific Collection Permit S-183040004-18313-001.

Data accessibility. The data collected and analysed for this study are available in Dryad: https://datadryad.org/stash/share/1S4OxcpO85xeWnD0EeQY1haBBjmkgIj5xQ7BwxUBoS4 [124].

Authors' contributions. F.W.: conceptualization, data curation, formal analysis, funding acquisition, investigation, methodology, project administration, supervision, visualization, writing—original draft, writing—review and editing; E.S.D.: conceptualization, data curation, formal analysis, investigation, methodology, project administration, supervision, visualization, writing—original draft, writing—review and editing; M.W.: data curation, formal analysis, methodology, writing—original draft, writing—review and editing; P.J.Y.: conceptualization, methodology, project administration, supervision, validation, visualization, writing—original draft, writing—review and editing.

All authors gave final approval for publication and agreed to be held accountable for the work performed therein.

Conflict of interest declaration. The authors declare they have no non-financial interests.

Funding. We obtained funding for this research from the UC Natural Reserve System, the Pasadena Audubon Society, the Santa Monica Bay Audubon Society and the National Geographic Society.

Acknowledgements. We thank Alex Wong, Jonathan Tanigaki, Katie Huang and Connie Kim for assistance in the field. We thank Daniel Blumstein, Peter Nonacs and the anonymous reviewers for comments on the manuscript. We would also like to acknowledge the receipt and usage of media from the Macaulay Library at the Cornell Lab of Ornithology.

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
