## [Peer Review File · Royal Society Open Science]

Review History

RSOS-211063.R0 (Original submission)

Review form: Reviewer 1

Is the manuscript scientifically sound in its present form?

No

Are the interpretations and conclusions justified by the results?

No

Is the language acceptable?

Yes

Do you have any ethical concerns with this paper?

No

Have you any concerns about statistical analyses in this paper?

No

Recommendation?

Major revision is needed (please make suggestions in comments)

Comments to the Author(s)

The study addresses a timely topic that will be of interest to a broad readership. Moreover, the publication of negative findings is crucial for the further development of the field and thus I feel this study is potentially a valuable addition to the literature. However, there are also some problems with the work.

A major concern is that the acoustic measurements were taken by hand from spectrograms (c.f. Zollinger et al. 2012 *Anim Behav* 84:e1-e9). Several studies have pointed out the manifold problems of this practice, such as measurement artefacts and the very low repeatability between and within observers (Zollinger et al. 2012 *Anim Behav* 84:e1-e9, Grace & Anderson 2015 *Behav Ecol Sociobiol* 69:253-263, Rios-Chelen et al. 2016 *Behaviour* 67-145-152, Brumm et al. 2017 *Methods Ecol Evol* 8:1617-1625). However, the deficient song data in this study is no deal breaker: the problem can be remedied by re-measuring the song recordings with proper acoustic methods (e.g., measuring durations in oscillograms and frequencies at set amplitude thresholds in power spectra, see Zollinger et al. 2012 *Anim Behav* 84:e1-e9).

Another concern is related to the lack of noise measurements. Because the main argument for an urban-rural song divergence is adaptation to anthropogenic noise, similar published studies usually measure environmental noise levels at the recording sites. This study used the distance to closest road as a proxy for noise but I'm not convinced that this is valid because traffic noise is not only determined by the distance to the road but also by the amount of traffic, vehicle speed, the types vehicles, road surface etc. Moreover, the urban sites in this study were all college campuses which were probably not particularly noisy. Anyway, without noise measurements this is impossible to tell.

Review form: Reviewer 2

Is the manuscript scientifically sound in its present form?

No

Are the interpretations and conclusions justified by the results?

No

Is the language acceptable?

Yes

Do you have any ethical concerns with this paper?

No

Have you any concerns about statistical analyses in this paper?

No

Recommendation?

Major revision is needed (please make suggestions in comments)

Comments to the Author(s)

This article compares acoustic properties of dark-eyed junco song between different urban and non-urban sites in California. Its goal is to test how common or general are urban differences in

song. Not having found differences in song between urban and non-urban sites, unlike previous work with this species, the discussion then considers a number of possibilities to explain the results.

The article is mostly well written, but the discussion is somewhat confusing and could be streamlined and better organized. For example, some of the possibilities equated in the discussion are weak and do not apply well to this species, while others are interesting and even highly novel, but the lengths to which they are explained do not reflect their differences in quality or applicability to the species.

I worked on the same topic and study system (G. Cardoso), and would be pleased seeing new studies developing this further. But I have concerns about the article, the first of which may invalidate its main finding. The main concerns are:

- 1- an extremely low sample size, especially in non-urban sites.
- 2- methods for acoustic measurements with faults and not well-argued.

1- The sample size (number of males) is extremely low for all non-urban populations, and for some urban populations as well, which is troublesome since within-population variation in dark-eyed junco song traits is very wide (reviewed in Cardoso & Reichard 2016). Even when joining populations together and adding recordings from the Macaulay Library, non-urban sample size is still low, and there are complications due to lumping populations (please see comment on line 242). Therefore, it is likely that the dataset affords very low statistical power to test whether song differs between urban and non-urban populations.

If the article is to retain its current focus, power analyses would need to be presented. For power analyses, the within-group variation can be estimated from the current data and, for the expected difference, authors can use results of past studies. For example, urban vs. non-urban differences in minimum song frequency are usually smaller than 100Hz for most species, but in the dark-eyed junco large differences of up to 500Hz were reported (Cardoso & Atwell 2011 Evolution, Reichard et al. 2020 Anim Behav).

- Only if power analyses show good power to detect population differences of, say, 50 or 100 Hz, are the current statements in the title, abstract and conclusions warranted.
- If power analyses show good power to detect large differences of 400 or 500 Hz, but not differences of 50 or 100Hz, then it can be concluded that such large differences are not commonplace, but it cannot be concluded that there are no consistent urban vs. non-urban differences.
- If power analyses show weak power even for large population differences, then results are inconclusive. If this is the case, it may be useful to consider re-orienting the article to questions that can be addressed with data from the better-sampled urban populations (e.g., see comment on line 367).

2- Although standard some years ago, there are aspects of the acoustic measurement methods that are now substandard. One is making frequency measurement with the cursor on the spectrogram, because this is not the most objective method and measurements could be influenced by background noise (Zollinger et al. 2012 Anim Behav, Brumm et al 2017 Methods Ecol Evol). I think that frequency measurements from the spectrogram can be used if other more automated methods are not feasible (e.g., the level of noise in many recordings prevents automated measurement) and if it is argued that care was taken (e.g., adjusting the dynamic range of spectrograms to distinguishing traces of song amidst noise; Cardoso & Atwell 2012 Anim Behav). Such a choice, however, needs to be explained.

Another issue is the use of a simple linear Hz scale to measure sound frequency and to compute frequency bandwidth. Since animals perceive and modulate sound frequency on a ratio (log) scale, measuring frequency with a linear Hz scale distorts the measurement of bandwidth, and overestimates differences in maximum frequency relative to differences in minimum

frequency (Cardoso 2013 Anim Behav). Reichard et al. (2020 Anim Behav) and Winandy et al. (2021, Front Ecol Evol 9:570420) are recent examples of how to do this, so as to evaluate differences in minimum and in maximum frequency in a comparable scale.

Other comments, by line number.

75-77 - This is clear for suboscines, for example, but less generally applicable in the case of oscine song, including junco song (Cardoso and Atwell 2011 Anim Behav and references therein). Since the manuscript deals with junco song, this point should be made with restraint.

83-84 - Not clear; please clarify what is meant here by "geographic distance between these two studies".

88-90 - This is a strange statement. The junco is one of the species with more work and discussion on urban and geographic song divergence (see also Cardoso & Reichard 2016 for discussion on general, non-urban geographic divergence). I realize that existing work deals with a particular urban population, and that this sentence does include the word "multiple"; still, it sounds strange. Instead, I would expect that the introduction summarizes what is known about urban song divergence in juncos, to then frame the objectives of the current work.

96-97 - Unclear; please check phrasing.

97-99 - Also unclear for the general reader, and the following statements do not clarify what is meant here by "song and in song type use within males but not across males". Also, it was not explained before that only males sing.

144 - More information is needed here. For example, what was the average number of songs per male, and was this similar across populations?

186-188 - This requires explanation for general readers to understand. In any case, it seems too soon to bring up this point, because the units for statistical analysis (songs, males, song types?) were not explained yet.

192 - Not clear. Please state that that recordings were normalized for (peak?) amplitude.

193, 198-199 - The method is poorly described. Mentioning "from the spectrogram" and a "selection box" is insufficient for general readers to understand. Please state how you chose the upper and lower limits of the "selection box", and explain that these correspond to your measured maximum and minimum frequencies. Also, please see main comment 2 about issues with using this method.

195 - I would suggest not writing "following Cardoso et al.", because there are differences in methods. One difference is that this past work measured frequency per syllable rather than per song, which has implications for the precision of measurements. Another is that, as discussed in Cardoso & Atwell (2012 Anim Behav), those measurements involved adjusting the dynamic range of spectrograms to optimize distinguishing traces of song amidst noise. Instead of writing "following ...", I would suggest describing in more detail the measurement method that was used.

198-199 - I was wondering whether peak frequency should have been measured and analyzed. Unlike minimum and maximum frequencies (comment 2), peak frequency is easy to measure objectively, and analyzing it would make the conclusions of the article more robust.

200-201 - Please see main comment 2 about issues with computing bandwidth in this manner.

201-203 - This is unclear, especially since earlier (lines 186-188) the manuscript alluded to another source of pseudoreplication (using males rather than song types as statistical units). Using males as statistical units is fine for the purpose of this paper, but this needs to be made clearer, and this averaging procedure needs to be better described and argued. For an individual male, were all its recorded songs averaged irrespective of their song type (in this case, the mean value reflects song repertoire usage), or were songs of the same type averaged and then those values averaged across song types (in this case, it reflects the mean of the male's song repertoire)?

213 - "when available" creates cloudiness in this description of models. Please consider re-organizing so as to explain for which models these covariates were used or not.

232 - The concerns in main comment 1 also apply to these LDAs.

240 - The majority of P values in Supplementary Table S1 are =1 or >0.9. These are extremely high, and I would think that they are excessively prevalent even if we were analyzing random data. Is there some type of correction for multiple comparisons that I missed and that pushes up the P values? If so, that should be explained. If not, then perhaps these prevalent $P > 0.9$ simply reflect insufficient sample size and low statistical power for the tests, which should be commented on.

242-247 - If there is geographic variation in song, then lumping all these populations in only two groups will increase within-group variation and make this test conservative. Perhaps geographic variation is not expected, and the previous analyses (lines 237, 240) did not find population differences in song, but, with such small sample sizes, that is not a strong argument for this lumping. This point should be discussed.

252 (also lines 258 onwards) - These data suggest ($P = 0.06$) that the minimum sound frequency at UCSD decreased by 250 Hz since 2006/7 to 2018/20. What may have happened? Are there noticeable differences in the UCSD environment, traffic or otherwise? Or, as discussed later on (line 367 onwards) this is no longer an island-population, but now immigration and emigration are more common? Perhaps this result should be more highlighted in the discussion (even if only suggestive) to try to make sense of the different results between this and past studies. Also, were the song recordings from 2006/7 re-measured for this study, or were the previous measurements re-used? In the latter case, could this result indicate differences in measuring method (since the method that was used is prone to some subjectivity)?

264-266 - It is very unclear what these "similar findings" refer to. The purpose of, or need for, the LDA analyses do not seem to have been well argued. Are they needed?

272-275 - Confusing sentence: it starts referring to differences with UCSD in 2006/7, but finishes commenting on the result in the previous sentence, and not this one. Please separate and develop the ideas in different sentences.

282-298 - This is theoretically possible on other species, but hardly so for juncos; see Cardoso et al. 2009 and Reichard et al. 2020 for evidence that sound frequency is extremely stable within song type and individual. In this regard, I did not understand the rationale in the last sentence, referring to Reichard et al. 2020.

Since this discussion is about junco song, the argument developed here does not appear supported. Also, I did not understand how plasticity would help explain your results, and their difference relative to past work on juncos, without changes in the environment or ecology? (please see comment on line 252).

299-308 – I agree partially with the argument here. A weak point of the song sharing argument is that juncos share very few songs with neighbors (Newman et al. 2008; see also discussion in Cardoso & Reichard 2016). The argument about junco repertoires harboring a large variation in frequency is valid, but again it does not explain differences in relation to past work. Instead, the fact that individual and population repertoires are acoustically diverse raises the issue of whether sample sizes were adequate (please see main comment 1).

309-318 – This argument appears to potentially undermine the design of the study. The choice of urban and non-urban sites appeared clear-cut when reading the first paragraph of the methods, but this paragraph puts it into question. Please clarify whether or not there were clear differences in anthropogenic noise levels between the sites classified as urban and non-urban. Namely, what were the habitats and the exposure to traffic in the non-urban sites, and how does that compare with habitat and exposure to traffic in the non-urban sites used in previous work (Slabbekoorn et al. 2007 Condor, Cardoso & Atwell 2011 Evolution)?

318 – Figure 2 was not mentioned in the results. This figure has problems and lacks explanations: the colors are difficult to distinguish and do not follow a logical order, the acronyms in the color legend were not defined and do not appear ordered logically, there appear to be two UCSD bars but this was not explained, boxplots are not well described (vertical lines are not the full range of data); font size and overall appearance need re-touching for clarity.

319-334 – Similar comment to that in lines 309-318. Also, do noise levels differ among the different urban sites? Lines 319-321 appear to belong to this paragraph, rather than the previous..

335-345 – This too is a reason for not lumping all urban populations in a single group. I think that this information should have appeared in the introduction, and that it should have been used to shape the hypothesis-testing goals and analyses.

355-367 – I think I am not partial saying that this paragraph does not adequately interpret past work on the UCSD population. While Slabbekoorn et al 2007 and Newman et al 2008 compared urban songs with a small number of mountain males (16 or less per mountain population), and differences in minimum frequency were sometimes statistically significant and sometimes not, when we studied a larger sample (101 UCSD and 50 LM males) this difference in frequency was very clear (Cardoso & Atwell 2011 Evolution, 2016 R Soc Open Sci).

362-363 – This is the most serious of issues, as it may invalidate the message in the abstract, title of the paper, and concluding paragraph. This statement, thus, does not suffice. Please see main comment 1.

367-375 – This is the good stuff! A novel and original idea that, unlike the previous paragraphs, might explain the different results in this and past studies. This update on the natural history of the UCSD and other urban populations, together with the different results in this paper vs. past work on the same system, plus the change in minimum frequency in UCSD over the past 10 years, can all combine to support a very interesting case of strong acoustic adaptation to the urban environment when the population is a sedentary island population, and subsequent erosion of that acoustic adaptation when philopatry is not strong anymore. This should definitely be in the introduction and in the results, and be used to shape / focus the hypotheses and analyses, rather than only briefly appearing here.

As is, focused on testing urban vs. non-urban differences in song, the article may be weak and not able to support its conclusions, because the non-urban samples are so small (please see the main comment 1). But the dataset for some urban populations, including UCSD at two different time

points, is better, and the article could be re-oriented to focus on comparisons among the well-sampled urban populations.

Figure 4 – These are very interesting new data, and their implications potentially important. I think that these data merit a more careful presentation, for example distinguishing observations during winter (these could be mountain birds in the altitudinal migration) and during breeding and post-breeding season (these would indicate that the resident population had indeed increased). I would suggest a methods section and a results section for these new data. It is difficult to distinguish small and large pins in such small panels, and I did not understand the difference between blue and red pins (there are many blue pins in places without sightings in the previous time interval...). Also, the maps do not have a scale, and it would be useful to place rectangles in Figure 1 indicating the areas of these maps.

380-381 – But differences among sites are not expected, given the mode of development of junco song, and the pattern geographic variation in song (see figure 13.2 in Cardoso & Reichard 2016).

382-395 – Perhaps the really biologically-meaningful difference in the results is not the comparison between UCSD in 2006/7 and UCLA in 2018, but the difference between UCSD in 2006/7 and 2018, with the difference with UCLA being just a consequence of that? Please see comment on lines 367-375. I realize that the P value for the longitudinal comparison at UCSD was marginal (0.06) and that a larger sample size of UCSD recordings might be useful. If useful, although at the time we did not publish our data, we would be happy to share it.

396-408 – This is another paragraph equating an explanation for which there is contrary evidence in juncos (see also comment on lines 282, 299). I agree that these points should be brought up. But, rather than equating these possibilities in long paragraphs to then go back and deny their applicability in juncos, I suggest that they could be stated more briefly and, from the onset, in the context of explaining why they are unlikely explanations in the junco.

409-413 – It is not clear how this relates to the interpretation of the results.

Decision letter (RSOS-211063.R0)

Dear Ms Wong,

The Editors assigned to your paper RSOS-211063 "No evidence of song divergence across multiple urban and non-urban populations of dark-eyed juncos (*Junco hyemalis*) in Southern California" have made a decision based on their reading of the paper and any comments received from reviewers.

Regrettably, in view of the reports received, the manuscript has been rejected in its current form. However, a new manuscript may be submitted which takes into consideration these comments.

We invite you to respond to the comments supplied below and prepare a resubmission of your manuscript. Below the referees' and Editors' comments (where applicable) we provide additional requirements. We provide guidance below to help you prepare your revision.

Please note that resubmitting your manuscript does not guarantee eventual acceptance, and we do not generally allow multiple rounds of revision and resubmission, so we urge you to make every effort to fully address all of the comments at this stage. If deemed necessary by the Editors, your manuscript will be sent back to one or more of the original reviewers for assessment. If the original reviewers are not available, we may invite new reviewers.

Please resubmit your revised manuscript and required files (see below) no later than 23-Jan-2022. Note: the ScholarOne system will 'lock' if resubmission is attempted on or after this deadline. If you do not think you will be able to meet this deadline, please contact the editorial office immediately.

Please note article processing charges apply to papers accepted for publication in Royal Society Open Science (<https://royalsocietypublishing.org/rsos/charges>). Charges will also apply to papers transferred to the journal from other Royal Society Publishing journals, as well as papers submitted as part of our collaboration with the Royal Society of Chemistry (<https://royalsocietypublishing.org/rsos/chemistry>). Fee waivers are available but must be requested when you submit your manuscript (<https://royalsocietypublishing.org/rsos/waivers>).

Thank you for submitting your manuscript to Royal Society Open Science and we look forward to receiving your resubmission. If you have any questions at all, please do not hesitate to get in touch.

on behalf of Dr David Wilson (Associate Editor) and Kevin Padian (Subject Editor)
openscience@royalsociety.org

Subject Editor Comments to Author:

Thanks for your submission, which all readers think is interesting and useful. They have methodological concerns, some of which may be answered relatively easily, and others of which may require additional work, if you feel this is reasonable. The reject/resub decision is in large measure to give you additional time to consider your response to these concerns. We'll look forward to your revised submission. Best wishes.

Associate Editor Comments to Author (Dr David Wilson):

The study compares the acoustic structure of junco songs recorded in urban versus rural locations, and, for one urban location, at two time periods separated by more than a decade. A strength of the paper is that it includes multiple urban and rural locations, rather than a single location of each type as many previous studies have done. The results show little or no divergence among locations and habitat types.

The reviewers and I agree that the topic is timely and important, and that the paper is generally well-written. However, we also identify several serious methodological issues that make the results and discussion difficult to assess. First, it is unclear if/how individual birds were identified/distinguished during recording (were they banded?), which calls into question the independence of the data. Second, we all agree that the acoustic measurements (mainly, taking frequency measurements directly from a spectrogram as opposed to using an amplitude-based

threshold on a power spectrum) are flawed and need to be re-done (reviewer 2 provides details and key references). Reviewer 2 notes that using a logarithmic scale on the frequency axis may be more appropriate, and I note that call rate should be calculated as the number of syllables minus one, divided by the period of time between the start of the first syllable and the start of the last. Third, no attempt is made to quantify noise levels at the different recording sites, which is necessary for assigning sites to urban or rural categories and for assessing the potential effects of noise on song structure. Although it would have been better to take such measures at the time of recording, measurements taken now could still be used to support a categorical difference in habitat types and the use of distance-to-nearest-road as a proxy for noise level. Such measurements may also show that noise levels are not categorical, but continuous, and that regression-style analyses are more appropriate. Fourth, the sample sizes of rural populations are very low and a power analysis is needed to determine whether the largely negative results are genuine or simply type II error. Finally, the number of statistical analyses is excessive, and many seem unrelated to the main hypothesis about habitat type. There are separate analyses for 6 dependent variables (5 song traits and 1 composite measure), 7 locations, urban versus rural habitats, with and without repository data, with and without populations with small sample sizes, and for 2006/07 versus 2017-2020 data. Rationale for analyzing geographic variation is not provided in the introduction, and the methods do not properly address it (for example, by relating song divergence to geographic distance among populations). I therefore recommend that analyses of geographic variation be removed from the manuscript, and that study population be included as a random effect rather than a fixed effect in all analyses. Inflation of experimentwise type I error due to multiple testing of the same hypotheses will also need to be considered.

Given the substantive methodological concerns, it currently is not possible to evaluate the results and discussion, and thus the main conclusions of the study. The underlying acoustic data seem solid (though perhaps insufficient), and it may or may not be possible to address our concerns in revision.

Reviewer comments to Author:

Reviewer: 1

Comments to the Author(s)

The study addresses a timely topic that will be of interest to a broad readership. Moreover, the publication of negative findings is crucial for the further development of the field and thus I feel this study is potentially a valuable addition to the literature. However, there are also some problems with the work.

A major concern is that the acoustic measurements were taken by hand from spectrograms (c.f. Zollinger et al. 2012 *Anim Behav* 84:e1-e9). Several studies have pointed out the manifold problems of this practice, such as measurement artefacts and the very low repeatability between and within observers (Zollinger et al. 2012 *Anim Behav* 84:e1-e9, Grace & Anderson 2015 *Behav Ecol Sociobiol* 69:253-263, Rios-Chelen et al. 2016 *Behaviour* 67-145-152, Brumm et al. 2017 *Methods Ecol Evol* 8:1617-1625). However, the deficient song data in this study is no deal breaker: the problem can be remedied by re-measuring the song recordings with proper acoustic methods (e.g., measuring durations in oscillograms and frequencies at set amplitude thresholds in power spectra, see Zollinger et al. 2012 *Anim Behav* 84:e1-e9).

Another concern is related to the lack of noise measurements. Because the main argument for an urban-rural song divergence is adaptation to anthropogenic noise, similar published studies usually measure environmental noise levels at the recording sites. This study used the distance to closest road as a proxy for noise but I'm not convinced that this is valid because traffic noise is not only determined by the distance to the road but also by the amount of traffic, vehicle speed, the types vehicles, road surface etc. Moreover, the urban sites in this study were all college

campuses which were probably not particularly noisy. Anyway, without noise measurements this is impossible to tell.

Reviewer: 2

Comments to the Author(s)

This article compares acoustic properties of dark-eyed junco song between different urban and non-urban sites in California. Its goal is to test how common or general are urban differences in song. Not having found differences in song between urban and non-urban sites, unlike previous work with this species, the discussion then considers a number of possibilities to explain the results.

The article is mostly well written, but the discussion is somewhat confusing and could be streamlined and better organized. For example, some of the possibilities equated in the discussion are weak and do not apply well to this species, while others are interesting and even highly novel, but the lengths to which they are explained do not reflect their differences in quality or applicability to the species.

I worked on the same topic and study system (G. Cardoso), and would be pleased seeing new studies developing this further. But I have concerns about the article, the first of which may invalidate its main finding. The main concerns are:

- 1- an extremely low sample size, especially in non-urban sites.
- 2- methods for acoustic measurements with faults and not well-argued.

1- The sample size (number of males) is extremely low for all non-urban populations, and for some urban populations as well, which is troublesome since within-population variation in dark-eyed junco song traits is very wide (reviewed in Cardoso & Reichard 2016). Even when joining populations together and adding recordings from the Macaulay Library, non-urban sample size is still low, and there are complications due to lumping populations (please see comment on line 242). Therefore, it is likely that the dataset affords very low statistical power to test whether song differs between urban and non-urban populations.

If the article is to retain its current focus, power analyses would need to be presented. For power analyses, the within-group variation can be estimated from the current data and, for the expected difference, authors can use results of past studies. For example, urban vs. non-urban differences in minimum song frequency are usually smaller than 100Hz for most species, but in the dark-eyed junco large differences of up to 500Hz were reported (Cardoso & Atwell 2011 *Evolution*, Reichard et al. 2020 *Anim Behav*).

- Only if power analyses show good power to detect population differences of, say, 50 or 100 Hz, are the current statements in the title, abstract and conclusions warranted.

- If power analyses show good power to detect large differences of 400 or 500 Hz, but not differences of 50 or 100Hz, then it can be concluded that such large differences are not commonplace, but it cannot be concluded that there are no consistent urban vs. non-urban differences.

- If power analyses show weak power even for large population differences, then results are inconclusive. If this is the case, it may be useful to consider re-orienting the article to questions that can be addressed with data from the better-sampled urban populations (e.g., see comment on line 367).

2- Although standard some years ago, there are aspects of the acoustic measurement methods that are now substandard. One is making frequency measurement with the cursor on the spectrogram, because this is not the most objective method and measurements could be influenced by background noise (Zollinger et al. 2012 *Anim Behav*, Brumm et al 2017 *Methods Ecol Evol*). I think that frequency measurements from the spectrogram can be used if other more automated methods are not feasible (e.g., the level of noise in many recordings prevents automated

measurement) and if it is argued that care was taken (e.g., adjusting the dynamic range of spectrograms to distinguishing traces of song amidst noise; Cardoso & Atwell 2012 Anim Behav). Such a choice, however, needs to be explained.

Another issue is the use of a simple linear Hz scale to measure sound frequency and to compute frequency bandwidth. Since animals perceive and modulate sound frequency on a ratio (log) scale, measuring frequency with a linear Hz scale distorts the measurement of bandwidth, and overestimates differences in maximum frequency relative to differences in minimum frequency (Cardoso 2013 Anim Behav). Reichard et al. (2020 Anim Behav) and Winandy et al. (2021, Front Ecol Evol 9:570420) are recent examples of how to do this, so as to evaluate differences in minimum and in maximum frequency in a comparable scale.

Other comments, by line number.

75-77 - This is clear for suboscines, for example, but less generally applicable in the case of oscine song, including junco song (Cardoso and Atwell 2011 Anim Behav and references therein). Since the manuscript deals with junco song, this point should be made with restraint.

83-84 - Not clear; please clarify what is meant here by "geographic distance between these two studies".

88-90 - This is a strange statement. The junco is one of the species with more work and discussion on urban and geographic song divergence (see also Cardoso & Reichard 2016 for discussion on general, non-urban geographic divergence). I realize that existing work deals with a particular urban population, and that this sentence does include the word "multiple"; still, it sounds strange. Instead, I would expect that the introduction summarizes what is known about urban song divergence in juncos, to then frame the objectives of the current work.

96-97 - Unclear; please check phrasing.

97-99 - Also unclear for the general reader, and the following statements do not clarify what is meant here by "song and in song type use within males but not across males". Also, it was not explained before that only males sing.

144 - More information is needed here. For example, what was the average number of songs per male, and was this similar across populations?

186-188 - This requires explanation for general readers to understand. In any case, it seems too soon to bring up this point, because the units for statistical analysis (songs, males, song types?) were not explained yet.

192 - Not clear. Please state that that recordings were normalized for (peak?) amplitude.

193, 198-199 - The method is poorly described. Mentioning "from the spectrogram" and a "selection box" is insufficient for general readers to understand. Please state how you chose the upper and lower limits of the "selection box", and explain that these correspond to your measured maximum and minimum frequencies. Also, please see main comment 2 about issues with using this method.

195 - I would suggest not writing "following Cardoso et al.", because there are differences in methods. One difference is that this past work measured frequency per syllable rather than per song, which has implications for the precision of measurements. Another is that, as discussed in Cardoso & Atwell (2012 Anim Behav), those measurements involved adjusting the dynamic range of spectrograms to optimize distinguishing traces of song amidst noise. Instead of writing

“following ...”, I would suggest describing in more detail the measurement method that was used.

198-199 – I was wondering whether peak frequency should have been measured and analyzed. Unlike minimum and maximum frequencies (comment 2), peak frequency is easy to measure objectively, and analyzing it would make the conclusions of the article more robust.

200-201 - Please see main comment 2 about issues with computing bandwidth in this manner.

201-203 – This is unclear, especially since earlier (lines 186-188) the manuscript alluded to another source of pseudoreplication (using males rather than song types as statistical units). Using males as statistical units is fine for the purpose of this paper, but this needs to be made clearer, and this averaging procedure needs to be better described and argued. For an individual male, were all its recorded songs averaged irrespective of their song type (in this case, the mean value reflects song repertoire usage), or were songs of the same type averaged and then those values averaged across song types (in this case, it reflects the mean of the male’s song repertoire)?

213 – “when available” creates cloudiness in this description of models. Please consider re-organizing so as to explain for which models these covariates were used or not.

232 – The concerns in main comment 1 also apply to these LDAs.

240 – The majority of P values in Supplementary Table S1 are =1 or >0.9. These are extremely high, and I would think that they are excessively prevalent even if we were analyzing random data. Is there some type of correction for multiple comparisons that I missed and that pushes up the P values? If so, that should be explained. If not, then perhaps these prevalent $P > 0.9$ simply reflect insufficient sample size and low statistical power for the tests, which should be commented too.

242-247 – If there is geographic variation in song, then lumping all these populations in only two groups will increase within-group variation and make this test conservative. Perhaps geographic variation is not expected, and the previous analyses (lines 237, 240) did not find population differences in song, but, with such small sample sizes, that is not a strong argument for this lumping. This point should be discussed.

252 (also lines 258 onwards) – These data suggest ($P=0.06$) that the minimum sound frequency at UCSD decreased by 250 Hz since 2006/7 to 2018/20. What may have happened? Are there noticeable differences in the UCSD environment, traffic or otherwise? Or, as discussed later on (line 367 onwards) this is no longer an island-population, but now immigration and emigration are more common? Perhaps this result should be more highlighted in the discussion (even if only suggestive) to try to make sense of the different results between this and past studies. Also, were the song recordings from 2006/7 re-measured for this study, or were the previous measurements re-used? In the latter case, could this result indicate differences in measuring method (since the method that was used is prone to some subjectivity)?

264-266 – It is very unclear what these “similar findings” refer to. The purpose of, or need for, the LDA analyses do not seem to have been well argued. Are they needed?

272-275 – Confusing sentence: it starts referring to differences with UCSD in 2006/7, but finishes commenting the result in the previous sentence, and not this one. Please separate and develop the ideas in different sentences.

282-298 – This is theoretically possible on other species, but hardly so for juncos; see Cardoso et al. 2009 and Reichard et al. 2020 for evidence that sound frequency is extremely stable within song type and individual. In this regard, I did not understand the rationale in the last sentence, referring to Reichard et al 2020.

Since this discussion is about junco song, the argument developed here does not appear supported. Also, I did not understand how plasticity would help explain your results, and their difference relative to past work on juncos, without changes in the environment or ecology? (please see comment on line 252).

299-308 – I agree partially with the argument here. A weak point of the song sharing argument is that juncos share very few songs with neighbors (Newman et al. 2008; see also discussion in Cardoso & Reichard 2016). The argument about junco repertoires harboring a large variation in frequency is valid, but again it does not explain differences in relation to past work. Instead, the fact that individual and population repertoires are acoustically diverse raises the issue of whether sample sizes were adequate (please see main comment 1).

309-318 – This argument appears to potentially undermine the design of the study. The choice of urban and non-urban sites appeared clear-cut when reading the first paragraph of the methods, but this paragraph puts it into question. Please clarify whether or not there were clear differences in anthropogenic noise levels between the sites classified as urban and non-urban. Namely, what were the habitats and the exposure to traffic in the non-urban sites, and how does that compare with habitat and exposure to traffic in the non-urban sites used in previous work (Slabbekoorn et al. 2007 Condor, Cardoso & Atwell 2011 Evolution)?

318 – Figure 2 was not mentioned in the results. This figure has problems and lacks explanations: the colors are difficult to distinguish and do not follow a logical order, the acronyms in the color legend were not defined and do not appear ordered logically, there appear to be two UCSD bars but this was not explained, boxplots are not well described (vertical lines are not the full range of data); font size and overall appearance need re-touching for clarity.

319-334 – Similar comment to that in lines 309-318. Also, do noise levels differ among the different urban sites? Lines 319-321 appear to belong to this paragraph, rather than the previous..

335-345 – This too is a reason for not lumping all urban populations in a single group. I think that this information should have appeared in the introduction, and that it should have been used to shape the hypothesis-testing goals and analyses.

355-367 – I think I am not partial saying that this paragraph does not adequately interpret past work on the UCSD population. While Slabbekoorn et al 2007 and Newman et al 2008 compared urban songs with a small number of mountain males (16 or less per mountain population), and differences in minimum frequency were sometimes statistically significant and sometimes not, when we studied a larger sample (101 UCSD and 50 LM males) this difference in frequency was very clear (Cardoso & Atwell 2011 Evolution, 2016 R Soc Open Sci).

362-363 – This is the most serious of issues, as it may invalidate the message in the abstract, title of the paper, and concluding paragraph. This statement, thus, does not suffice. Please see main comment 1.

367-375 – This is the good stuff! A novel and original idea that, unlike the previous paragraphs, might explain the different results in this and past studies. This update on the natural history of the UCSD and other urban populations, together with the different results in this paper vs. past work on the same system, plus the change in minimum frequency in UCSD over the past 10 years, can all combine to support a very interesting case of strong acoustic adaptation to the

urban environment when the population is a sedentary island population, and subsequent erosion of that acoustic adaptation when philopatry is not strong anymore. This should definitely be in the introduction and in the results, and be used to shape / focus the hypotheses and analyses, rather than only briefly appearing here.

As is, focused on testing urban vs. non-urban differences in song, the article may be weak and not able to support its conclusions, because the non-urban samples are so small (please see the main comment 1). But the dataset for some urban populations, including UCSD at two different time points, is better, and the article could be re-oriented to focus on comparisons among the well-sampled urban populations.

Figure 4 – These are very interesting new data, and their implications potentially important. I think that these data merit a more careful presentation, for example distinguishing observations during winter (these could be mountain birds in the altitudinal migration) and during breeding and post-breeding season (these would indicate that the resident population had indeed increased). I would suggest a methods section and a results section for these new data.

It is difficult to distinguish small and large pins in such small panels, and I did not understand the difference between blue and red pins (there are many blue pins in places without sightings in the previous time interval...). Also, the maps do not have a scale, and it would be useful to place rectangles in Figure 1 indicating the areas of these maps.

380-381 – But differences among sites are not expected, given the mode of development of junco song, and the pattern geographic variation in song (see figure 13.2 in Cardoso & Reichard 2016).

382-395 – Perhaps the really biologically-meaningful difference in the results is not the comparison between UCSD in 2006/7 and UCLA in 2018, but the difference between UCSD in 2006/7 and 2018, with the difference with UCLA being just a consequence of that? Please see comment on lines 367-375. I realize that the P value for the longitudinal comparison at UCSD was marginal (0.06) and that a larger sample size of UCSD recordings might be useful. If useful, although at the time we did not publish our data, we would be happy to share it.

396-408 – This is another paragraph equating an explanation for which there is contrary evidence in juncos (see also comment on lines 282, 299). I agree that these points should be brought up. But, rather than equating these possibilities in long paragraphs to then go back and deny their applicability in juncos, I suggest that they could be stated more briefly and, from the onset, in the context of explaining why they are unlikely explanations in the junco.

409-413 – It is not clear how this relates to the interpretation of the results.

===PREPARING YOUR MANUSCRIPT===

Please ensure that you include an acknowledgements' section before your reference list/bibliography. This should acknowledge anyone who assisted with your work, but does not

qualify as an author per the guidelines at <https://royalsociety.org/journals/ethics-policies/openness/>.

===PREPARING YOUR REVISION IN SCHOLARONE===

- Ensure that your data access statement meets the requirements at <https://royalsociety.org/journals/authors/author-guidelines/#data>. You should ensure that you cite the dataset in your reference list. If you have deposited data etc in the Dryad repository, please include both the 'For publication' link and 'For review' link at this stage.
- If you are requesting an article processing charge waiver, you must select the relevant waiver option (if requesting a discretionary waiver, the form should have been uploaded at Step 3 'File upload' above).
- If you have uploaded ESM files, please ensure you follow the guidance at <https://royalsociety.org/journals/authors/author-guidelines/#supplementary-material> to include a suitable title and informative caption. An example of appropriate titling and captioning may be found at https://figshare.com/articles/Table_S2_from_Is_there_a_trade-off_between_peak_performance_and_performance_breadth_across_temperatures_for_aerobic_scops_in_teleost_fishes_/3843624.

Author's Response to Decision Letter for (RSOS-211063.R0)

See Appendix A.

RSOS-220178.R0

Review form: Reviewer 2

Is the manuscript scientifically sound in its present form?

No

Are the interpretations and conclusions justified by the results?

No

Is the language acceptable?

Yes

Do you have any ethical concerns with this paper?

No

Have you any concerns about statistical analyses in this paper?

Yes

Recommendation?

Major revision is needed (please make suggestions in comments)

Comments to the Author(s)

I now read the revised version of this article, and find that the authors made a very good, detailed and laborious revision.

One remaining concern, however, is about the power analysis, which I have difficulty interpreting. The method section is very short, only stating the model and name of the package (line 259), and the results (line 273) are also short: “showed high power (>0.99)”. I would like to ask authors to explain what this power analysis does, why only one result is shown for all acoustic traits, and how this result should be interpreted. I would think that statistical power is the function of three things, sample size, within-group variance in the data, and expected among group difference, and that the same model and dataset can have high power to detect a large mean difference between groups, but low power to detect a small difference between groups. In other words, what is the magnitude of the urban vs. non-urban difference for which these models show such high power (>0.99)? And what would be the power of the models to detect the sort of differences in urban vs. non-urban song generally reported in the literature for other species, or to detect the large difference shown before in the San Diego population (about 500 Hz)? As noted in the previous round of reviews (comment #1), I would think that interpretation of the negative results in the manuscript depends on answering these questions.

Other comments by line number:

Line 31 – In this revised version peak frequency is also analysed.

100 – Juncos can have a repertoire of 2 to 8 song types, but usually use only 1 or 2 per bout.

200-203 – I did not notice this before, but the papers here did not publish the dataset from the 2006/07 breeding season. Reference 70, cited here, published data for minimum frequency, but not for the other song traits. Reference 65, not cited here, did published data for the other traits as well, but averages per song type and not averages per male (and in this manuscript it appears that analyses are done per male?). I think that the full dataset per male was never published, but if useful I can provide it.

For clarity, please state where the dataset comes from, how the dataset was organized (e.g., mean values per male, per song type, per combination of song type & male..), and how it was then analyzed.

204-206 - I am not sure I understand which pseudo-replication issue is mentioned in these lines, and why it can “artificially inflate sample size”. The paper cited here (ref. 71) shows that song traits are mostly a property of individual song types and not of individual males (i.e., acoustic measurements are consistent for the same song type irrespective of the male singing it, but measurements are not consistent across the song type repertoire of individual males) and, therefore, argues that song types are better statistical units than males. It is not clear how, or if, this relates with the statement in lines 204-206, and why this statement refers to the UCSD 2006-07 season only, rather than to the other data as well.

Speculating a little, perhaps what happened is that the authors used the dataset from reference 65 (this is not clear; please see comment on line 200) which is a dataset per song type. And, because the number of song types recorded in the UCSD population (168) is larger than the number of males recorded (101), perhaps the authors used song type as a substitute for male and, thus, here mention “artificially inflating sample size”? Please note that variance of acoustic traits across song types is larger than variance across males (ref. 71), so that results of statistical tests may differ if using song type or male as statistical units, and the former should not be used as substitute for the latter. Again, if the dataset per male is useful I can provide it freely upon request.

217-218 – Not sure I follow this explanation: “counted the number of syllables, divided by the song length, and subtracted by 1 to obtain the trill rate”. Please clarify.

The most accurate manner of computing trill rate would be the number of syllables minus 1, then divided by the duration from the onset of the first syllable to the onset of the last syllable. Alternatively, rate could be computed as the total number of syllables divided by the duration from the onset of the first syllable to the end of the last syllable; this second option is slightly less accurate because the last syllable does not complete its cycle (it lacks the small interval afterwards)

227 – Perhaps remove “likely” here, because there is no ambiguity on the method used.

245 – Perhaps replace “were grouped as” by “were considered” (similarly to line 246), because data from these sites were not lumped into a single group.

257 – Why is it that random effects hinder computing statistical power?

275-277 – It is not clear how these estimates are interpreted because, although later in the discussion (lines 422-423) it is stated that that “our power analyses and effect size confidence intervals suggest that if differences exist, they are of small effect”, it was not explained why these estimated values are considered small.

301 – Typo (remove “and in”).

316 – Typo (appear not to differ...).

422-423 – If I understand well, the power analysis was done for the main model including all sites across California, and not for the specific comparison discussed in this paragraph (longitudinal comparison in UCSD). Please rephrase to avoid suggesting that statistical power is high for this specific comparison.

Figure 2, panel F – Bandwidth cannot be negative (cf. line 231: “we subtracted the log₁₀ transformed minimum frequency from the log₁₀ transformed maximum frequency”).

Figure 3 – Please check the scale of the Y axis. Values do not appear to be in log₁₀kHz (compare with axes in figure 2).

Figure 4 – The figure reads better now, but some weaknesses remain:

- Panels referring to the 1980's and 1990's are empty even in the native range, where juncos did exist. This probably means that there are no reliable data on ebird for those decades, making those panels not useful or maybe even misleading. If this is true, please discuss it in the text, and perhaps consider removing these panels. Keeping only the panels for the 2000's and 2010's appears sufficient for the purposes of the article.

- Font size in the maps and, especially, the scale are too small to be readable. Please consider increasing font size, especially in the scale.

Review form: Reviewer 3

Is the manuscript scientifically sound in its present form?

No

Are the interpretations and conclusions justified by the results?

No

Is the language acceptable?

Yes

Do you have any ethical concerns with this paper?

No

Have you any concerns about statistical analyses in this paper?

Yes

Recommendation?

Major revision is needed (please make suggestions in comments)

Comments to the Author(s)

In this study, the authors use the songs of seven populations (4 urban, 3 rural) of dark-eyed juncos in southern California to test for differences between urban and non-urban juncos. They measured 6 different song characteristics and found limited evidence for differences. This result was surprising given the assumed differences in song transmission constraints between the urban and non-urban sites and previously published comparisons among junco populations in the same area. They also compare one more recently established urban population to an older urban population at two different time points.

This article has already been reviewed by two additional reviewers and extensive revisions have been made that improve the paper. However, many concerns remain.

Major Points:

1. The sound analysis and sampling effort remain difficult to follow, and some of that confusion is rooted in the use of different terms such as song, song bout, and syllable. It's not clear to me whether the measured unit was "songs" or "syllables" (see more detail below in comments on L232-234). Was every recorded song for each individual male measured and then averaged before the analysis? That approach would have the effect of wiping out a large amount of variation given that each male typically sings multiple song types that are acoustically distinct (see points below and Cardoso et al. 2009). Song types are generally differentiated from one another by visual inspection of the shape of the syllable being repeated in the trill. Also, not accounting for shared song types across males still leads to pseudoreplication because the same song type is being sampled multiple times. This concern is important because the sample sizes in some of these populations are already quite low.
2. Related to point 1, providing the recording effort measured as "song bouts" is not as informative as total number of songs and total number of song types recorded across all bouts (L161-162). The ideal presentation here would be to include the average number of songs per male, the range of the number of songs recorded per male, and the number of song types recorded per individual male (both average and range). Also, in L162-165 are you referring to the average number of songs recorded per male?
3. The "HowLoud" data are informative, but the manuscript does not provide enough context for the reader to evaluate the validity of those data. How are the measurements collected? In the discussion, the authors undercut the argument that these sites have distinct noise profiles by arguing that urban encroachment on rural areas could explain the lack of differences in song characteristics. It is important to note that detailed measurements of these noise profiles are lacking in the current study and essential for the future.

4. Is a power analysis not possible for the GLMM? Reviewer 2 laid out a detailed approach to a power analysis, but that recommendation seems to have been mostly ignored without much justification. I have almost no experience in this realm of statistics, so am unable to advise.

5. The discussion section could be more concise and focused. I include many suggestions below.

Minor Points:

L31 – Peak Frequency?

L68-69 – Please combine these paragraphs.

L71-73; 75-76 – Suggest combining these ideas. The link between increased amplitude and increased frequency is the Lombard Effect. Perhaps move the intervening point about song length and frequency to the end of this paragraph as another example of independence between frequency and other song traits?

L100 – “with approximately two to eight songs per bout” – it would be more consistent with the literature to refer to this as repertoire size rather than songs per bout. For example, males will often sing only one song in a single song bout. They do not necessarily cycle through their entire repertoire in every song bout.

L102-108 – Instead of picking two studies here, I recommend focusing on broader trends. Geographic variation in junco song has been reviewed previously by Cardoso & Reichard 2016, so it might help to highlight that work, which includes data from populations other than the urban v. rural comparison in San Diego.

It seems important to note that junco song repertoires are small, but song type sharing is low, which means that there is a large amount of variation between males. Also, later work by Cardoso and colleagues, with much larger sample sizes, found clear differences in minimum but not maximum frequency. See Cardoso and Atwell 2011, *Evolution* 65-1: 295–300, and Cardoso et al. 2009 *Behav. Ecol.* 20:901–907. This disparity in results across years also presents an opportunity to emphasize how sampling at different points in time could lead to different outcomes as populations potentially evolve in response to noise and other pressures. It also highlights the potential importance of sample size. Both issues are highly relevant to this study.

Finally, in this paragraph, it seems helpful to refer to the population as “San Diego” rather than “urban juncos (L105, 110, etc)” because this is a population that features prominently in this study.

L112 – The Reichard et al. study emphasized not only parental effects, but also early life exposure to noise as a mechanism maintaining this divergence.

L118-119 – I’m not sure if the second point is tested in this study. The authors make assumptions about differences in habitat and noise, but there is very little quantitative data presented in support of either assumption.

L119-120 – “whether changes in song...” This study compares whether two urban populations with similar time since colonization produce similar songs. This comparison does not really assess how much the songs in each population have changed since colonization, because there isn’t a baseline comparison for either population. Basically, we have no idea how similar the

songs of the founders were in each population and cannot assess how much they have changed. I recommend striking “changes” from this point.

L132-135 – What is the actual source of these measurements? Is “HowLoud” relying on decibel meters on the ground or some other measure? This tool will be unfamiliar to most and more background is needed here.

L137-139 – It’s very difficult to interpret these data as presented. Can the sound scores be anchored to specific decibel levels?

L162-164 – “Mean number of songs...” – unclear what is meant by “songs” here. Song types? Songs recorded? In the previous and subsequent sentence the unit is song bouts.

L165 – Please move the definition of “song bout” earlier so that it appears before the term is used for the first time.

L168-169 – Unclear what is meant here. If accurate, perhaps change to “In urban locations, males were identified and differentiated by color bands. A subset of males ($N=X$) were unbanded, but unbanded males consistently defended stable territories neighbored by banded males, which made it possible to differentiate among unbanded individuals.”

You haven’t mentioned anything about the banding status of non-urban males. How were you able to differentiate between males? This issue was explained more thoroughly in your response to the editor. Please include more of that information here.

L196-198 – Another point to add is that these recordings may have been initially stored in a compressed audio format if they were recorded on a phone, which is probably likely.

L204-206 – And this is a pseudoreplication issue that you cannot control for statistically? For example, you can add song type as a random effect in your models. Please elaborate on the specifics of this issue.

L214-215 – What is meant here by “average” minimum and maximum frequency? Below you report that you measured each syllable within a song bout, which could be hundreds of songs and syllables that were then averaged. Also, why is peak frequency the only measure not listed as “average?”

L225-228 – Despite the limitations here, I think that this approach is reasonable. However, the authors should point out here that these visual measurements are not reliably repeatable among individuals and likely increased the amount of variation within each dataset, which could mask population differences.

L232-234 – Some of this information needs to be moved earlier. A variable number of individual songs were recorded from each male (short v. long bouts). Did you measure every individual syllable in every song for every male and then calculate average values for each male as this section suggests? Averaging individual syllables rather than songs is problematic because you aren’t really measuring the minimum and maximum frequency of the entire song. The minimum and maximum frequency within a trill usually occurs in the middle syllables, so if you average the middle syllables with the rest of the syllables in the song, you will decrease the true frequency bandwidth of the song. Based on your description of your method for creating the selection box (L216-217) and the available raw data, it seems like you were measuring “songs” not “syllables,” but this section is giving me pause. Please clarify your terminology and methodology.

In addition, averaging multiple song types from each male is not the best approach given what we know about junco song. Junco song types are highly repeatable across males, but the individual songs within a male's repertoire have distinct acoustic characteristics (Cardoso et al. 2009). So, by average song types for each male, the total variation in each population has been reduced. By shifting the focus to song types rather than individual males, the manuscript might actually increase its sample size because there were likely more song types recorded than individual males.

L237-246 – Were the Repository songs not included in this analysis? Those songs are not mentioned in the analysis section, but they show up in the methods and in Figure 2.

L252-253 – How should these Effect Sizes be interpreted? For example, is there a threshold for what constitutes a small/medium/large effect?

L254-255 – Unnecessary repetition here with line 251.

L257-258 – Why is this the case? A previous reviewer gave a detailed explanation for how a power analysis could be undertaken.

L271-274 – If location is eliminated as a random effect, do these results still hold?

L273-274 – This power analysis was the same for all variables measured? Also, it seems like the current approach has only assessed power for the urban v. nonurban comparison, which has a much larger sample size than the individual population-level comparisons. If accurate, you should clarify here that you did not assess power for the among populations comparisons.

L306-308 – Peak Frequency?

L308-309 – It would also be useful here to highlight the point that dark-eyed junco song is highly variable due to the large amount of innovation and improvisation that happens during song development. As a result, few males share songs with their neighbors. This amount of variation adds risk to small sample sizes by increasing the likelihood that outliers will be captured and skew the data.

L310 – “real differences between populations” - The power analysis and effect sizes only assess differences between urban and non-urban juncos, so the term “populations” here is misleading because these data are a collection of multiple populations lumped into two discrete categories.

L315 – “similar noisy environment” - This is an untested assumption.

L318-319 – “across urban and non-urban settings...” - this phrasing suggests multiple points of comparison when it seems to be one urban and one rural population.

L331 – “the urban...” environment?

L335-352 – This paragraph expends a lot of text on an explanation that is not consistent with the result of the current study (although it wasn't directly tested because background noise levels were not measured), and not supported by a more direct test of the plasticity hypothesis in juncos (Reichard et al. 2020). I recommend reducing the text in this paragraph by half and cutting most of the discussion of specific species. Essentially, plasticity is one explanation that has some support, but not in juncos.

L353-363 – This paragraph has similar limitations to the previous paragraph. Juncos have small repertoires, and they don't share many songs among males. So, they are very limited in their repertoire-based plasticity. I think this paragraph should be largely removed from the manuscript. The potential for plasticity in song types can be mentioned in the previous paragraph, but again, it isn't an explanation that is tested in this study, and it's also unlikely to be prominent in juncos.

L372 – This example was not statistically significant and doesn't seem worth highlighting given the small sample size and presumably low statistical power. This paragraph also represents a possibility that is speculative and outside the scope of the study at hand. There are no direct measurements of noise pollution at any of these sites, and even with cases of urban encroachment, the noise pollution is far less severe and consistent.

L396 – Important to note here that 1980 to early 2000s was enough time for distinct difference to appear between UCSD and distant rural populations.

L391-410 – These two paragraphs could be combined to eliminate repetition. The Zollinger et al. and Reichard et al. studies both came to a similar conclusion with respect to noise-induced plasticity. The UCSD population was “older” when they were sampled, but only 10-20 years older. So, at UCLA/UCSB/Occidental it could be that insufficient time has passed or some effect of the colonizing population as the authors assert.

L411-423 – It seems strange here to lead with studies that had smaller sample sizes and were conducted closer to the original colonization event at UCSD. The previous paragraphs just emphasized the point that a lack of population difference could be explained by an insufficient amount of time to diverge. So, it seems prudent to emphasize later studies with larger sample sizes, more statistical power, and a longer time since colonization.

L436 – “found non-statistically significant difference” – change to “found no detectable differences”

L436-437 – In this newer analysis, UCSD 2006/2007 was not directly compared to the other sites besides UCLA, so this statement doesn't seem appropriate.

L441 – “despite strong differences in noise” – please note that these differences were not actually measured.

L448-451 – This paragraph is mostly spent arguing why this result might not be meaningful without any interpretation of why we might expect the population to shift over the intervening decade. It might be worth linking this potential difference to the nice description in the previous paragraph relating to how these urban populations are no longer islands and an increase in gene flow/immigration could shift some of these traits even if the noise profile remains constant.

L453-454 – change “breeding seasons” to “sampling years”

L460-462 – “did not adjust their song characteristics in response to urban noise in a similar way...” – an underlying assumption here is that the founding populations had similar existing variation in song and/or the acoustic environments at UCLA and UCSD are similar. Neither assumption is tested in this study. I recommend eliminating this interpretation. This study shows that these two populations differ in their song structure, but the underlying causes and evolutionary history of any acoustic shifts remain unknown.

Conclusion – I recommend highlighting the potential for continued shifts over time in all these populations. The two UCSD comparisons were interesting and should not be totally discounted here. Also, it is important to highlight the importance of measuring noise pollution alongside song to better assess environmental selective pressures and the potential for noise-induced plasticity.

Figure 2 – “Repository” is not explained in the figure caption. Given that the new analysis has effectively collapsed all of these individual populations into separate urban and non-urban groups, I recommend adding an additional set of bars to each graph that shows the data for the entire urban and non-urban groups.

Figure 4 – This is an amazing visual of how rapidly these populations have expanded!

Decision letter (RSOS-220178.R0)

Dear Ms Wong

The Editors assigned to your paper RSOS-220178 "No evidence of repeated song divergence across multiple urban and non-urban populations of dark-eyed juncos (*Junco hyemalis*) in Southern California" have now received comments from reviewers and would like you to revise the paper in accordance with the reviewer comments and any comments from the Editors. Please note this decision does not guarantee eventual acceptance.

Please submit your revised manuscript and required files (see below) no later than 21 days from today's (ie 10-May-2022) date. Note: the ScholarOne system will 'lock' if submission of the revision is attempted 21 or more days after the deadline. If you do not think you will be able to meet this deadline please contact the editorial office immediately.

on behalf of Dr David Wilson (Associate Editor) and Kevin Padian (Subject Editor)
 openscience@royalsociety.org

Editor comments:

Thank you for your resubmission. We note that many comments from reviewers on a previous submission to another journal were not addressed sufficiently in their view, and that still some lingering concerns remain. We ask you to address these diligently because otherwise we will not be able to consider the manuscript further. Best wishes with your revisions.

Associate Editor Comments to Author (Dr David Wilson):

This revised manuscript was evaluated by one of the original reviewers and by a new expert in the field. Both reviewers and I agree that the revisions were extensive and significantly improved the manuscript, but we also note that several issues remain. The reviewers provide detailed feedback that is consistent with my own assessment of the manuscript. Some of the key issues include clarifying the power analysis, explaining the data/models underlying the background noise estimates, clarifying various aspects of the sound analysis (here a multi-panel figure showing spectrograms of a short song bout and a blow-out of one or two songs, with annotations of bouts, songs, song types, syllables, phrases, and key measurements; another panel showing a power spectrum and its associated measures, and whether the power spectrum is a mean power spectrum based on a syllable or the entire song), how the raw acoustic data were organized and reduced prior to analysis, whether song type should be considered in the analysis, and condensing the somewhat lengthy discussion. In addition to the reviewers' comments, I would like the authors to reduce redundancy in the figures and tables. For example, table 2 shows mean \pm SE per trait per location, but this is largely redundant with figure 2 and could be deleted. Tables 3 and 4 show p-values, effect sizes, and 95% CIs, yet most of these numbers are reported directly in the text; any missing values, such as the 6 p-values from table 3, could be incorporated into the text and the tables deleted. Please address these comments and all of the excellent comments provided by the reviewers.

Reviewer comments to Author:

Reviewer: 2

Comments to the Author(s)

I now read the revised version of this article, and find that the authors made a very good, detailed and laborious revision.

One remaining concern, however, is about the power analysis, which I have difficulty interpreting. The method section is very short, only stating the model and name of the package (line 259), and the results (line 273) are also short: "showed high power (>0.99)". I would like to ask authors to explain what this power analysis does, why only one result is shown for all acoustic traits, and how this result should be interpreted. I would think that statistical power is the function of three things, sample size, within-group variance in the data, and expected among group difference, and that the same model and dataset can have high power to detect a large mean difference between groups, but low power to detect a small difference between groups. In other words, what is the magnitude of the urban vs. non-urban difference for which these models show such high power (>0.99)? And what would be the power of the models to detect the sort of differences in urban vs. non-urban song generally reported in the literature for other species, or to detect the large difference shown before in the San Diego population (about 500

Hz)? As noted in the previous round of reviews (comment #1), I would think that interpretation of the negative results in the manuscript depends on answering these questions.

Other comments by line number:

Line 31 – In this revised version peak frequency is also analysed.

100 – Juncos can have a repertoire of 2 to 8 song types, but usually use only 1 or 2 per bout.

200-203 – I did not notice this before, but the papers here did not publish the dataset from the 2006/07 breeding season. Reference 70, cited here, published data for minimum frequency, but not for the other song traits. Reference 65, not cited here, did published data for the other traits as well, but averages per song type and not averages per male (and in this manuscript it appears that analyses are done per male?). I think that the full dataset per male was never published, but if useful I can provide it.

For clarity, please state where the dataset comes from, how the dataset was organized (e.g., mean values per male, per song type, per combination of song type & male.), and how it was then analyzed.

204-206 - I am not sure I understand which pseudo-replication issue is mentioned in these lines, and why it can “artificially inflate sample size”. The paper cited here (ref. 71) shows that song traits are mostly a property of individual song types and not of individual males (i.e., acoustic measurements are consistent for the same song type irrespective of the male singing it, but measurements are not consistent across the song type repertoire of individual males) and, therefore, argues that song types are better statistical units than males. It is not clear how, or if, this relates with the statement in lines 204-206, and why this statement refers to the UCSD 2006-07 season only, rather than to the other data as well.

Speculating a little, perhaps what happened is that the authors used the dataset from reference 65 (this is not clear; please see comment on line 200) which is a dataset per song type. And, because the number of song types recorded in the UCSD population (168) is larger than the number of males recorded (101), perhaps the authors used song type as a substitute for male and, thus, here mention “artificially inflating sample size”? Please note that variance of acoustic traits across song types is larger than variance across males (ref. 71), so that results of statistical tests may differ if using song type or male as statistical units, and the former should not be used as substitute for the latter. Again, if the dataset per male is useful I can provide it freely upon request.

217-218 – Not sure I follow this explanation: “counted the number of syllables, divided by the song length, and subtracted by 1 to obtain the trill rate”. Please clarify.

The most accurate manner of computing trill rate would be the number of syllables minus 1, then divided by the duration from the onset of the first syllable to the onset of the last syllable. Alternatively, rate could be computed as the total number of syllables divided by the duration from the onset of the first syllable to the end of the last syllable; this second option is slightly less accurate because the last syllable does not complete its cycle (it lacks the small interval afterwards)

227 – Perhaps remove “likely” here, because there is no ambiguity on the method used.

245 – Perhaps replace “were grouped as” by “were considered” (similarly to line 246), because data from these sites were not lumped into a single group.

257 – Why is it that random effects hinder computing statistical power?

275-277 – It is not clear how these estimates are interpreted because, although later in the discussion (lines 422-423) it is stated that that “our power analyses and effect size confidence intervals suggest that if differences exist, they are of small effect”, it was not explained why these estimated values are considered small.

301 – Typo (remove “and in”).

316 – Typo (appear not to differ...).

422-423 – If I understand well, the power analysis was done for the main model including all sites across California, and not for the specific comparison discussed in this paragraph (longitudinal comparison in UCSD). Please rephrase to avoid suggesting that statistical power is high for this specific comparison.

Figure 2, panel F – Bandwidth cannot be negative (cf. line 231: “we subtracted the log10 transformed minimum frequency from the log10 transformed maximum frequency”).

Figure 3 – Please check the scale of the Y axis. Values do not appear to be in log10kHz (compare with axes in figure 2).

Figure 4 – The figure reads better now, but some weaknesses remain:

- Panels referring to the 1980’s and 1990’s are empty even in the native range, where juncos did exist. This probably means that there are no reliable data on ebird for those decades, making those panels not useful or maybe even misleading. If this is true, please discuss it in the text, and perhaps consider removing these panels. Keeping only the panels for the 2000’s and 2010’s appears sufficient for the purposes of the article.
- Font size in the maps and, especially, the scale are too small to be readable. Please consider increasing font size, especially in the scale.

Reviewer: 3

Comments to the Author(s)

In this study, the authors use the songs of seven populations (4 urban, 3 rural) of dark-eyed juncos in southern California to test for differences between urban and non-urban juncos. They measured 6 different song characteristics and found limited evidence for differences. This result was surprising given the assumed differences in song transmission constraints between the urban and non-urban sites and previously published comparisons among junco populations in the same area. They also compare one more recently established urban population to an older urban population at two different time points.

This article has already been reviewed by two additional reviewers and extensive revisions have been made that improve the paper. However, many concerns remain.

Major Points:

1. The sound analysis and sampling effort remain difficult to follow, and some of that confusion is rooted in the use of different terms such as song, song bout, and syllable. It’s not clear to me whether the measured unit was “songs” or “syllables” (see more detail below in comments on L232-234). Was every recorded song for each individual male measured and then averaged before the analysis? That approach would have the effect of wiping out a large amount of variation given that each male typically sings multiple song types that are acoustically distinct (see points below and Cardoso et al. 2009). Song types are generally differentiated from one another by visual inspection of the shape of the syllable being repeated in the trill. Also, not accounting for

shared song types across males still leads to pseudoreplication because the same song type is being sampled multiple times. This concern is important because the sample sizes in some of these populations are already quite low.

2. Related to point 1, providing the recording effort measured as “song bouts” is not as informative as total number of songs and total number of song types recorded across all bouts (L161-162). The ideal presentation here would be to include the average number of songs per male, the range of the number of songs recorded per male, and the number of song types recorded per individual male (both average and range). Also, in L162-165 are you referring to the average number of songs recorded per male?

3. The “HowLoud” data are informative, but the manuscript does not provide enough context for the reader to evaluate the validity of those data. How are the measurements collected? In the discussion, the authors undercut the argument that these sites have distinct noise profiles by arguing that urban encroachment on rural areas could explain the lack of differences in song characteristics. It is important to note that detailed measurements of these noise profiles are lacking in the current study and essential for the future.

4. Is a power analysis not possible for the GLMM? Reviewer 2 laid out a detailed approach to a power analysis, but that recommendation seems to have been mostly ignored without much justification. I have almost no experience in this realm of statistics, so am unable to advise.

5. The discussion section could be more concise and focused. I include many suggestions below.

Minor Points:

L31 - Peak Frequency?

L68-69 - Please combine these paragraphs.

L71-73; 75-76 - Suggest combining these ideas. The link between increased amplitude and increased frequency is the Lombard Effect. Perhaps move the intervening point about song length and frequency to the end of this paragraph as another example of independence between frequency and other song traits?

L100 - “with approximately two to eight songs per bout” - it would be more consistent with the literature to refer to this as repertoire size rather than songs per bout. For example, males will often sing only one song in a single song bout. They do not necessarily cycle through their entire repertoire in every song bout.

L102-108 - Instead of picking two studies here, I recommend focusing on broader trends. Geographic variation in junco song has been reviewed previously by Cardoso & Reichard 2016, so it might help to highlight that work, which includes data from populations other than the urban v. rural comparison in San Diego.

It seems important to note that junco song repertoires are small, but song type sharing is low, which means that there is a large amount of variation between males. Also, later work by Cardoso and colleagues, with much larger sample sizes, found clear differences in minimum but not maximum frequency. See Cardoso and Atwell 2011, *Evolution* 65-1: 295-300, and Cardoso et al. 2009 *Behav. Ecol.* 20:901-907. This disparity in results across years also presents an opportunity to emphasize how sampling at different points in time could lead to different outcomes as populations potentially evolve in response to noise and other pressures. It also highlights the potential importance of sample size. Both issues are highly relevant to this study.

Finally, in this paragraph, it seems helpful to refer to the population as “San Diego” rather than “urban juncos (L105, 110, etc)” because this is a population that features prominently in this study.

L112 - The Reichard et al. study emphasized not only parental effects, but also early life exposure to noise as a mechanism maintaining this divergence.

L118-119 - I’m not sure if the second point is tested in this study. The authors make assumptions about differences in habitat and noise, but there is very little quantitative data presented in support of either assumption.

L119-120 - “whether changes in song...” This study compares whether two urban populations with similar time since colonization produce similar songs. This comparison does not really assess how much the songs in each population have changed since colonization, because there isn’t a baseline comparison for either population. Basically, we have no idea how similar the songs of the founders were in each population and cannot assess how much they have changed. I recommend striking “changes” from this point.

L132-135 - What is the actual source of these measurements? Is “HowLoud” relying on decibel meters on the ground or some other measure? This tool will be unfamiliar to most and more background is needed here.

L137-139 - It’s very difficult to interpret these data as presented. Can the sound scores be anchored to specific decibel levels?

L162-164 - “Mean number of songs...” - unclear what is meant by “songs” here. Song types? Songs recorded? In the previous and subsequent sentence the unit is song bouts.

L165 - Please move the definition of “song bout” earlier so that it appears before the term is used for the first time.

L168-169 - Unclear what is meant here. If accurate, perhaps change to “In urban locations, males were identified and differentiated by color bands. A subset of males (N=X) were unbanded, but unbanded males consistently defended stable territories neighbored by banded males, which made it possible to differentiate among unbanded individuals.”

You haven’t mentioned anything about the banding status of non-urban males. How were you able to differentiate between males? This issue was explained more thoroughly in your response to the editor. Please include more of that information here.

L196-198 - Another point to add is that these recordings may have been initially stored in a compressed audio format if they were recorded on a phone, which is probably likely.

L204-206 - And this is a pseudoreplication issue that you cannot control for statistically? For example, you can add song type as a random effect in your models. Please elaborate on the specifics of this issue.

L214-215 - What is meant here by “average” minimum and maximum frequency? Below you report that you measured each syllable within a song bout, which could be hundreds of songs and syllables that were then averaged. Also, why is peak frequency the only measure not listed as “average?”

L225-228 – Despite the limitations here, I think that this approach is reasonable. However, the authors should point out here that these visual measurements are not reliably repeatable among individuals and likely increased the amount of variation within each dataset, which could mask population differences.

L232-234 – Some of this information needs to be moved earlier. A variable number of individual songs were recorded from each male (short v. long bouts). Did you measure every individual syllable in every song for every male and then calculate average values for each male as this section suggests? Averaging individual syllables rather than songs is problematic because you aren't really measuring the minimum and maximum frequency of the entire song. The minimum and maximum frequency within a trill usually occurs in the middle syllables, so if you average the middle syllables with the rest of the syllables in the song, you will decrease the true frequency bandwidth of the song. Based on your description of your method for creating the selection box (L216-217) and the available raw data, it seems like you were measuring "songs" not "syllables," but this section is giving me pause. Please clarify your terminology and methodology.

In addition, averaging multiple song types from each male is not the best approach given what we know about junco song. Junco song types are highly repeatable across males, but the individual songs within a male's repertoire have distinct acoustic characteristics (Cardoso et al. 2009). So, by average song types for each male, the total variation in each population has been reduced. By shifting the focus to song types rather than individual males, the manuscript might actually increase its sample size because there were likely more song types recorded than individual males.

L237-246 – Were the Repository songs not included in this analysis? Those songs are not mentioned in the analysis section, but they show up in the methods and in Figure 2.

L252-253 – How should these Effect Sizes be interpreted? For example, is there a threshold for what constitutes a small/medium/large effect?

L254-255 – Unnecessary repetition here with line 251.

L257-258 – Why is this the case? A previous reviewer gave a detailed explanation for how a power analysis could be undertaken.

L271-274 – If location is eliminated as a random effect, do these results still hold?

L273-274 – This power analysis was the same for all variables measured? Also, it seems like the current approach has only assessed power for the urban v. nonurban comparison, which has a much larger sample size than the individual population-level comparisons. If accurate, you should clarify here that you did not assess power for the among populations comparisons.

L306-308 – Peak Frequency?

L308-309 – It would also be useful here to highlight the point that dark-eyed junco song is highly variable due to the large amount of innovation and improvisation that happens during song development. As a result, few males share songs with their neighbors. This amount of variation adds risk to small sample sizes by increasing the likelihood that outliers will be captured and skew the data.

L310 – "real differences between populations" - The power analysis and effect sizes only assess differences between urban and non-urban juncos, so the term "populations" here is misleading because these data are a collection of multiple populations lumped into two discrete categories.

L315 - “similar noisy environment” - This is an untested assumption.

L318-319 - “across urban and non-urban settings...” - this phrasing suggests multiple points of comparison when it seems to be one urban and one rural population.

L331 - “the urban...” environment?

L335-352 - This paragraph expends a lot of text on an explanation that is not consistent with the result of the current study (although it wasn't directly tested because background noise levels were not measured), and not supported by a more direct test of the plasticity hypothesis in juncos (Reichard et al. 2020). I recommend reducing the text in this paragraph by half and cutting most of the discussion of specific species. Essentially, plasticity is one explanation that has some support, but not in juncos.

L353-363 - This paragraph has similar limitations to the previous paragraph. Juncos have small repertoires, and they don't share many songs among males. So, they are very limited in their repertoire-based plasticity. I think this paragraph should be largely removed from the manuscript. The potential for plasticity in song types can be mentioned in the previous paragraph, but again, it isn't an explanation that is tested in this study, and it's also unlikely to be prominent in juncos.

L372 - This example was not statistically significant and doesn't seem worth highlighting given the small sample size and presumably low statistical power. This paragraph also represents a possibility that is speculative and outside the scope of the study at hand. There are no direct measurements of noise pollution at any of these sites, and even with cases of urban encroachment, the noise pollution is far less severe and consistent.

L396 - Important to note here that 1980 to early 2000s was enough time for distinct difference to appear between UCSD and distant rural populations.

L391-410 - These two paragraphs could be combined to eliminate repetition. The Zollinger et al. and Reichard et al. studies both came to a similar conclusion with respect to noise-induced plasticity. The UCSD population was “older” when they were sampled, but only 10-20 years older. So, at UCLA/UCSB/Occidental it could be that insufficient time has passed or some effect of the colonizing population as the authors assert.

L411-423 - It seems strange here to lead with studies that had smaller sample sizes and were conducted closer to the original colonization event at UCSD. The previous paragraphs just emphasized the point that a lack of population difference could be explained by an insufficient amount of time to diverge. So, it seems prudent to emphasize later studies with larger sample sizes, more statistical power, and a longer time since colonization.

L436 - “found non-statistically significant difference” - change to “found no detectable differences”

L436-437 - In this newer analysis, UCSD 2006/2007 was not directly compared to the other sites besides UCLA, so this statement doesn't seem appropriate.

L441 - “despite strong differences in noise” - please note that these differences were not actually measured.

L448-451 – This paragraph is mostly spent arguing why this result might not be meaningful without any interpretation of why we might expect the population to shift over the intervening decade. It might be worth linking this potential difference to the nice description in the previous paragraph relating to how these urban populations are no longer islands and an increase in gene flow/immigration could shift some of these traits even if the noise profile remains constant.

L453-454 – change “breeding seasons” to “sampling years”

L460-462 – “did not adjust their song characteristics in response to urban noise in a similar way...” – an underlying assumption here is that the founding populations had similar existing variation in song and/or the acoustic environments at UCLA and UCSD are similar. Neither assumption is tested in this study. I recommend eliminating this interpretation. This study shows that these two populations differ in their song structure, but the underlying causes and evolutionary history of any acoustic shifts remain unknown.

Conclusion – I recommend highlighting the potential for continued shifts over time in all these populations. The two UCSD comparisons were interesting and should not be totally discounted here. Also, it is important to highlight the importance of measuring noise pollution alongside song to better assess environmental selective pressures and the potential for noise-induced plasticity.

Figure 2 – “Repository” is not explained in the figure caption. Given that the new analysis has effectively collapsed all of these individual populations into separate urban and non-urban groups, I recommend adding an additional set of bars to each graph that shows the data for the entire urban and non-urban groups.

Figure 4 – This is an amazing visual of how rapidly these populations have expanded!

===PREPARING YOUR MANUSCRIPT===

Your revised paper should include the changes requested by the referees and Editors of your manuscript. You should provide two versions of this manuscript and both versions must be provided in an editable format:
 one version identifying all the changes that have been made (for instance, in coloured highlight, in bold text, or tracked changes);
 a 'clean' version of the new manuscript that incorporates the changes made, but does not highlight them. This version will be used for typesetting if your manuscript is accepted.

If you have been asked to revise the written English in your submission as a condition of publication, you must do so, and you are expected to provide evidence that you have received

language editing support. The journal would prefer that you use a professional language editing service and provide a certificate of editing, but a signed letter from a colleague who is a fluent speaker of English is acceptable. Note the journal has arranged a number of discounts for authors using professional language editing services (<https://royalsociety.org/journals/authors/benefits/language-editing/>).

===PREPARING YOUR REVISION IN SCHOLARONE===

<https://royalsociety.org/journals/authors/author-guidelines/#supplementary-material> to

include a suitable title and informative caption. An example of appropriate titling and captioning may be found at https://figshare.com/articles/Table_S2_from_Is_there_a_trade-off_between_peak_performance_and_performance_breadth_across_temperatures_for_aerobic_scops_in_teleost_fishes_/3843624.

Author's Response to Decision Letter for (RSOS-220178.R0)

See Appendix B.

Decision letter (RSOS-220178.R1)

Dear Ms Wong

The Editors assigned to your paper RSOS-220178.R1 "No evidence of repeated song divergence across multiple urban and non-urban populations of dark-eyed juncos (*Junco hyemalis*) in Southern California" have now received comments from reviewers and would like you to revise the paper in accordance with the reviewer comments and any comments from the Editors. Please note this decision does not guarantee eventual acceptance.

Please submit your revised manuscript and required files (see below) no later than 21 days from today's (ie 06-Jun-2022) date. Note: the ScholarOne system will 'lock' if submission of the revision is attempted 21 or more days after the deadline. If you do not think you will be able to meet this deadline please contact the editorial office immediately.

on behalf of Dr David Wilson (Associate Editor) and Kevin Padian (Subject Editor)
 openscience@royalsociety.org

Editor comments:

Thank you for your attention to the reviewers' comments. As you see, our AE feels that the MS has been much improved but also that there remain some issues to address. Please do so as effectively as you have done with previous comments, because we will not be able to entertain the manuscript further if these are not addressed. Best wishes.

Associate Editor Comments to Author (Dr David Wilson):

The authors have invested considerable effort and addressed most of the comments raised in the second round of review. However, there remains several issues that need to be addressed. Line numbers in my comments below refer to the version without track changes.

L99: change wording to '...song types, though singing males usually use only...'

L134: this sentence could be made more informative because it is still unclear whether HowLoud is interpolating among SPL values measured on the ground at certain locations, or modeling based on the distribution of known noise sources. I think it is the latter. Perhaps use similar wording to that used in your response letter (assuming it is your own and not derived directly from HowLoud): eg, 'HowLoud utilizes an established transportation noise model from the Federal Highway Administration to model ambient noise, taking into account landscape variables and traffic. The model takes into account the presence and distribution of local sources of noise (e.g., commercial sources) and proximity to airport(s).' The key point is that noise level estimates are derived from a model, not interpolated or extrapolated from actual noise measurements taken nearby the study sites.

L139: the sentence has contradictory statements; I believe it should say '50 (very loud) to 100 (very quiet)

L164: cite figure 2 when defining 'phrases' (and on L240 when describing spectrogram frequency measures). More generally, I think the term 'phrase' is somewhat confusing. I believe most researchers would consider the 'phrase' depicted in figure 2 as a 'song', and multiple songs as a song bout. Confusion persists because the term 'song' is used elsewhere, for instance, in the measure 'song length' on L223. I recommend using 'song' instead of 'phrase' throughout, but, at a minimum, being consistent.

L166: delete 'approximately' since it is a calculated average, and provide the range in the number of songs recorded per male as requested by the reviewer.

L218 (and throughout): replace 'power spectra' with 'mean power spectrum' since your spectrum is based on a selection of time rather than an instant of time (both are possible in Raven). Use 'spectrum' instead of 'spectra' because 'spectrum' is singular and better matches the singular 'spectrogram'.

L221-223: reword as: '...(1) minimum frequency, (2) maximum frequency, and (3) peak frequency of the middle syllable within each phrase, as well as the (4) length and (5) trill rate of the entire phrase.' [since length and trill rate are not based on the middle syllable]. Also, I would remove the averaging from this sentence since the sentence describes the raw measures. The averaging is explained separately below.

L229: change 'relative from' to 'relative to'

L230-231: Change wording to 'We measured minimum, maximum, and peak frequency from each phrase (or song?) within a song bout, and averaged...' Also, clarify whether averaging was done across all phrases recorded per song type from each male, or from across all urban or all rural males. I think averages were calculated across all phrases of the same song type from among all urban males, and separately from among all rural males, but this is still unclear. Perhaps a simple example of how the averaging was done would help.

L235-236: need to provide the mean, sd, and range for the number of song types recorded per male, as requested by reviewer 3.

L237: the wording here indicates that UCSD06/07 was compared to UCSD18-20 and to UCLA, but does not reflect that UCSD18-20 was also compared to UCLA (as expected under objective 2). Perhaps re-word as 'For the comparisons among UCSD06/07, UCSD18-20, and UCLA frequency measurements,...' Related to this, the comparison of UCSD18-20 to UCLA appears to be missing from the results (L309-323), though it is mentioned briefly in the figure 4 caption (i.e., that there were no significant differences) and referred to in the discussion (L449). Finally, it is now clear how comparing UCSD06/07 to UCLA relates to the objectives in the introduction. Objective 2 implies a comparison between UCSD18-20 and UCLA, and objective 3 implies a comparison between UCSD06/07 and UCSD18-20. Need to align the description of this analysis with the objectives.

L249: this sentence about averaging seems redundant with two paragraphs earlier and could be deleted

L294: change to '...were broadly similar among populations and between urban and rural juncos'

L324: need to mention briefly here what was different about these alternate analyses.

L328: I'm assuming you mean 'When comparing urban and rural juncos,...' rather than the 7 populations? Please clarify.

L328-333 and L340-344: the power analyses reported on these lines would benefit from a sentence in the methods explaining how to interpret the values. By 'power', do you mean power to detect a particular effect size? If so, what effect size? Also, please explain how the percentages are to be interpreted? Does 4% mean that the model had only a 4% probability of detecting a difference of a certain effect size in the urban versus rural comparison? If so, these numbers seem very low, and seem to strongly contrast with the relatively high power for detecting medium effect sizes reported on L333-339. A bit more explanation would be helpful.

L362-364: unclear which result this statement is based upon. When you say 'significant differences between UCLA and UCSD song traits', is this based on the findings that UCSD18-20 is not different from UCSD06/07, whereas UCLA and UCSD06/07 are different? If so, that could be stated more explicitly, especially since UCSD18-20 and UCLA are not different from each other (as noted in figure 4 caption).

L364-365 (and L374): It is similarly unclear which result supports the claim that San Diego juncos shifted their songs upon colonizing the city whereas UCLA juncos did not. There was no comparison of rural juncos and San Diego juncos at the time of colonization in the 1980s, or of rural juncos and UCLA juncos during their colonization in the 2000s. If anything, the results seem to show that UCLA songs diverged from UCSD06/07 over time (L317-320; assuming that UCLA birds are descended from the UCSD population) and that UCSD songs remained consistent over that same time frame (L311-316). These statements need to be linked more explicitly to specific results.

L481-483: again, it is unclear which result supports this statement that UCLA and UCSD were not similar to each other, when the figure 4 caption specifically states 'no significant differences between UCLA and UCSD 2018-2020.'

Figure 1: Replace 'Natural site' with 'non-urban site' in legend for consistency with figure caption and main text.

Figure 2 caption: when describing panel B, the caption should read 'In the spectrogram, we received...' rather than 'In the power spectra, for the same syllable, we received...' since panel B shows a spectrogram, not a power spectrum.

Figure 4: I agree with the reviewer that the frequency values in panels B and C (and possibly D) are not log₁₀-transformed. For example, the median minimum frequency in panel B is shown as approximately 1.2, which, when back-transformed to the original scale, would be 15.8 kHz (i.e., $\log_{10}(15.8)=1.2$). I notice that the frequency scale on panels C, D, and E of what is now figure 3 have also changed from the previous version of the manuscript, and now appear incorrect as well. For example, the median minimum frequency in figure 3 panel C is approximately 1.5, which, when back-transformed to the original scale, is 31.6 kHz. These are not realistic minimum frequency values. I suspect that the frequency values shown in figures 3 and 4 have been transformed using the natural logarithm rather than log₁₀. If so, back-transforming 1.2 would give an untransformed value of 3.3 kHz, and back-transforming 1.5 would give an untransformed value of 4.5 kHz; 3.3 and 4.5 kHz are much more typical values for minimum frequency in junco song, having measured them many times myself.

===PREPARING YOUR MANUSCRIPT===

Your revised paper should include the changes requested by the referees and Editors of your manuscript. You should provide two versions of this manuscript and both versions must be provided in an editable format:
 one version identifying all the changes that have been made (for instance, in coloured highlight, in bold text, or tracked changes);
 a 'clean' version of the new manuscript that incorporates the changes made, but does not highlight them. This version will be used for typesetting if your manuscript is accepted.

While not essential, it will speed up the preparation of your manuscript proof if accepted if you format your references/bibliography in Vancouver style (please see

<https://royalsociety.org/journals/authors/author-guidelines/#formatting>). You should include DOIs for as many of the references as possible.

If you have been asked to revise the written English in your submission as a condition of publication, you must do so, and you are expected to provide evidence that you have received language editing support. The journal would prefer that you use a professional language editing service and provide a certificate of editing, but a signed letter from a colleague who is a fluent speaker of English is acceptable. Note the journal has arranged a number of discounts for authors using professional language editing services (<https://royalsociety.org/journals/authors/benefits/language-editing/>).

===PREPARING YOUR REVISION IN SCHOLARONE===

Author's Response to Decision Letter for (RSOS-220178.R1)

See Appendix C.

Decision letter (RSOS-220178.R2)

Dear Ms Wong

On behalf of the Editors, we are pleased to inform you that your Manuscript RSOS-220178.R2 "No evidence of repeated song divergence across multiple urban and non-urban populations of dark-eyed juncos (*Junco hyemalis*) in Southern California" has been accepted for publication in Royal Society Open Science subject to minor revision in accordance with the referees' reports. Please find the feedback from the Editors below my signature.

We invite you to respond to the comments and revise your manuscript. Below the Editor's comments we provide additional requirements. Final acceptance of your manuscript is dependent on these requirements being met. We provide guidance below to help you prepare your revision.

Please submit your revised manuscript and required files (see below) no later than 7 days from today's (ie 11-Jul-2022) date. Note: the ScholarOne system will 'lock' if submission of the revision is attempted 7 or more days after the deadline. If you do not think you will be able to meet this deadline please contact the editorial office immediately.

on behalf of Dr David Wilson (Associate Editor) and Kevin Padian (Subject Editor)
 openscience@royalsociety.org

Associate Editor Comments to Author (Dr David Wilson):

I thank the authors for carefully addressing all previous concerns, and congratulate them on a fine manuscript. A few minor suggestions (line numbers refer to the version without track changes):

L120: add "(approximately 15 years later)" after "time-since-colonization"

L137: change "transpiration" to "transportation"

L169: change "song" to "songs"

L228: change "subtracted by 1" to "subtracted 1"

L243: perhaps add a sentence here or elsewhere in this paragraph to state explicitly that mean song traits per song type, rather than per individual male, was used as the unit of replication in statistical analyses.

L277: change "to similar time since colonization and UCSD over time" to "to each other at a similar time since colonization (approximately 20 years), and UCSD to itself over time."

L294: change "for an effect" to "an effect"

L332: add "(i.e., approximately 20 years after colonization for each)" after "populations"

L340: change "locations presented" to "locations, as presented"

L357: change "pick up" to "detect"

Figure 3 caption: should this say "six characteristics" instead of "five"?

Table 1 caption: Awkward wording; suggest "Number of dark-eyed juncos recorded at each of the 7 locations visited, plus the total number of urban and non-urban male dark-eyed juncos recorded. The total number of non-urban dark-eyed juncos includes 19 natural songs..."

===PREPARING YOUR MANUSCRIPT===

one version should clearly identify all the changes that have been made (for instance, in coloured highlight, in bold text, or tracked changes);

===PREPARING YOUR REVISION IN SCHOLARONE===

-- Ensure that your data access statement meets the requirements at <https://royalsociety.org/journals/authors/author-guidelines/#data>. You should ensure that you cite the dataset in your reference list. If you have deposited data etc in the Dryad repository, please only include the 'For publication' link at this stage. You should remove the 'For review' link.

-- If you are requesting an article processing charge waiver, you must select the relevant waiver option (if requesting a discretionary waiver, the form should have been uploaded, see 'File upload' above).

-- If you have uploaded any electronic supplementary (ESM) files, please ensure you follow the guidance at <https://royalsociety.org/journals/authors/author-guidelines/#supplementary-material> to include a suitable title and informative caption. An example of appropriate titling and captioning may be found at https://figshare.com/articles/Table_S2_from_Is_there_a_trade-off_between_peak_performance_and_performance_breadth_across_temperatures_for_aerobic_scope_in_teleost_fishes_/3843624.

Author's Response to Decision Letter for (RSOS-220178.R2)

See Appendix D.

Decision letter (RSOS-220178.R3)

Dear Ms Wong:

I am pleased to inform you that your manuscript entitled "No evidence of repeated song divergence across multiple urban and non-urban populations of dark-eyed juncos (*Junco hyemalis*) in Southern California" is now accepted for publication in Royal Society Open Science.

Please ensure that you send to the editorial office an editable version of your accepted manuscript, and individual files for each figure and table included in your manuscript. You can send these in a zip folder if more convenient. Failure to provide these files may delay the processing of your proof.

Please remember to make any data sets or code libraries 'live' prior to publication, and update any links as needed when you receive a proof to check - for instance, from a private 'for review' URL to a publicly accessible 'for publication' URL. It is also good practice to add data sets, code and other digital materials to your reference list.

Royal Society Open Science is a fully open access journal. A payment may be due before your article is published. Our partner Copyright Clearance Center's RightsLink for Scientific Communications will contact the corresponding author about your open access options from the email domain @copyright.com (if you have any queries regarding fees, please see <https://royalsocietypublishing.org/rsos/charges> or contact authorfees@royalsociety.org).

on behalf of Dr David Wilson (Associate Editor) and Professor Kevin Padian (Subject Editor).

Follow Royal Society Publishing on Twitter: @RSocPublishing
Follow Royal Society Publishing on Facebook:
<https://www.facebook.com/RoyalSocietyPublishing/>
Read Royal Society Publishing's blog:
<https://royalsociety.org/blog/blogsearchpage/?category=Publishing>

Appendix A

Dear Ms Wong,

The Editors assigned to your paper RSOS-211063 "No evidence of song divergence across multiple urban and non-urban populations of dark-eyed juncos (*Junco hyemalis*) in Southern California" have made a decision based on their reading of the paper and any comments received from reviewers.

Regrettably, in view of the reports received, the manuscript has been rejected in its current form. However, a new manuscript may be submitted which takes into consideration these comments.

We invite you to respond to the comments supplied below and prepare a resubmission of your manuscript. Below the referees' and Editors' comments (where applicable) we provide additional requirements. We provide guidance below to help you prepare your revision.

Please note that resubmitting your manuscript does not guarantee eventual acceptance, and we do not generally allow multiple rounds of revision and resubmission, so we urge you to make every effort to fully address all of the comments at this stage. If deemed necessary by the Editors, your manuscript will be sent back to one or more of the original reviewers for assessment. If the original reviewers are not available, we may invite new reviewers.

Please resubmit your revised manuscript and required files (see below) no later than 23-Jan-2022. Note: the ScholarOne system will 'lock' if resubmission is attempted on or after this deadline. If you do not think you will be able to meet this deadline, please contact the editorial office immediately.

Please note article processing charges apply to papers accepted for publication in Royal Society Open Science (<https://royalsocietypublishing.org/rsos/charges>). Charges will also apply to papers transferred to the journal from other Royal Society Publishing journals, as well as papers submitted as part of our collaboration with the Royal Society of Chemistry (<https://royalsocietypublishing.org/rsos/chemistry>). Fee waivers are available but must be requested when you submit your manuscript (<https://royalsocietypublishing.org/rsos/waivers>).

Thank you for submitting your manuscript to Royal Society Open Science and we look forward to receiving your resubmission. If you have any questions at all, please do not hesitate to get in touch.

on behalf of Dr David Wilson (Associate Editor) and Kevin Padian (Subject Editor)
openscience@royalsociety.org

We are very grateful for all your comments and advice, which have substantially improved the manuscript and conclusions therein.

For easy reference, we have color-coded this letter as follows: The reviewers' original comments are in regular typeface. Our responses are in blue. Line numbers, tables, and figures are shown in red and refer to the PDF manuscript version without Track-Changes, for ease of reading. We also include a PDF with Track-Changes. In addition, we numbered each comment to ease reference for the few times we refer to other comments.

We appreciate the opportunity to revise our manuscript and hope it is now suitable for publication in Royal Society Open Science.

Yours sincerely,

Felisha Wong, Eleanor Diamant, and Pamela Yeh on behalf of all authors

Subject Editor Comments to Author:

1. Thanks for your submission, which all readers think is interesting and useful. They have methodological concerns, some of which may be answered relatively easily, and others of which may require additional work, if you feel this is reasonable. The reject/resub decision is in large measure to give you additional time to consider your response to these concerns. We'll look forward to your revised submission. Best wishes.

We very much appreciate the advice and comments provided by Dr. David Wilson and the referees.

Associate Editor Comments to Author (Dr David Wilson):

2. The study compares the acoustic structure of junco songs recorded in urban versus rural locations, and, for one urban location, at two time periods separated by more than a decade. A strength of the paper is that it includes multiple urban and rural locations, rather than a single location of each type as many previous studies have done. The results show little or no divergence among locations and habitat types.

The reviewers and I agree that the topic is timely and important, and that the paper is generally well-written. However, we also identify several serious methodological issues that make the results and discussion difficult to assess. First, it is unclear if/how individual birds were identified/distinguished during recording (were they banded?), which calls into question the independence of the data.

Thank you for viewing this work as timely and important. We have edited our methods to reflect that the majority of the juncos recorded were individually color-banded (lines 167-169). The ones that were not color-banded were often sedentary in a given territory and surrounded by banded individuals in urban environments, making us fairly confident of their independence. In the case of some portions of non-urban habitat, when individuals were not banded, they were recorded with distance from each other (>30m from each other). We have now further clarified how we have calculated song characteristics per individual, avoiding pseudoreplication (lines 232-234).

3. Second, we all agree that the acoustic measurements (mainly, taking frequency measurements directly from a spectrogram as opposed to using an amplitude-based threshold on a power spectrum) are flawed and need to be re-done (reviewer 2 provides details and key references). Reviewer 2 notes that using a logarithmic scale on the frequency axis may be more appropriate, and I note that call rate should be calculated as the number of syllables minus one, divided by the period of time between the start of the first syllable and the start of the last.

Following your advice and that of the reviewers, we have re-extracted acoustic measurements using power spectra for all recordings. We have also transformed frequency data to be on a logarithmic scale, and included this in all of our analyses. Because the UCSD 2006/2007 dataset was extracted using a spectrogram, analyses comparing the UCSD 2006/2007 population are still using the spectrogram-extracted data, though logarithmically transformed. We have also adjusted trill rate by subtracting 1. The re-analyses were a significant undertaking, comprising 5 months of work by the two first authors, one of whom was working full-time. We have updated our methods to reflect these changes (lines 214-221). Our

conclusions are broadly the same, and we are grateful for the opportunity to provide less biased data and more properly assess our research questions.

4. Third, no attempt is made to quantify noise levels at the different recording sites, which is necessary for assigning sites to urban or rural categories and for assessing the potential effects of noise on song structure. Although it would have been better to take such measures at the time of recording, measurements taken now could still be used to support a categorical difference in habitat types and the use of distance-to-nearest-road as a proxy for noise level. Such measurements may also show that noise levels are not categorical, but continuous, and that regression-style analyses are more appropriate.

While we were not able to return to many of these sites, and the consequences of campus closures due to COVID-19 have inevitably affected noise, constraints in the first author's life, and closures at sites of the Angeles Forest due to the 2020 Bobcat Fire, we attempted to better account for noise by using outside tools. Specifically, we used the online tool "HowLoud," which is used in industry to measure and score noise pollution, to estimate and compare noise pollution across these sites. We describe this in our methods and have removed distance to road from our analyses, due to the excess of statistical analyses and the issues raised here. "HowLoud" demonstrates that our urban sites have substantially more noise than all of our non-urban sites, and that these areas are non-overlapping in their level of noise pollution (lines 132-139).

5. Fourth, the sample sizes of rural populations are very low and a power analysis is needed to determine whether the largely negative results are genuine or simply type II error.

We agree that the small sample sizes, particularly of the rural populations, raise potential issues of type II error in our conclusions. We have adhered to your guidance, as well as that of reviewers, and have made the following methodological and discursive changes to the manuscript: 1. We have removed excess statistical analyses and variables in our models (including those suggested in comment 5), including repository data in all analyses between urban and non-urban populations (lines 237-246); 2. In these models we now have urban/non-urban as our fixed effect and population as a random effect (lines 237-241); 3. We test these models against linear models with no random effect to assess if population matters while also increasing our power (line 277-280); 4. We assess power for the latter linear model, as our sampling design was not conducive to power analyses that included population as a random effect (lines 257-260) and linear models were equally supported (lines 278-280); 4. We focus on effect sizes and include confidence intervals for all effect sizes to determine the relative effect of urbanization on each song characteristic, even if urbanization is not found to be statistically significant (lines 270-280; 284-296); and 5. We have softened our language to more accurately state that we found no strong effect of urban population on each song trait, though there might be small effects we do not have the power to assess (lines 308-315).

6. Finally, the number of statistical analyses is excessive, and many seem unrelated to the main hypothesis about habitat type. There are separate analyses for 6 dependent variables (5 song traits and 1 composite measure), 7 locations, urban versus rural habitats, with and without repository data, with and without populations with small sample sizes, and for 2006/07 versus 2017-2020 data. Rationale for analyzing geographic variation is not provided in the introduction, and the methods do not properly address it (for example, by relating song divergence to geographic distance among populations). I therefore recommend that analyses of geographic variation be removed from the manuscript, and that study population be included as a random effect rather than a fixed effect in all analyses. Inflation of experimentwise type I error due to multiple testing of the same hypotheses will also need to be considered.

Thank you for bringing this important issue to our attention. Beyond the shifts detailed in response to comment 5, we have also added a correction to our alpha in both GLMMs to account for multiple testing of the same hypotheses (line 243). We have moved previous analyses into the ESM in case they would be of interest to any readers, and instead focus on the conclusions from the analyses presented in the updated main text. Even with a correction to our alpha, we have found that the contemporary UCLA population statistically differs from the UCSD 2006/2007 population. Though findings between contemporary UCSD and historic UCSD juncos are not significant with an adjusted alpha, which may be due to issues of power, we focus our discussion on the strong effect sizes and findings that are marginally significant with restraint and nuance (lines 284-295).

7. Given the substantive methodological concerns, it currently is not possible to evaluate the results and discussion, and thus the main conclusions of the study. The underlying acoustic data seem solid (though perhaps insufficient), and it may or may not be possible to address our concerns in revision.

We appreciate the opportunity to address these methodological concerns, re-present our data and conclusions in a more appropriate, accurate, and nuanced way, and hope that these changes have shown our data to be sufficient and conclusions of value to the field.

Reviewer comments to Author:

Reviewer: 1

Comments to the Author(s)

8. The study addresses a timely topic that will be of interest to a broad readership. Moreover, the publication of negative findings is crucial for the further development of the field and thus I feel this study is potentially a valuable addition to the literature. However, there are also some problems with the work.

Thank you for your kind words. We have addressed the problems outlined below with how these findings were determined and presented.

9. A major concern is that the acoustic measurements were taken by hand from spectrograms (c.f. Zollinger et al. 2012 Anim Behav 84:e1-e9). Several studies have pointed out the manifold problems of this practice, such as measurement artefacts and the very low repeatability between and within observers (Zollinger et al. 2012 Anim Behav 84:e1-e9, Grace & Anderson 2015 Behav Ecol Sociobiol 69:253-263, Rios-Chelen et al. 2016 Behaviour 67:145-152, Brumm et al. 2017 Methods Ecol Evol 8:1617-1625). However, the deficient song data in this study is no deal breaker: the problem can be remedied by re-measuring the song recordings with proper acoustic methods (e.g., measuring durations in oscillograms and frequencies at set amplitude thresholds in power spectra, see Zollinger et al. 2012 Anim Behav 84:e1-e9).

We have re-extracted all recorded song using power spectra as outlined in our response to comment 3, above. We thank the reviewer for the plethora of resources outlining the importance of changing our acoustic methods. We have now cited these additional papers (lines 225-228).

10. Another concern is related to the lack of noise measurements. Because the main argument for an urban-rural song divergence is adaptation to anthropogenic noise, similar published studies usually measure environmental noise levels at the recording sites. This study used the distance to closest road as a proxy for noise but I'm not convinced that this is valid because traffic noise is not only determined by the distance to the road but also by the amount of traffic, vehicle speed, the types vehicles, road surface etc.

Moreover, the urban sites in this study were all college campuses which were probably not particularly noisy. Anyway, without noise measurements this is impossible to tell.

Thank you for this valid point. We have added a discussion of noise and noise data in response to comment 4. Based on the HowLoud tool, although our urban campus sites are quieter than their surroundings, they are substantially louder than our non-urban sites (lines 132-139). Nonetheless, due to the lack of meaningful information presented by “distance to road” and the valid concerns raised by reviewers, we have removed that variable from our analysis.

Reviewer: 2

Comments to the Author(s)

11. This article compares acoustic properties of dark-eyed junco song between different urban and non-urban sites in California. Its goal is to test how common or general are urban differences in song. Not having found differences in song between urban and non-urban sites, unlike previous work with this species, the discussion then considers a number of possibilities to explain the results.

The article is mostly well written, but the discussion is somewhat confusing and could be streamlined and better organized. For example, some of the possibilities equated in the discussion are weak and do not apply well to this species, while others are interesting and even highly novel, but the lengths to which they are explained do not reflect their differences in quality or applicability to the species.

We agree with the comments outlined here and below and have taken advantage of your guidance and resources to better streamline our discussion and evaluate what we think may be interesting results. We outline these changes below. Importantly, we have refocused our discussion and conclusion on the differences between UCLA and UCSD 2006/2007 given their high sample size and similar time-since-colonization (lines 411-456). We also further discuss differences between UCSD 2006/2007 and the contemporary population (lines 436-456). We have added needed nuance to the conclusions of our results upon re-analysis.

12. I worked on the same topic and study system (G. Cardoso), and would be pleased seeing new studies developing this further. But I have concerns about the article, the first of which may invalidate its main finding. The main concerns are:

- 1- an extremely low sample size, especially in non-urban sites.
- 2- methods for acoustic measurements with faults and not well-argued.

We outline changes to account for low sample sizes and changed acoustic methods below and in response to comments 3-6 and 9-10. Crucially, we have made our comments more nuanced in terms of the findings, and discuss the possibility that low sample size is affecting our results (lines 308-309).

13. 1- The sample size (number of males) is extremely low for all non-urban populations, and for some urban populations as well, which is troublesome since within-population variation in dark-eyed junco song traits is very wide (reviewed in Cardoso & Reichard 2016). Even when joining populations together and adding recordings from the Macaulay Library, non-urban sample size is still low, and there are complications due to lumping populations (please see comment on line 242). Therefore, it is likely that the dataset affords very low statistical power to test whether song differs between urban and non-urban populations.

If the article is to retain its current focus, power analyses would need to be presented. For power analyses, the within-group variation can be estimated from the current data and, for the expected difference, authors can use results of past studies. For example, urban vs. non-urban differences in minimum song frequency are usually smaller than 100Hz for most species, but in the dark-eyed junco

large differences of up to 500Hz were reported (Cardoso & Atwell 2011 Evolution, Reichard et al. 2020 Anim Behav).

- Only if power analyses show good power to detect population differences of, say, 50 or 100 Hz, are the current statements in the title, abstract and conclusions warranted.
- If power analyses show good power to detect large differences of 400 or 500 Hz, but not differences of 50 or 100Hz, then it can be concluded that such large differences are not commonplace, but it cannot be concluded that there are no consistent urban vs. non-urban differences.
- If power analyses show weak power even for large population differences, then results are inconclusive. If this is the case, it may be useful to consider re-orienting the article to questions that can be addressed with data from the better-sampled urban populations (e.g., see comment on line 367).

Thank you for pointing these issues out and providing further guidance on how to respond to them. We have outlined how we have handled and re-oriented the interpretations of our findings in response to comment 5. We agree that we cannot claim that differences do not exist. However, reanalysis and focusing on effect sizes and confidence intervals suggest that if there are differences across populations, they are not strongly different (lines 308-311; 270-280; 284-297). We have re-oriented our discussion towards highlighting the results comparing better-sampled urban populations (UCLA, UCSD 2006/2007, UCLA 2018-2020). Even when taking into account an adjusted alpha, these populations have significant differences (lines 290-295).

14. 2- Although standard some years ago, there are aspects of the acoustic measurement methods that are now substandard. One is making frequency measurement with the cursor on the spectrogram, because this is not the most objective method and measurements could be influenced by background noise (Zollinger et al. 2012 Anim Behav, Brumm et al 2017 Methods Ecol Evol). I think that frequency measurements from the spectrogram can be used if other more automated methods are not feasible (e.g., the level of noise in many recordings prevents automated measurement) and if it is argued that care was taken (e.g., adjusting the dynamic range of spectrograms to distinguishing traces of song amidst noise; Cardoso & Atwell 2012 Anim Behav). Such a choice, however, needs to be explained.

We have now re-extracted sound using power spectra (lines 214-221), which hopefully accounts for the biases that exist in spectrogram measurements. We also add more detail to the spectrogram method, which was used in data to compare to the UCSD 2006/2007 data (lines 222-228).

15. Another issue is the use of a simple linear Hz scale to measure sound frequency and to compute frequency bandwidth. Since animals perceive and modulate sound frequency on a ratio (log) scale, measuring frequency with a linear Hz scale distorts the measurement of bandwidth, and overestimates differences in maximum frequency relative to differences in minimum frequency (Cardoso 2013 Anim Behav). Reichard et al. (2020 Anim Behav) and Winandy et al. (2021, Front Ecol Evol 9:570420) are recent examples of how to do this, so as to evaluate differences in minimum and in maximum frequency in a comparable scale.

We have now log-transformed our frequency measurements (lines 229-232), which are the data used in all analyses. We also appreciate these references and have now cited them in our manuscript (line 229-230).

Other comments, by line number.

16. 75-77 - This is clear for suboscines, for example, but less generally applicable in the case of oscine song, including junco song (Cardoso and Atwell 2011 Anim Behav and references therein). Since the manuscript deals with junco song, this point should be made with restraint.

Adding restraint to this statement and arguing instead that amplitude and frequency are likely independent in oscines, we have added Cardoso and Atwell (2011) as well as Seaver (1942) in lines 77-79.

17. 83-84 – Not clear; please clarify what is meant here by “geographic distance between these two studies”.

We have added needed clarity to this statement; song sparrow populations seem to have different dialects which could influence differences in findings and potential interaction with urban noise (lines 82-87).

18. 88-90 – This is a strange statement. The junco is one of the species with more work and discussion on urban and geographic song divergence (see also Cardoso & Reichard 2016 for discussion on general, non-urban geographic divergence). I realize that existing work deals with a particular urban population, and that this sentence does include the word “multiple”; still, it sounds strange. Instead, I would expect that the introduction summarizes what is known about urban song divergence in juncos, to then frame the objectives of the current work.

Thank you for this helpful critique, and for the reference which we now cite (line 91). We have re-organized the introduction to focus on what is known about urban and non-urban differences generally in juncos, and specifically in song (lines 89-93). Then, we frame the objectives of the current work and how it can add to the understanding of urban and non-urban juncos, particularly with the addition of data from independent urban populations, particularly that in Los Angeles (lines 95-97).

19. 96-97 – Unclear; please check phrasing.

We have now broken up and expanded on the ideas presented in this sentence (lines 98-104). We hope it is now clearer.

20. 97-99 – Also unclear for the general reader, and the following statements do not clarify what is meant here by “song and in song type use within males but not across males”. Also, it was not explained before that only males sing.

We have clarified and added more information about junco song (lines 98-104) and clarified that though females have been found to sing rarely for territorial purposes, male juncos are common singers (lines 101-102).

21. 144 – More information is needed here. For example, what was the average number of songs per male, and was this similar across populations?

We have added more information on the mean \pm SD of number of songs per male per population (lines 161-165). These vary strongly between populations. However, they seem to vary similarly across urban and non-urban populations.

22. 186-188 – This requires explanation for general readers to understand. In any case, it seems too soon to bring up this point, because the units for statistical analysis (songs, males, song types?) were not explained yet.

We have now explained that the units for statistical analysis are the average syllable per individual male. We present this at the beginning of this paragraph (lines 232-234) to add context for general readers.

23. 192 – Not clear. Please state that that recordings were normalized for (peak?) amplitude.

Yes. Recordings were normalized for amplitude, now better explained in line 210.

24. 193, 198-199 – The method is poorly described. Mentioning “from the spectrogram” and a “selection box” is insufficient for general readers to understand. Please state how you chose the upper and lower limits of the “selection box”, and explain that these correspond to your measured maximum and minimum frequencies. Also, please see main comment 2 about issues with using this method.

We have now added a more complete description of how we used the spectrogram (lines 214-228). We also changed the method used to determine minimum and maximum frequency for purposes of our analysis as we used the power spectra analysis method (lines 214-221).

25. 195 – I would suggest not writing “following Cardoso et al.”, because there are differences in methods. One difference is that this past work measured frequency per syllable rather than per song, which has implications for the precision of measurements. Another is that, as discussed in Cardoso & Atwell (2012 Anim Behav), those measurements involved adjusting the dynamic range of spectrograms to optimize distinguishing traces of song amidst noise. Instead of writing “following ...”, I would suggest describing in more detail the measurement method that was used.

Thank you for raising this issue. We have changed the language here and removed that we were following Cardoso et al. (line 216). Instead, we have expanded on our methods (lines 216-228).

26. 198-199 – I was wondering whether peak frequency should have been measured and analyzed. Unlike minimum and maximum frequencies (comment 2), peak frequency is easy to measure objectively, and analyzing it would make the conclusions of the article more robust.

We have now added and measured peak frequency (line 215), finding no strong difference between urban and non-urban populations in peak frequency (lines 270-272).

27. 200-201 - Please see main comment 2 about issues with computing bandwidth in this manner.

In response to comment 2, comments by other reviewers, and this comment, we have re-extracted all data recorded using power spectra (lines 230-232).

28. 201-203 – This is unclear, especially since earlier (lines 186-188) the manuscript alluded to another source of pseudoreplication (using males rather than song types as statistical units). Using males as statistical units is fine for the purpose of this paper, but this needs to be made clearer, and this averaging procedure needs to be better described and argued. For an individual male, were all its recorded songs averaged irrespective of their song type (in this case, the mean value reflects song repertoire usage), or were songs of the same type averaged and then those values average across song types (in this case, it reflects the mean of the male’s song repertoire)?

We have expanded on the averaging method of each syllable and clarified that it is a reflection of song repertoire usage (lines 232-234). All individual males had their respective songs averaged irrespective of song type; however, when extracting data using the power spectra and visually looking at the spectrogram, individual males often maintained the same song type throughout each song.

29. 213 – “when available” creates cloudiness in this description of models. Please consider re-organizing so as to explain for which models these covariates were used or not.

We have now changed the models to only include urban/non-urban and location (lines 237-256). We have moved previous models to the **ESM**.

30. 232 – The concerns in main comment 1 also apply to these LDAs.

In response to comment 6, we have removed LDA analyses.

31. 240 – The majority of P values in Supplementary Table S1 are =1 or >0.9. These are extremely high, and I would think that they are excessively prevalent even if we were analyzing random data. Is there some type of correction for multiple comparisons that I missed and that pushes up the P values? If so, that should be explained. If not, then perhaps these prevalent $P > 0.9$ simply reflect insufficient sample size and low statistical power for the tests, which should be commented too.

We agree with your interpretations. We have now removed many of these analyses from the study, and moved those with location as a fixed effect to the ESM should it be of interest to any readers. We also provide more discussion on the effects of small sample size, and more nuanced interpretations of our results given our low power and limited population size for many populations (lines 308-315).

32. 242-247 – If there is geographic variation in song, then lumping all these populations in only two groups will increase within-group variation and make this test conservative. Perhaps geographic variation is not expected, and the previous analyses (lines 237, 240) did not find population differences in song, but, with such small sample sizes, that is not a strong argument for this lumping. This point should be discussed.

We have now used a GLMM to account for differences between populations, treating population as a random effect in each analysis while treating urban/non-urban as a fixed effect (lines 237-246).

33. 252 (also lines 258 onwards) – These data suggest ($P=0.06$) that the minimum sound frequency at UCSD decreased by 250 Hz since 2006/7 to 2018/20. What may have happened? Are there noticeable differences in the UCSD environment, traffic or otherwise? Or, as discussed later on (line 367 onwards) this is no longer an island-population, but now immigration and emigration are more common? Perhaps this result should be more highlighted in the discussion (even if only suggestive) to try make sense of the different results between this and past studies.

To our knowledge, we haven't seen any noticeable differences in the UCSD environment, though those recording data were not present at UCSD in 2006/2007. We have now restructured the discussion to highlight differences between UCLA 2018-2020, UCSD 2018-2020, and UCSD 2006/2007 (lines 402-456). We find the comparison to UCLA to be quite relevant given a similar time-since-colonisation and UCLA being in a particularly noisy urban environment (West Los Angeles). We discuss the heterogeneity at UCSD that might relate to why UCSD is no longer strongly different.

34. Also, were the song recordings from 2006/7 re-measured for this study, or were the previous measurements re-used? In the latter case, could this result indicate differences in measuring method (since the method that was used is prone to some subjectivity)?

While we did not re-measure the song recordings from 2006/2007, we did do our best to account for measuring method differences by also using frequency data for UCLA and UCSD 2018-2020 that was taken visually from spectrogram analysis (lines 222-228) and have now acknowledged this potential source of bias (lines 253-256).

35. 264-266 – It is very unclear what these “similar findings” refer to. The purpose of, or need for, the LDA analyses do not seem to have been well argued. Are they needed?

We have now removed the LDAs from the main text and supplement as they have added more confusion and are excessive.

36. 272-275 – Confusing sentence: it starts referring to differences with UCDS in 2006/7, but finishes commenting the result in the previous sentence, and not this one. Please separate and develop the ideas in different sentences.

We have now re-organized this paragraph, more clearly highlighting differences between ideas taking into account sample size issues. We have broken up sentences to separate ideas (lines 308-311).

37. 282-298 – This is theoretically possible on other species, but hardly so for juncos; see Cardoso et al. 2009 and Reichard et al. 2020 for evidence that sound frequency is extremely stable within song type and individual. In this regard, I did not understand the rationale in the last sentence, referring to Reichard et al 2020.

Since this discussion is about junco song, the argument developed here does not appear supported. Also, I did not understand how plasticity would help explain your results, and their difference relative to past work on juncos, without changes in the environment or ecology? (please see comment on line 252).

We have rephrased parts of this paragraph and added clarity that our findings are not in line with expectations of phenotypic plasticity (lines 335-337) unlike studies on other songbird species. Instead, the findings by Reichard et al. combined with our results suggest that songs may not adjust that much to noisy conditions (lines 349-352).

38. 299-308 – I agree partially with the argument here. A weak point of the song sharing argument is that juncos share very few songs with neighbors (Newman et al. 2008; see also discussion in Cardoso & Reichard 2016). The argument about junco repertoires harboring a large variation in frequency is valid, but again it does not explain differences in relation to past work. Instead, the fact that individual and population repertoires are acoustically diverse raises the issue of whether sample sizes were adequate (please see main comment 1).

We have now added clarification and discussed how we don't have the proper sample sizes to truly assess repertoire size and its relationship with urban/non-urban populations (lines 362-363).

40. 309-318 – This argument appears to potentially undermine the design of the study. The choice of urban and non-urban sites appeared clear-cut when reading the first paragraph of the methods, but this paragraph puts it into question. Please clarify whether or not there were clear differences in anthropogenic noise levels between the sites classified as urban and non-urban. Namely, what were the habitats and the exposure to traffic in the non-urban sites, and how does that compare with habitat and exposure to traffic in the non-urban sites used in previous work (Slabbekoorn et al. 2007 Condor, Cardoso & Atwell 2011 Evolution)?

We have now added discussion in the methods that demonstrates that noise is strongly different between urban and non-urban groups, but roughly similar between urban populations (lines 132-139). The discussion in this paragraph speaks to the heterogeneity of urban environments themselves that might underly variation in junco song selection across an urban habitat, even if they are still noisier than non-urban habitats (lines 364-377).

41. 318 – Figure 2 was not mentioned in the results. This figure has problems and lacks explanations: the colors are difficult to distinguish and do not follow a logical order, the acronyms in the color legend were not defined and do not appear ordered logically, there appear to be two UCSD bars but this was not explained, boxplots are not well described (vertical lines are not the full range of data); font size and

overall appearance need re-touching for clarity.

We now mention Figure 2 in line 270 of our results. We appreciate the critique on our data visualization and have included a caption explaining the acronyms and have made changes to the visualization of the figure so it's more easily read.

42. 319-334 – Similar comment to that in lines 309-318. Also, do noise levels differ among the different urban sites? Lines 319-321 appear to belong to this paragraph, rather than the previous.

We have clarified that soundscores indicated higher noise across our urban sites in comparison to our non-urban sites, even if noise levels do vary (lines 132-139). Lines 373-376 were meant to connect ideas to the next paragraph as a transition, and so we have left them in their original location.

43. 335-345 – This too is a reason for not lumping all urban populations in a single group. I think that this information should have appeared in the introduction, and that it should have been used to shape the hypothesis-testing goals and analyses.

We hope that we have properly addressed this by including location as a random effect and urban/nonurban as a fixed effect (lines 237-246).

44. 355-367 – I think I am not partial saying that this paragraph does not adequately interpret past work on the UCSD population. While Slabbekoorn et al 2007 and Newman et al 2008 compared urban songs with a small number of mountain males (16 or less per mountain population), and differences in minimum frequency were sometimes statistically significant and sometimes not, when we studied a larger sample (101 UCSD and 50 LM males) this difference in frequency was very clear (Cardoso & Atwell 2011 *Evolution*, 2016 *R Soc Open Sci*).

We agree with you and have added much needed explanation and references to this section, highlighting the importance of higher samples sizes specifically (lines 418-423). Thank you for pointing out our oversight here, and believe we now have better interpreted past work.

45. 362-363 – This is the most serious of issues, as it may invalidate the message in the abstract, title of the paper, and concluding paragraph. This statement, thus, does not suffice. Please see main comment 1.

We have altered the title and added nuance to our conclusions (lines 424-456), including the concluding paragraph (lines 459-462) clarifying that no *strong* differences were found and, instead, re-oriented towards differences found between populations with large sample sizes.

46. 367-375 – This is the good stuff! A novel and original idea that, unlike the previous paragraphs, might explain the different results in this and past studies. This update on the natural history of the UCSD and other urban populations, together with the different results in this paper vs. past work on the same system, plus the change in minimum frequency in UCSD over the past 10 years, can all combine to support a very interesting case of strong acoustic adaptation to the urban environment when the population is a sedentary island population, and subsequent erosion of that acoustic adaptation when philopatry is not strong anymore. This should definitely be in the introduction and in the results, and be used to shape / focus the hypotheses and analyses, rather than only briefly appearing here.

As is, focused on testing urban vs. non-urban differences in song, the article may be weak and not able to support its conclusions, because the non-urban samples are so small (please see the main comment 1). But the dataset for some urban populations, including UCSD at two different time points, is better, and the article could be re-oriented to focus on comparisons among the well-sampled urban populations.

We appreciate your input and excitement about this part of the work. We have re-emphasized these findings and these questions in the introduction (lines 88-97) and the discussion (lines 424-456). In our updated analyses, the effect size in comparing UCSD 2006/2007 and UCSD 2018-2020 was very small, even incorporating confidence intervals. We do think, particularly given the large sample size of UCLA, that our other questions are of value. However, we have further emphasized these questions specifically and oriented our concluding paragraph towards them (lines 459-465).

47. Figure 4 – These are very interesting new data, and their implications potentially important. I think that these data merit a more careful presentation, for example distinguishing observations during winter (these could be mountain birds in the altitudinal migration) and during breeding and post-breeding season (these would indicate that the resident population had indeed increased). I would suggest a methods section and a results section for these new data.

It is difficult to distinguish small and large pins in such small panels, and I did not understand the difference between blue and red pins (there are many blue pins in places without sightings in the previous time interval...). Also, the maps do not have a scale, and it would be useful to place rectangles in Figure 1 indicating the areas of these maps.

We have added these into the methods and results (lines 300-302) and re-done the maps to include scales and more visual clarity (Figure 4). We used *eBird* data (unique observations of juncos), accounting for only the breeding season (April-July). We have re-done this figure in this manuscript to include vetted data *only*, as recommended by *eBird*, which has strongly decreased sightings of birds included. However, the 2010-2020 data in comparison with recorded breeding range extents do demonstrate an expansion of Juncos in San Diego and in Los Angeles over the past 20 years!

48. 380-381 – But differences among sites are not expected, given the mode of development of junco song, and the pattern geographic variation in song (see figure 13.2 in Cardoso & Reichard 2016).

We have rephrased to clarify that we meant differences *between* urban and non-urban sites (rather than *among* urban or among non-urban sites) (lines 440-447).

49. 382-395 – Perhaps the really biologically-meaningful difference in the results is not the comparison between UCSD in 2006/7 and UCLA in 2018, but the difference between UCSD in 2006/7 and 2018, with the difference with UCLA being just a consequence of that? Please see comment on lines 367-375. I realize that the P value for the longitudinal comparison at UCSD was marginal (0.06) and that a larger sample size of UCSD recordings might be useful. If useful, although at the time we did not publish our data, we would be happy to share it.

We have now shifted to focusing on effect sizes in combination with p-values. In the new analyses, UCSD 2006/2007 did not differ strongly from UCSD 2018-2020, even if a couple of traits are marginally significant. Nonetheless, we still found strong differences between UCLA and UCSD which we think are biologically meaningful given that birds are coping with novel strong urban stressors yet not adjusting song in a similar fashion as UCSD. We are grateful for your generosity; based on our analyses it seems we had a substantial amount of data from 2006/7.

50. 396-408 – This is another paragraph equating an explanation for which there is contrary evidence in juncos (see also comment on lines 282, 299). I agree that these points should be brought up. But, rather than equating these possibilities in long paragraphs to then go back and deny their applicability in juncos, I suggest that they could be stated more briefly and, from the onset, in the context of explaining why they are unlikely explanations in the junco.

Based on your advice and feedback, we have moved this section to earlier in the discussion and trimmed it down (lines 325-334).

51. 409-413 – It is not clear how this relates to the interpretation of the results.

We have now removed this statement.

Appendix B

Dear Ms Wong

The Editors assigned to your paper RSOS-220178 "No evidence of repeated song divergence across multiple urban and non-urban populations of dark-eyed juncos (*Junco hyemalis*) in Southern California" have now received comments from reviewers and would like you to revise the paper in accordance with the reviewer comments and any comments from the Editors. Please note this decision does not guarantee eventual acceptance.

Please submit your revised manuscript and required files (see below) no later than 21 days from today's (ie 10-May-2022) date. Note: the ScholarOne system will 'lock' if submission of the revision is attempted 21 or more days after the deadline. If you do not think you will be able to meet this deadline please contact the editorial office immediately.

on behalf of Dr David Wilson (Associate Editor) and Kevin Padian (Subject Editor)
openscience@royalsociety.org

We are grateful for all of the time spent giving us your comments and advice. We have implemented the advice as best as we could and it has substantially improved the manuscript and conclusions therein.

*For easy reference, we have color-coded this letter as follows: The reviewers' original comments are in **bold typeface**. Our responses are in **blue**. Line numbers, tables, and figures are shown in **red** and refer to the PDF manuscript version with Tracked Changes, for ease of reading. We also include a PDF without Tracked Changes. In addition, we numbered each comment to ease reference for the few times we refer to other comments.*

We appreciate the opportunity to revise our manuscript and hope it is now suitable for publication in Royal Society Open Science.

Yours sincerely,

Felisha Wong, Eleanor Diamant, and Pamela Yeh on behalf of all authors

Editor comments:

1. Thank you for your resubmission. We note that many comments from reviewers on a previous submission to another journal were not addressed sufficiently in their view, and that still some lingering concerns remain. We ask you to address these diligently because otherwise we will not be able to consider the manuscript further. Best wishes with your revisions.

We appreciate the advice and comments provided by the handling editor, Dr. David Wilson and the reviewers.

Associate Editor Comments to Author (Dr David Wilson):

2. This revised manuscript was evaluated by one of the original reviewers and by a new expert in the field. Both reviewers and I agree that the revisions were extensive and significantly improved the manuscript, but we also note that several issues remain. The reviewers provide detailed feedback that is consistent with my own assessment of the manuscript. Some of the key issues include clarifying the power analysis, explaining the data/models underlying the background noise estimates, clarifying various aspects of the sound analysis (here a multi-panel figure showing spectrograms of a short song bout and a blow-out of one or two songs, with annotations of bouts, songs, song types, syllables, phrases, and key measurements; another panel showing a power spectrum and its associated measures, and whether the power spectrum is a mean power spectrum based on a syllable or the entire song), how the raw acoustic data were organized and reduced prior to analysis, whether song type should be considered in the analysis, and condensing the somewhat lengthy discussion. In addition to the reviewers' comments, I would like the authors to reduce redundancy in the figures and tables. For example, table 2 shows mean +/- SE per trait per location, but this is largely redundant with figure 2 and could be deleted. Tables 3 and 4 show p-values, effect sizes, and 95% CIs, yet most of these numbers are reported directly in the text; any missing values, such as the 6 p-values from table 3, could be incorporated into the text and the tables deleted. Please address these comments and all of the excellent comments provided by the reviewers.

We are grateful to previous and current reviewers for providing feedback that has led to the significant improvement of our manuscript. The key issues pointed out have helped us revise our manuscript, adding new analyses and interpretation, including noting the limits of our results.

For the key issues Dr. Wilson has pointed out, we have now re-done our power analysis and provided simulations based on the recommendations of previous Reviewer 1 and Reviewers here for most of our song traits. We have expanded our methods section to make these methods clear (lines 382-384), as well as in our results section (lines 477-496). We previously ran into issues of singularity during our power analyses when running simulations. Making our mixed models into Bayesian linear mixed models, and a lower simulation number than we tried previously (100x instead of 1000x) led to a successful power analysis for all traits. We apologize for not explaining these issues in the previous letter, but we are grateful to have had this time to work out issues in our code and troubleshoot the power analysis. Importantly, this process led to us being able to simulate changes in effect sizes, including in determining the power level to detect a change of 500 Hz between urban and non-urban juncos based on the variation in the populations we found (much higher than our effect sizes for frequency traits). We have added this

information for reference in lines 382-384; 477-496 and hope it helps the reviewers and readers interpret our findings, as it has also helped us. Secondly, we have added key information on how ambient noise levels between sites were modeled by HowLoud (lines 147-162). We also clarify that a major limitation is our lack of recorded noise profiles in each recording (line 223; lines 526-528), which we now recommend for future work in lines 821-823. Thirdly, a multi-panel figure is a great idea and would be of help to readers, as well as anyone curious or interested in repeating these methods! We have now made and added this figure as Figure 2. We thank you and reviewers for pointing out places of redundancy and areas where we can cut discussion to better streamline our paper and make it more readable. Taking your advice, we have revised the discussion by substantially editing out less relevant discussion points, based on feedback from editors and reviewers. We have also removed Tables 2-4 and added p-values into our results section text (lines 403-473). We believe these revisions have substantially improved the manuscript and thank you and the reviewers for the time and diligence in commenting.

Reviewer comments to Author:

Reviewer: 2

Comments to the Author(s)

3. I now read the revised version of this article, and find that the authors made a very good, detailed and laborious revision.

One remaining concern, however, is about the power analysis, which I have difficulty interpreting. The method section is very short, only stating the model and name of the package (line 259), and the results (line 273) are also short: “showed high power (>0.99)”. I would like to ask authors to explain what this power analysis does, why only one result is shown for all acoustic traits, and how this result should be interpreted. I would think that statistical power is the function of three things, sample size, within-group variance in the data, and expected among group difference, and that the same model and dataset can have high power to detect a large mean difference between groups, but low power to detect a small difference between groups.

In other words, what is the magnitude of the urban vs. non-urban difference for which these models show such high power (>0.99)? And what would be the power of the models to detect the sort of differences in urban vs. non-urban song generally reported in the literature for other species, or to detect the large difference shown before in the San Diego population (about 500 Hz)? As noted in the previous round of reviews (comment #1), I would think that interpretation of the negative results in the manuscript depends on answering these questions.

Thank you for these important points. We have re-done our power analysis, broadened our methodology section, including effect size thresholds based on the literature and based on calculation of a change in 500 Hz (lines 350-391). We have now added a section to the results specifically outlining our power analysis based on our tested models, and a range of effect sizes to determine how effective our models would be at detecting changes in frequency traits, but lower power for song length and trill rate (lines 477-496). We found high power at detecting urban vs non-urban song with our mixed model for (at least) medium effect sizes (Cohen's $d = 0.05$), but low power at detecting the mean differences we found in most of our results. With the variance in our data and our sample size per group, Cohen's d would be ~ 1.3 , well above this threshold. Thus, we believe we would have sufficient power to find this difference. We have now explicitly stated this in lines 483-489. Previously, our power analysis resulted in issues of singularity when simulating from our linear mixed models. We have since improved in our coding and analytical skills to solve this issue by using a Bayesian linear mixed model and lowering our number of simulations. We thank you for bringing up this point and hope that our work on the power analysis, its explanation, and its finding build more confidence in our conclusions, and provide the reader with more information to interpret these results, as it has aided us in interpreting what differences we may be able to find, as well.

Other comments by line number:

4. Line 31 – In this revised version peak frequency is also analysed.

We thank you for pointing this out and have made this addition accordingly (line 30).

5. 100 – Juncos can have a repertoire of 2 to 8 song types, but usually use only 1 or 2 per bout.

This is an important and relevant comment, especially given how we initially conducted a study based on individual males! Thank you. We have updated our language to better reflect this (lines 107-110).

6. 200-203 – I did not notice this before, but the papers here did not publish the dataset from the 2006/07 breeding season. Reference 70, cited here, published data for minimum frequency, but not for the other song traits. Reference 65, not cited here, did published data for the other traits as well, but averages per song type and not averages per male (and in this manuscript it appears that analyses are done per male?). I think that the full dataset per male was never published, but if useful I can provide it.

For clarity, please state where the dataset comes from, how the dataset was organized (e.g., mean values per male, per song type, per combination of song type & male.), and how it was then analyzed.

Thank you for bringing this up, for your generous offer, and for providing this dataset. Given your comments, as well as that of Reviewer 3, we have now reanalyzed results in the main text with song type as the statistical unit. We hope that this comparison is now more accurate. Our results are generally the same as prior, but we are hoping they are now more accurate without the issues of pseudoreplication.

7. 204-206 - I am not sure I understand which pseudo-replication issue is mentioned in these lines, and why it can “artificially inflate sample size”. The paper cited here (ref. 71) shows that song traits are mostly a property of individual song types and not of individual males (i.e., acoustic measurements are consistent for the same song type irrespective of the male singing it, but measurements are not consistent across the song type repertoire of individual males) and, therefore, argues that song types are better statistical units than males. It is not clear how, or if, this relates with the statement in lines 204-206, and why this statement refers to the UCSD 2006-07 season only, rather than to the other data as well.

Speculating a little, perhaps what happened is that the authors used the dataset from reference 65 (this is not clear; please see comment on line 200) which is a dataset per song type. And, because the number of song types recorded in the UCSD population (168) is larger than the number of males recorded (101), perhaps the authors used song type as a substitute for male and, thus, here mention “artificially inflating sample size”? Please note that variance of acoustic traits across song types is larger than variance across males (ref. 71), so that results of statistical tests may differ if using song type or male as statistical units, and the former should not be used as substitute for the latter. Again, if the dataset per male is useful I can provide it freely upon request.

We apologize for our poor wording that caused confusion. To clarify, we were pointing out a potential pseudo-replication issue in our analysis because we did not have male ID for the 2006/2007 UCSD dataset and therefore thought there would be males counted multiple times if males sang multiple song type, as you have speculated. We have now removed this line (line 256) as we are now testing song type as a statistical unit.

Thank you also for bringing up an important point of focusing on individual males instead of song type. We have now added an analysis of song type (lines 285-291) and our results and implications remain unchanged.

8. 217-218 – Not sure I follow this explanation: “counted the number of syllables, divided by the song length, and subtracted by 1 to obtain the trill rate”. Please clarify.

The most accurate manner of computing trill rate would be the number of syllables minus 1, then divided by the duration from the onset of the first syllable to the onset of the last syllable.

Alternatively, rate could be computed as the total number of syllables divided by the duration from the onset of the first syllable to the end of the last syllable; this second option is slightly less accurate because the last syllable does not complete its cycle (it lacks the small interval afterwards)

Thank you for pointing out our confusing language here. We did indeed compute trill rate with the former, more accurate method. We have re-worded this to more clearly explain what we did. Essentially, we took the number of syllables minus 1, and then, after finding song length duration with a selection box from the onset of the first syllable to the onset of the last syllable, divided by the duration (lines 266-270).

9. 227 – Perhaps remove “likely” here, because there is no ambiguity on the method used.

We have removed the word “likely” (lines 298-300).

10. 245 – Perhaps replace “were grouped as” by “were considered” (similarly to line 246), because data from these sites were not lumped into a single group.

We have replaced “were grouped as” with “were considered as” (line 340).

11. 257 – Why is it that random effects hinder computing statistical power?

We were initially having issues simulating power curves based on the design and low sample size for one location as our initial simulations to calculate power resulted in singularity. However, we have now solved this issue for the majority of traits by building our models through a Bayesian approach (BLMM) and decreasing our simulation number from 1000 to 100 (lines 330-334; lines 386-389).

12. 275-277 – It is not clear how these estimates are interpreted because, although later in the discussion (lines 422-423) it is stated that that “our power analyses and effect size confidence intervals suggest that if differences exist, they are of small effect”, it was not explained why these estimated values are considered small.

Thank you for pointing out needed clarification. We have now calculated Cohen’s d to determine effect sizes, which have established thresholds between “small”, “medium”, and “large” effects (0.2, 0.5, and 0.8 respectively). We have clarified this in the text now as well in lines 351-382. We have also learned how to manually alter the effect sizes of our simulated mixed models based on simulated mean values using our measured variation in traits – which we did not know how to do before – so that we could determine the power of our mixed models. For a reference to previously found differences in minimum frequency change, we have added the expected Cohen’s d based on a 500 Hz shift in song while keeping our found variance within each group and our sample sizes (lines 382-384).

13. 301 – Typo (remove “and in”).

We have now done so (line 500). Thank you for pointing this out.

14. 316 – Typo (appear not to differ...).

Sorry our wording here was confusing and led to misinterpretation. We meant to say that the noise pollution landscape appears to differ between urban and non-urban landscapes—not that song characteristics were different—, so the results that we have are somewhat surprising. We have now changed the wording and hope it is clearer (lines 526-530).

15. 422-423 – If I understand well, the power analysis was done for the main model including all sites across California, and not for the specific comparison discussed in this paragraph (longitudinal comparison in UCSD). Please rephrase to avoid suggesting that statistical power is high for this specific comparison.

Yes, you are correct in your interpretation. We apologize for not clarifying that in text and not conducting a power analysis for the longitudinal comparison as well. We have now added power analyses for the longitudinal comparison and found high power for minimum and maximum frequency, but not bandwidth frequency and trill rate, based on our linear models (lines 477-496). We have added in methodology for these as calculating standardized effect sizes was slightly more complicated because effect sizes were determined by *post-hoc* pairwise calculations (lines 385-388).

16. Figure 2, panel F – Bandwidth cannot be negative (cf. line 231: “we subtracted the log10 transformed minimum frequency from the log10 transformed maximum frequency”).

We have now updated this figure (now Figure 3) to reflect the correct methodology as well as use song type as the statistical unit. Thank you for bringing up this oversight.

17. Figure 3 – Please check the scale of the Y axis. Values do not appear to be in log10kHz (compare with axes in figure 2).

The scale for Figure 3 (now Figure 4) does not match Figure 2’s (now Figure 3) scale because of the difference in the approach used to find the values. In Figure 2 (now Figure 3), the minimum, maximum, and peak frequency were obtained via a power analysis approach while in Figure 3 (now Figure 4), the minimum, maximum, and peak frequency were obtained via spectrum analysis because, for Figure 3 (now Figure 4), we did not have the raw recordings and the dataset we used for UCSD 2006/2007 were from songs that were analyzed via a spectrogram. We have outlined this methodology in lines 292-300.

18. Figure 4 – The figure reads better now, but some weaknesses remain:

- Panels referring to the 1980’s and 1990’s are empty even in the native range, where juncos did exist. This probably means that there are no reliable data on ebird for those decades, making those panels not useful or maybe even misleading. If this is true, please discuss it in the text, and perhaps consider removing these panels. Keeping only the panels for the 2000’s and 2010’s appears sufficient for the purposes of the article.

- Font size in the maps and, especially, the scale are too small to be readable. Please consider increasing font size, especially in the scale.

We appreciate this advice and agree with your view on what should be included. We have now redone these panels, focusing only on 2000s and 2010s, creating a clearer background to highlight the points on the map in new Figure 5. We have added a larger scale, as well, and hope it is now more legible to the reader.

Reviewer: 3

Comments to the Author(s)

19. In this study, the authors use the songs of seven populations (4 urban, 3 rural) of dark-eyed juncos in southern California to test for differences between urban and non-urban juncos. They measured 6 different song characteristics and found limited evidence for differences. This result was surprising given the assumed differences in song transmission constraints between the urban and non-urban sites and previously published comparisons among junco populations in the same area. They also compare one more recently established urban population to an older urban population at two different time points.

This article has already been reviewed by two additional reviewers and extensive revisions have been made that improve the paper. However, many concerns remain.

Thank you for your detailed comments that will help the readability and accuracy of our manuscript.

Major Points:

20. The sound analysis and sampling effort remain difficult to follow, and some of that confusion is rooted in the use of different terms such as song, song bout, and syllable. It's not clear to me whether the measured unit was "songs" or "syllables" (see more detail below in comments on L232-234). Was every recorded song for each individual male measured and then averaged before the analysis? That approach would have the effect of wiping out a large amount of variation given that each male typically sings multiple song types that are acoustically distinct (see points below and Cardoso et al. 2009). Song types are generally differentiated from one another by visual inspection of the shape of the syllable being repeated in the trill. Also, not accounting for shared song types across males still leads to pseudoreplication because the same song type is being sampled multiple times. This concern is important because the sample sizes in some of these populations are already quite low.

We apologize for the confusion in our methods, here. We measured syllables and then averaged syllables across males (now song types). We have clarified this approach more in lines 285-291 and explain that the males would generally sing with 1 or 2 song types at a time (lines 107-110), which limits the variation. In our recordings, each male that we had recorded, despite being recorded on different days and different times, visually sang a consistent song type. However, the issue you and Reviewer 2 bring up of variation and statistical unit has led us to re-do our analyses considering song type, rather than individual male, as the statistical unit (lines 303-304).

21. Related to point 1, providing the recording effort measured as "song bouts" is not as informative as total number of songs and total number of song types recorded across all bouts (L161-162). The ideal presentation here would be to include the average number of songs per male, the range of the number of songs recorded per male, and the number of song types recorded per individual male (both average and range). Also, in L162-165 are you referring to the average number of songs recorded per male?

Male juncos generally sing 1 or 2 song types within one song. In our recordings, we found that one male would generally only sing 1 type throughout the recordings we had, even if the recordings were taken on different days and times. We have now catalogued the song types and included information about the average number of song types per male \pm SD (Table 1). In those lines, we were referring to the average number of song bouts and have now clarified this in text.

22. The “HowLoud” data are informative, but the manuscript does not provide enough context for the reader to evaluate the validity of those data. How are the measurements collected? In the discussion, the authors undercut the argument that these sites have distinct noise profiles by arguing that urban encroachment on rural areas could explain the lack of differences in song characteristics. It is important to note that detailed measurements of these noise profiles are lacking in the current study and essential for the future.

Thank you for your diligence in pointing out our lack of context for ambient noise data. We have now added more detail for how HowLoud gets their measurements into our methods (lines 147-162). Briefly, HowLoud utilizes an established transportation noise model from the Federal Highway Administration to model ambient noise taking into account landscape variables and traffic. They also take into account local sources of noise (e.g., commercial sources) and proximity to airport(s). Based on the model they cite, decibels associated with these sources are modeled over landscape variables and are aggregated and scaled to create ambient noise “soundscores”. The model HowLoud relies upon is used in industry by law in California and in public planning to determine relative noise pollution. We agree that noise profiles are important for future work and have now added this suggestion into our Discussion and Conclusion (lines 526-528 and 821-823).

23. Is a power analysis not possible for the GLMM? Reviewer 2 laid out a detailed approach to a power analysis, but that recommendation seems to have been mostly ignored without much justification. I have almost no experience in this realm of statistics, so am unable to advise.

We are appreciative of your and Reviewer 2’s feedback and have spent time resolving initial issues in our power analysis. We initially were having difficulty simulating power curves, as they were resulting in singularity. We were having trouble determining how to artificially change effect sizes in specific model variables based on an artificial mean value. We therefore went with a package in R where we could artificially change effect sizes that are more interpretable than mixed model coefficients (e.g., F or Cohen’s d) though this model did not support random variables with uneven sample sizes. However, we have improved our coding skills and have solved the issue of singularity by building Bayesian linear mixed models, decreasing our simulation number from 1000 to 100, utilizing different packages to determine coefficients that correspond to Cohen’s d effect sizes (emmeans package), and setting a random seed that worked our analyses and is repeatable. We have also determined how to change effect sizes within an LMM object (such as a BLMM) so that we could simulate whether we could capture a 500 Hz shift in frequency. We have now redone the analyses and revised our description in methods in lines 385-391 and results in lines 477-496.

24. The discussion section could be more concise and focused. I include many suggestions below.

Thank you for these helpful suggestions. Your feedback has made our discussion is more streamlined.

Minor Points:

25. L31 – Peak Frequency?

We have now added peak frequency accordingly (line 30).

26. L68-69 – Please combine these paragraphs.

We have combined these paragraphs. Thank you for the suggestion (line 67).

27. L71-73; 75-76 – Suggest combining these ideas. The link between increased amplitude and increased frequency is the Lombard Effect. Perhaps move the intervening point about song length and frequency to the end of this paragraph as another example of independence between frequency and other song traits?

Thank you for this suggestion on how to combine ideas and better our writing. We have now moved the interjecting points about song length and frequency independence to the end of the paragraph (lines 73-79).

28. L100 – “with approximately two to eight songs per bout” – it would be more consistent with the literature to refer to this as repertoire size rather than songs per bout. For example, males will often sing only one song in a single song bout. They do not necessarily cycle through their entire repertoire in every song bout.

We have now updated our language accordingly to explain that this is repertoire size (lines 107-110).

29. L102-108 – Instead of picking two studies here, I recommend focusing on broader trends. Geographic variation in junco song has been reviewed previously by Cardoso & Reichard 2016, so it might help to highlight that work, which includes data from populations other than the urban v. rural comparison in San Diego.

It seems important to note that junco song repertoires are small, but song type sharing is low, which means that there is a large amount of variation between males. Also, later work by Cardoso and colleagues, with much larger sample sizes, found clear differences in minimum but not maximum frequency. See Cardoso and Atwell 2011, *Evolution* 65-1: 295–300, and Cardoso et al. 2009 *Behav. Ecol.* 20:901–907. This disparity in results across years also presents an opportunity to emphasize how sampling at different points in time could lead to different outcomes as populations potentially evolve in response to noise and other pressures. It also highlights the potential importance of sample size. Both issues are highly relevant to this study.

Finally, in this paragraph, it seems helpful to refer to the population as “San Diego” rather than “urban juncos (L105, 110, etc)” because this is a population that features prominently in this study.

Thank you for your helpful suggestions on how to improve this portion of our introduction. We have now added more clarity and focused more on the broader trends as suggested (lines 110-114). We also made a note that junco song repertoires are small and that song type sharing is quite low, leading to a large amount of variation (lines 107-114). We added the clarification of “San Diego urban juncos” as opposed to just “urban juncos” (line 115).

30. L112 – The Reichard et al. study emphasized not only parental effects, but also early life exposure to noise as a mechanism maintaining this divergence.

We have now changed the wording to better incorporate this information (lines 120-122).

31. L118-119 – I’m not sure if the second point is tested in this study. The authors make assumptions about differences in habitat and noise, but there is very little quantitative data presented in support of either assumption.

We agree with this comment and thank the reviewer for pointing this out. We have removed this second point entirely (lines 129-135).

32. L119-120 – “whether changes in song...” This study compares whether two urban populations with similar time since colonization produce similar songs. This comparison does not really assess how much the songs in each population have changed since colonization, because there isn’t a baseline comparison for either population. Basically, we have no idea how similar the songs of the founders were in each population and cannot assess how much they have changed. I recommend striking “changes” from this point.

We have done as suggested and removed the word “changes” (line 132), which, we agree, is not what we are testing.

33. L132-135 – What is the actual source of these measurements? Is “HowLoud” relying on decibel meters on the ground or some other measure? This tool will be unfamiliar to most and more background is needed here.

We apologize for not explaining HowLoud well enough in our previous submission. We have now added more detail and background for how HowLoud gets their measurements into our methods (lines 147-162). These do not directly rely on decibel meters on the ground; they are based on modeling sources of noise based on the Federal Highway Administration’s Traffic Noise Model. This model takes sources of noise (traffic, airports, local other sources such as restaurants) and landscape variables to estimate ambient noise. As such, estimated ambient noise is based on data around decibel levels from different noise sources and aggregated. It is legally required to be used in the State of California to measure potential noise pollution and is used by public planning.

34. L137-139 – It’s very difficult to interpret these data as presented. Can the sound scores be anchored to specific decibel levels?

We asked HowLoud for specific decibel levels but they would not give them to us. However, they did give us access to components of their soundscores (e.g., traffic score, local sources score, and airport score). Here, soundscores are most heavily influenced by traffic noise. We have accessed documentation for the models HowLoud is based on and an example of that model in a city (San Francisco). Soundscores should be anchored to specific estimated ambient decibel levels; soundscores are objective across space. We have cited these sources in the main text where we have added more explanation to the HowLoud soundscores (lines 147-121). In addition, we now bring up the need for noise profiles in future work in our Discussion and Conclusion (526-528 and 821-823).

35. L162-164 – “Mean number of songs...” – unclear what is meant by “songs” here. Song types? Songs recorded? In the previous and subsequent sentence the unit is song bouts.

Thank you for pointing out that this was unclear. Here, we are referring to the mean number of song phrases recorded per male on average for each location. We have now added some clarification to that section (lines 191-196).

36. L165 – Please move the definition of “song bout” earlier so that it appears before the term is used for the first time.

We have moved the definition of “song bout” to be earlier in the passage so that it is defined prior to its use (lines 190-191).

37. L168-169 – Unclear what is meant here. If accurate, perhaps change to “In urban locations, males were identified and differentiated by color bands. A subset of males (N=X) were unbanded,

but unbanded males consistently defended stable territories neighbored by banded males, which made it possible to differentiate among unbanded individuals.”

You haven’t mentioned anything about the banding status of non-urban males. How were you able to differentiate between males? This issue was explained more thoroughly in your response to the editor. Please include more of that information here.

Thank you for this suggestion! We have now clarified exactly how many were banded/unbanded and how we identified different individuals in both urban and non-urban environments (lines 196-203).

38. L196-198 – Another point to add is that these recordings may have been initially stored in a compressed audio format if they were recorded on a phone, which is probably likely.

This is a good point. We’ve now included it at the end of that paragraph (lines 242-250).

39. L204-206 – And this is a pseudoreplication issue that you cannot control for statistically? For example, you can add song type as a random effect in your models. Please elaborate on the specifics of this issue.

Thank you for this important critique. We have now re-analyzed the data to use song type as the statistical unit instead of each individual male (lines 285-291). We were concerned that the UCSD 2006/2007 dataset was divided by song types and not males, potentially containing more song types than males if males sang multiple types. Our focus on song type as a statistical unit should now eliminate any pseudoreplication issue.

40. L214-215 – What is meant here by “average” minimum and maximum frequency? Below you report that you measured each syllable within a song bout, which could be hundreds of songs and syllables that were then averaged. Also, why is peak frequency the only measure not listed as “average?”

We have now revised to include the fact that we obtained the average peak frequency of each song phrase (lines 285-286). What we meant by average is that we found each male to have sung stable song types and these frequencies we got for those song types were averaged. We now average per song type (rather than across song types per male). We have now revised this in lines 285-291.

41. L225-228 – Despite the limitations here, I think that this approach is reasonable. However, the authors should point out here that these visual measurements are not reliably repeatable among individuals and likely increased the amount of variation within each dataset, which could mask population differences.

Thank you, we agree with the limitations of this approach. We have added this comment in the paper for this section (lines 295-298).

42. L232-234 – Some of this information needs to be moved earlier. A variable number of individual songs were recorded from each male (short v. long bouts). Did you measure every individual syllable in every song for every male and then calculate average values for each male as this section suggests? Averaging individual syllables rather than songs is problematic because you aren’t really measuring the minimum and maximum frequency of the entire song. The minimum and maximum frequency within a trill usually occurs in the middle syllables, so if you average the middle syllables with the rest of the syllables in the song, you will decrease the true frequency bandwidth of the song. Based on your description of your method for creating the selection box (L216-217) and the

available raw data, it seems like you were measuring “songs” not “syllables,” but this section is giving me pause. Please clarify your terminology and methodology.

In addition, averaging multiple song types from each male is not the best approach given what we know about junco song. Junco song types are highly repeatable across males, but the individual songs within a male’s repertoire have distinct acoustic characteristics (Cardoso et al. 2009). So, by average song types for each male, the total variation in each population has been reduced. By shifting the focus to song types rather than individual males, the manuscript might actually increase its sample size because there were likely more song types recorded than individual males.

We appreciate these suggestions and have now clarified the terminology to state that we measured the minimum and maximum frequency (as well as the peak frequency) of the middle syllables for each song phrase (line 266). We have also now added a statistical approach that tests song type as the statistical unit instead of individual male and hope this improves the accuracy of our findings (lines 285-291).

43. L237-246 – Were the Repository songs not included in this analysis? Those songs are not mentioned in the analysis section, but they show up in the methods and in Figure 2.

Thank you for pointing out our omission. The repository songs were indeed included in this analysis, so we have now added a new sentence in the methods section (line 341-343) to explain how they were used in the analysis. We used the repository data to supplement the non-urban mountain site songs that we had in order to compare between urban/non-urban sites.

44. L252-253 – How should these Effect Sizes be interpreted? For example, is there a threshold for what constitutes a small/medium/large effect?

This is an important point, also asked by another reviewer in comment X, above. We have now calculated Cohen’s d effect sizes which have associated thresholds between “small”, “medium”, and “large” effects (0.2, 0.5, 0.8 respectively). For reference, we have calculated the expected effect size of a change of 500 Hz – well above a “large” effect for which we have high power in our models with minimum and maximum frequency as a response variable. We have included an interpretation of effect sizes in the text (lines 350-382).

45. L254-255 – Unnecessary repetition here with line 251.

Thank you for pointing this out. We have removed the repetition.

46. L257-258 – Why is this the case? A previous reviewer gave a detailed explanation for how a power analysis could be undertaken.

Thank you for this comment. This has been noted by you and the other reviewer (comments 3, 11, and 23) We initially were having difficulty simulating power curves, as they were resulting in singularity. We were having trouble determining how to artificially change effect sizes in specific model variables based on an artificial mean value. We therefore went with a package in R where we could artificially change effect sizes that are more interpretable than mixed model coefficients (e.g., F or Cohen’s d) though this model did not support random variables with uneven sample sizes. However, we have improved our coding skills with these simulations and have resolved these issues by building Bayesian mixed models that do not result in singularity, decreasing our simulation number from 1000 to 100, utilizing different packages to determine coefficients that correspond to Cohen’s d effect sizes (emmeans package), and setting a random seed that worked for the majority of our analyses. We have also determined how to change effect sizes within an LMM object (such as a BLMM) so that we could simulate whether we could

capture a 500 Hz shift in frequency. We have now redone the analyses and revised our description in methods in lines 382-384 and results in lines 477-496. We hope our re-analysis is sufficient and better qualifies our findings.

47. L271-274 – If location is eliminated as a random effect, do these results still hold?

Yes, great point. We ran the models with and without location as a random effect. The results are generally the same. We have now mentioned this in lines 389-391, 422-423.

48. L273-274 – This power analysis was the same for all variables measured? Also, it seems like the current approach has only assessed power for the urban v. nonurban comparison, which has a much larger sample size than the individual population-level comparisons. If accurate, you should clarify here that you did not assess power for the among populations comparisons.

In our revised power analysis, each model was independently used for a power analysis so that differences in variance for each group could be accounted for. For this case, we ran a power analysis for the mixed model, which has the urban vs. nonurban comparison as a fixed effect and individual populations as a random effect. We have added a methods paragraph explaining our power analysis in lines 385-391.

49. L306-308 – Peak Frequency?

We have added that in (line 506).

50. L308-309 – It would also be useful here to highlight the point that dark-eyed junco song is highly variable due to the large amount of innovation and improvisation that happens during song development. As a result, few males share songs with their neighbors. This amount of variation adds risk to small sample sizes by increasing the likelihood that outliers will be captured and skew the data.

Great point! We have now added this point at the beginning of our discussion (lines 507-512).

51. L310 – “real differences between populations” - The power analysis and effect sizes only assess differences between urban and non-urban juncos, so the term “populations” here is misleading because these data are a collection of multiple populations lumped into two discrete categories.

We apologize for the inaccuracy here. We have now updated our terminology to state “differences between urban and non-urban juncos” (lines 512-514).

52. L315 – “similar noisy environment” – This is an untested assumption.

We have now rephrased this statement and added a disclaimer that we did not directly measure noise profiles (lines 526-530).

53. L318-319 – “across urban and non-urban settings...” – this phrasing suggests multiple points of comparison when it seems to be one urban and one rural population.

We have altered this based on your recommendation to clarify that one urban and multiple non-urban populations were compared in the previous studies as three rural populations were considered in the previous study (lines 530-532).

54. L331 – “the urban...” environment?

We have corrected this typo (line 543).

55. L335-352 – This paragraph expends a lot of text on an explanation that is not consistent with the result of the current study (although it wasn’t directly tested because background noise levels were not measured), and not supported by a more direct test of the plasticity hypothesis in juncos (Reichard et al. 2020). I recommend reducing the text in this paragraph by half and cutting most of the discussion of specific species. Essentially, plasticity is one explanation that has some support, but not in juncos.

Thank you for your recommendations. We have overall cut down on the explanation in this paragraph, removing the example species to spend less time focusing on this (lines 547-689).

56. L353-363 – This paragraph has similar limitations to the previous paragraph. Juncos have small repertoires, and they don’t share many songs among males. So, they are very limited in their repertoire-based plasticity. I think this paragraph should be largely removed from the manuscript. The potential for plasticity in song types can be mentioned in the previous paragraph, but again, it isn’t an explanation that is tested in this study, and it’s also unlikely to be prominent in juncos.

We agree and we have now removed this paragraph from the manuscript (line 546). We are hopeful that these edits have made our discussion more concise and relevant.

57. L372 – This example was not statistically significant and doesn’t seem worth highlighting given the small sample size and presumably low statistical power. This paragraph also represents a possibility that is speculative and outside the scope of the study at hand. There are no direct measurements of noise pollution at any of these sites, and even with cases of urban encroachment, the noise pollution is far less severe and consistent.

We agree and have now removed this paragraph (line 546). We agree that noise pollution, even with urban encroachment, is lower in non-urban settings. Instead, we have recommended noise profiles be included and measured in future studies (lines 821-823).

58. L396 – Important to note here that 1980 to early 2000s was enough time for distinct difference to appear between UCSD and distant rural populations.

We have now revised to note this in the corresponding paragraph (lines 710-712).

59. L391-410 – These two paragraphs could be combined to eliminate repetition. The Zollinger et al. and Reichard et al. studies both came to a similar conclusion with respect to noise-induced plasticity. The UCSD population was “older” when they were sampled, but only 10-20 years older. So, at UCLA/UCSB/Occidental it could be that insufficient time has passed or some effect of the colonizing population as the authors assert.

We have combined the two paragraphs (line 716).

60. L411-423 – It seems strange here to lead with studies that had smaller sample sizes and were conducted closer to the original colonization event at UCSD. The previous paragraphs just emphasized the point that a lack of population difference could be explained by an insufficient amount of time to diverge. So, it seems prudent to emphasize later studies with larger sample sizes,

more statistical power, and a longer time since colonization.

Thank you for this suggestion to improve the structure of our discussion section. We have removed this as differences between UCSD 2006/2007 and 2018-2020 did not hold in our new analysis. We also now better emphasize the discussion on the time since original colonization event at UCSD.

61. L436 – “found non-statistically significant difference” – change to “found no detectable differences”

We have changed the wording to be clearer (lines 740-742).

62. L436-437 – In this newer analysis, UCSD 2006/2007 was not directly compared to the other sites besides UCLA, so this statement doesn’t seem appropriate.

You are correct. We have changed the wording accordingly to make it clearer (lines 740-742).

63. L441 – “despite strong differences in noise” – please note that these differences were not actually measured.

We have clarified that these differences were not actually measured but were estimated (lines 744-747).

64. L448-451 – This paragraph is mostly spent arguing why this result might not be meaningful without any interpretation of why we might expect the population to shift over the intervening decade. It might be worth linking this potential difference to the nice description in the previous paragraph relating to how these urban populations are no longer islands and an increase in gene flow/immigration could shift some of these traits even if the noise profile remains constant.

Thank you for this helpful suggestion. We have now linked this to the description in the paragraph prior (line 790).

65. L453-454 – change “breeding seasons” to “sampling years”

We have now made this change (lines 794-795).

66. L460-462 – “did not adjust their song characteristics in response to urban noise in a similar way...” – an underlying assumption here is that the founding populations had similar existing variation in song and/or the acoustic environments at UCLA and UCSD are similar. Neither assumption is tested in this study. I recommend eliminating this interpretation. This study shows that these two populations differ in their song structure, but the underlying causes and evolutionary history of any acoustic shifts remain unknown.

We agree and have now eliminated the interpretation.

67. Conclusion – I recommend highlighting the potential for continued shifts over time in all these populations. The two UCSD comparisons were interesting and should not be totally discounted here. Also, it is important to highlight the importance of measuring noise pollution alongside song to better assess environmental selective pressures and the potential for noise-induced plasticity.

We have now edited the conclusion to highlight the possibility that there will be further shifts over time in all of the studied populations and the importance of measuring noise to assess potential noise-induced plasticity (lines 818-826).

68. Figure 2 – “Repository” is not explained in the figure caption. Given that the new analysis has effectively collapsed all of these individual populations into separate urban and non-urban groups, I recommend adding an additional set of bars to each graph that shows the data for the entire urban and non-urban groups.

Thank you for the feedback on how we can clarify our figures given the changes in our manuscript. We have revised our figures accordingly to include the urban and non-urban group bars (now Figure 3). We have explained Repository in the figure caption (now Figure 3).

69. Figure 4 (now Figure 5)– This is an amazing visual of how rapidly these populations have expanded!

Yes, it was surprising to us!

Appendix C

Dear Ms Wong

The Editors assigned to your paper RSOS-220178.R1 "No evidence of repeated song divergence across multiple urban and non-urban populations of dark-eyed juncos (*Junco hyemalis*) in Southern California" have now received comments from reviewers and would like you to revise the paper in accordance with the reviewer comments and any comments from the Editors. Please note this decision does not guarantee eventual acceptance.

Please submit your revised manuscript and required files (see below) no later than 21 days from today's (ie 06-Jun-2022) date. Note: the ScholarOne system will 'lock' if submission of the revision is attempted 21 or more days after the deadline. If you do not think you will be able to meet this deadline please contact the editorial office immediately.

on behalf of Dr David Wilson (Associate Editor) and Kevin Padian (Subject Editor)
openscience@royalsociety.org

We are very grateful for the effort, time, and comments given. We hope we have thoroughly addressed all of the comments given.

*For easy reference, we have color-coded this letter as follows: The reviewers' original comments are in **bold typeface**. Our responses are in **blue**. Line numbers, tables, and figures are shown in **red** and refer to the PDF manuscript version with Tracked Changes, for ease of reading. We also include a PDF without Tracked Changes. In addition, we numbered each comment to ease reference for the few times we refer to other comments.*

We appreciate the opportunity to revise our manuscript and hope it is now suitable for publication in Royal Society Open Science.

Yours sincerely,

Felisha Wong, Eleanor Diamant, and Pamela Yeh on behalf of all authors

Editor comments:

1. Thank you for your attention to the reviewers' comments. As you see, our AE feels that the MS has been much improved but also that there remain some issues to address. Please do so as effectively as you have done with previous comments, because we will not be able to entertain the manuscript further if these are not addressed. Best wishes.

Thank you. We have now addressed these new comments carefully.

Associate Editor Comments to Author (Dr David Wilson):

2. The authors have invested considerable effort and addressed most of the comments raised in the second round of review. However, there remains several issues that need to be addressed. Line numbers in my comments below refer to the version without track changes.

3. L99: change wording to '...song types, though singing males usually use only...'

Thank you for this suggestion. We have made this change (lines 101-104).

4. L134: this sentence could be made more informative because it is still unclear whether HowLoud is interpolating among SPL values measured on the ground at certain locations, or modeling based on the distribution of known noise sources. I think it is the latter. Perhaps use similar wording to that used in your response letter (assuming it is your own and not derived directly from HowLoud): eg, 'HowLoud utilizes an established transportation noise model from the Federal Highway Administration to model ambient noise, taking into account landscape variables and traffic. The model takes into account the presence and distribution of local sources of noise (e.g., commercial sources) and proximity to airport(s).' The key point is that noise level estimates are derived from a model, not interpolated or extrapolated from actual noise measurements taken nearby the study sites.

Yes, you are correct in that HowLoud values are based on the distribution of known noise sources rather than values measured on the ground. We apologize for not making this clearer. We have changed the wording to reflect what we said in the response letter and now directly state that noise levels are derived from models rather than extrapolated from actual noise measurements close to the study site (lines 138-140).

5. L139: the sentence has contradictory statements; I believe it should say '50 (very loud) to 100 (very quiet)

You are correct. We thank you for pointing this out and have made the appropriate changes (lines 158-159).

6. L164: cite figure 2 when defining 'phrases' (and on L240 when describing spectrogram frequency measures). More generally, I think the term 'phrase' is somewhat confusing. I believe most researchers would consider the 'phrase' depicted in figure 2 as a 'song', and multiple songs as a song bout. Confusion persists because the term 'song' is used elsewhere, for instance, in the

measure 'song length' on L223. I recommend using 'song' instead of 'phrase' throughout, but, at a minimum, being consistent.

We understand the confusion and so have now changed the wording to song here (lines 187-189). We have also replaced phrase with song throughout the paper including Figure 2.

7. L166: delete 'approximately' since it is a calculated average, and provide the range in the number of songs recorded per male as requested by the reviewer.

We have now deleted the word “approximately” as suggested (lines 189) and provide the range in the number of songs (and song types) recorded per male across all populations (lines 187-189) while keeping mean and SD values per male per population we previously reported (lines 189-192).

8. L218 (and throughout): replace 'power spectra' with 'mean power spectrum' since your spectrum is based on a selection of time rather than an instant of time (both are possible in Raven). Use 'spectrum' instead of 'spectra' because 'spectrum' is singular and better matches the singular 'spectrogram'.

Thank you for pointing this out. We have replaced power spectra with mean power spectrum throughout the paper.

9. L221-223: reword as: '...(1) minimum frequency, (2) maximum frequency, and (3) peak frequency of the middle syllable within each phrase, as well as the (4) length and (5) trill rate of the entire phrase.' [since length and trill rate are not based on the middle syllable]. Also, I would remove the averaging from this sentence since the sentence describes the raw measures. The averaging is explained separately below.

We have now reworded this phrase accordingly and removed the word averaging from the sentence (lines 254-256).

10. L229: change 'relative from' to 'relative to'

We have changed the wording here to be “relative to” (line 272).

11. L230-231: Change wording to 'We measured minimum, maximum, and peak frequency from each phrase (or song?) within a song bout, and averaged...' Also, clarify whether averaging was done across all phrases recorded per song type from each male, or from across all urban or all rural males. I think averages were calculated across all phrases of the same song type from among all urban males, and separately from among all rural males, but this is still unclear. Perhaps a simple example of how the averaging was done would help.

We appreciate the attention to detail in making our paper clearer. We have now changed the wording accordingly. Averaging was done across all songs recorded per song type from each male – not across all urban or all rural males. We have clarified this, as well, and have now included an example (lines 173-278).

12. L235-236: need to provide the mean, sd, and range for the number of song types recorded per male, as requested by reviewer 3.

Thank you for pointing out this oversight. We have now added information on the mean, SD, and range for number of song types recorded per male in the population as a whole (lines 283-284).

13. L237: the wording here indicates that UCSD06/07 was compared to UCSD18-20 and to UCLA, but does not reflect that UCSD18-20 was also compared to UCLA (as expected under objective 2). Perhaps re-word as 'For the comparisons among UCSD06/07, UCSD18-20, and UCLA frequency measurements,...' Related to this, the comparison of UCSD18-20 to UCLA appears to be missing from the results (L309-323), though it is mentioned briefly in the figure 4 caption (i.e., that there were no significant differences) and referred to in the discussion (L449). Finally, it is now clear how comparing UCSD06/07 to UCLA relates to the objectives in the introduction. Objective 2 implies a comparison between UCSD18-20 and UCLA, and objective 3 implies a comparison between UCSD06/07 and UCSD18-20. Need to align the description of this analysis with the objectives.

We apologize that we were unclear here. UCSD was colonized in the 1980s whereas UCLA was likely colonized in the early to mid 2000s, potentially independently. We have clarified this assumption in the discussion (lines 491-494) and clarified our question in lines 123-124. As such, we compared UCSD 2006/2007 to UCLA and to UCSD contemporary populations. We did not compare UCLA and UCSD 2018/2020 in this analysis, however we do present pairwise comparisons for all contemporary populations in the supplement. Adding more comparisons would lower power and bring up more issues of multiple comparisons. Given our study question 2 about time-since-colonization, we did not need to do a post-hoc pairwise comparison across all three groups, and could instead compare to the "reference" level 0 in the model: UCSD 2006/2007. We have now added some explanation in text (lines 330-334) to clarify this.

14. L249: this sentence about averaging seems redundant with two paragraphs earlier and could be deleted

We agree with this redundancy and have deleted this line (line 306).

15. L294: change to '...were broadly similar among populations and between urban and rural juncos'

We have changed this accordingly (lines 366).

16. L324: need to mention briefly here what was different about these alternate analyses.

We have now added clarification about what was different in the analyses presented in the ESM (lines 414-416).

17. L328: I'm assuming you mean 'When comparing urban and rural juncos,...' rather than the 7 populations? Please clarify.

Thank you for pointing this out. We do mean the analysis comparing urban and non-urban classifications (that takes into account population as a random effect) using a Bayesian mixed-effect model to analyze differences. We have now clarified this in line 436. We do conduct pairwise comparisons between the 7 different populations and now clearly state that when referencing the ESM in lines 414-416.

18. L328-333 and L340-344: the power analyses reported on these lines would benefit from a sentence in the methods explaining how to interpret the values. By 'power', do you mean power to detect a particular effect size? If so, what effect size? Also, please explain how the percentages are to be interpreted? Does 4% mean that the model had only a 4% probability of detecting a difference of a certain effect size in the urban versus rural comparison? If so, these numbers seem very low, and seem to strongly contrast with the relatively high power for detecting medium effect

sizes reported on L333-339. A bit more explanation would be helpful.

Thank you for highlighting what needed further explanation in our new analyses. You are correct that by “power” we mean the ability to detect a particular effect size. We ran power analyses to detect the observed effect sizes (across all analyses), power to detect the effect size a 500 Hz difference (for minimum frequency) would garner, and finally the power to detect a medium effect size. In sum, that we had high power to detect the latter two, we conclude that if frequency shifts do exist in the urban populations generally, they are likely of very low effect size. We have now added clarification in our methods section (lines 347-351) and some clarifying language in our results section (lines 441-443; lines 448-449) We hope this analysis and interpretations are now clearer.

19. L362-364: unclear which result this statement is based upon. When you say 'significant differences between UCLA and UCSD song traits', is this based on the findings that UCSD18-20 is not different from UCSD06/07, whereas UCLA and UCSD06/07 are different? If so, that could be stated more explicitly, especially since UCSD18-20 and UCLA are not different from each other (as noted in figure 4 caption).

We apologize for the lack of clarity and have explicitly stated that we found significant differences between UCLA and UCSD 2006/2007 (lines 485). We have also made the appropriate changes in the caption for Figure 4.

20. L364-365 (and L374): It is similarly unclear which result supports the claim that San Diego juncos shifted their songs upon colonizing the city whereas UCLA juncos did not. There was no comparison of rural juncos and San Diego juncos at the time of colonization in the 1980s, or of rural juncos and UCLA juncos during their colonization in the 2000s. If anything, the results seem to show that UCLA songs diverged from UCSD06/07 over time (L317-320; assuming that UCLA birds are descended from the UCSD population) and that UCSD songs remained consistent over that same time frame (L311-316). These statements need to be linked more explicitly to specific results.

We hope that our new changes have helped clarify our logic and analyses here. You are correct that we did not have data at the time of colonization—just at a similar time-since-colonization. Therefore, we can say that at a similar time-since-colonization, songs were different between these two populations and did not result in the same urban phenotype, but we don't know if they changed in similar directions over time. We assume that UCLA juncos colonized independently, which we now clarify in lines 487-491, given that urban juncos are sedentary and likely from migratory birds becoming sedentary, and that San Diego juncos began as an island population at UCSD, indicative of low dispersal distance (lines 491-494). We have added clarification in the main text that we are pointing to results comparing UCSD 06/07 and UCLA 2018-2020 while adding this important caveat.

21. L481-483: again, it is unclear which result supports this statement that UCLA and UCSD were not similar to each other, when the figure 4 caption specifically states 'no significant differences between UCLA and UCSD 2018-2020.'

Thank you for pointing out the oversight in the Figure 4 caption. We failed to update the caption after our re-analyses. We did not compare UCSD 2018-2020 and UCLA 2018-2020 in these analyses, though pairwise comparisons are presented in the ESM, where we didn't find significant differences between contemporary populations. We were comparing UCSD 2006/2007 and UCLA 2018-2020 to support this statement, where we did find significant differences at a similar time-since-colonization. We hope that our explanation in reply to comment 20, and the additions to text help clarify this logic now. Specifically,

we're looking at the resulting song phenotype ~15-20 years following colonization in both cities. These dates were the mid 2000s for UCSD and the late 2010s, early 2020s for UCLA.

22. Figure 1: Replace 'Natural site' with 'non-urban site' in legend for consistency with figure caption and main text.

This figure has been updated for consistency.

23. Figure 2 caption: when describing panel B, the caption should read 'In the spectrogram, we received...' rather than 'In the power spectra, for the same syllable, we received...' since panel B shows a spectrogram, not a power spectrum.

We thank you for pointing out this error and have fixed it accordingly.

24. Figure 4: I agree with the reviewer that the frequency values in panels B and C (and possibly D) are not log₁₀-transformed. For example, the median minimum frequency in panel B is shown as approximately 1.2, which, when back-transformed to the original scale, would be 15.8 kHz (i.e., $\log_{10}(15.8)=1.2$). I notice that the frequency scale on panels C, D, and E of what is now figure 3 have also changed from the previous version of the manuscript, and now appear incorrect as well. For example, the median minimum frequency in figure 3 panel C is approximately 1.5, which, when back-transformed to the original scale, is 31.6 kHz. These are not realistic minimum frequency values. I suspect that the frequency values shown in figures 3 and 4 have been transformed using the natural logarithm rather than log₁₀. If so, back-transforming 1.2 would give an untransformed value of 3.3 kHz, and back-transforming 1.5 would give an untransformed value of 4.5 kHz; 3.3 and 4.5 kHz are much more typical values for minimum frequency in junco song, having measured them many times myself.

We now understand that within R Studio, the default is a natural log. Thank you for taking the time to write out this detailed comment about the issue. We have now updated our code to reflect a log-10 transformation of all frequency values and have re-done all of our analyses using the log-10 transformation values across the main text and the ESM. We also have redone **Figure 3 and 4** with the log-10 transformed values.

Appendix D

Dear Ms Wong

On behalf of the Editors, we are pleased to inform you that your Manuscript RSOS-220178.R2 "No evidence of repeated song divergence across multiple urban and non-urban populations of dark-eyed juncos (*Junco hyemalis*) in Southern California" has been accepted for publication in Royal Society Open Science subject to minor revision in accordance with the referees' reports. Please find the feedback from the Editors below my signature.

We invite you to respond to the comments and revise your manuscript. Below the Editor's comments we provide additional requirements. Final acceptance of your manuscript is dependent on these requirements being met. We provide guidance below to help you prepare your revision.

Please submit your revised manuscript and required files (see below) no later than 7 days from today's (ie 11-Jul-2022) date. Note: the ScholarOne system will 'lock' if submission of the revision is attempted 7 or more days after the deadline. If you do not think you will be able to meet this deadline please contact the editorial office immediately.

on behalf of Dr David Wilson (Associate Editor) and Kevin Padian (Subject Editor)
openscience@royalsociety.org

We are excited to learn that our manuscript has been accepted contingent upon the minor comments detailed below. We have now edited the manuscript accordingly and look forward to its publication in Royal Society Open Science.

*For easy reference, we have color-coded this letter as follows: The editor's original comments are in **bold typeface**. Our responses are in **blue**. Line numbers, tables, and figures are shown in **red** and refer to the PDF manuscript version, for ease of reading.*

Yours sincerely,

Felisha Wong, Eleanor Diamant, and Pamela Yeh on behalf of all authors

Associate Editor Comments to Author (Dr David Wilson):

I thank the authors for carefully addressing all previous concerns, and congratulate them on a fine manuscript. A few minor suggestions (line numbers refer to the version without track changes):

Thank you for the kind words and suggested further minor edits to clarify language and correct typos. We have now done so as suggested.

L120: add "(approximately 15 years later)" after "time-since-colonization"

We have now added this clarifying text (line 120).

L137: change "transpiration" to "transportation"

We have now fixed this typo (line 138).

L169: change "song" to "songs"

We have now done so (line 170).

L228: change "subtracted by 1" to "subtracted 1"

We have edited accordingly (line 229).

L243: perhaps add a sentence here or elsewhere in this paragraph to state explicitly that mean song traits per song type, rather than per individual male, was used as the unit of replication in statistical analyses.

Thank you for this suggestion. We have now added a sentence explicitly stating that we used mean song traits per song type, rather than per individual male, as the unit of statistical replication (lines 244-245).

L277: change "to similar time since colonization and UCSD over time" to "to each other at a similar time since colonization (approximately 20 years), and UCSD to itself over time."

We have now made this change and thank you for the wording suggestion (line 279).

L294: change "for an effect" to "an effect"

We have now made this change (line 296).

L332: add "(i.e., approximately 20 years after colonization for each)" after "populations"

We have now made this addition (line 334).

L340: change "locations presented" to "locations, as presented"

Thank you for catching this grammatical error. We have now made these edits (line 343).

L357: change "pick up" to "detect"

We agree that this is a more accurate term. We have edited the sentence accordingly (line 360).

Figure 3 caption: should this say "six characteristics" instead of "five"?

Thank you for this note! It should have read "six." We have now fixed this sentence.

Table 1 caption: Awkward wording; suggest "Number of dark-eyed juncos recorded at each of the 7 locations visited, plus the total number of urban and non-urban male dark-eyed juncos recorded. The total number of non-urban dark-eyed juncos includes 19 natural songs..."

We have now made these changes and hope the wording is clearer and less clunky.